# Amylin deposition activates HIF1α and 6-phosphofructo-2-kinase/fructose-2, 6-biphosphatase 3 (PFKFB3) signaling in failing hearts of non-human primates

Miao Liu [1,2], Nan Li [1,2], Chun Qu [1,2], Yilin Gao [1], Lijie Wu [1] & Liangbiao George Hu [1✉]

Hyperamylinemia induces amylin aggregation and toxicity in the pancreas and contributes to the development of type-2 diabetes (T2D). Cardiac amylin deposition in patients with obesity and T2D was found to accelerate heart dysfunction. Non-human primates (NHPs) have similar genetic, metabolic, and cardiovascular processes as humans. However, the underlying mechanisms of cardiac amylin in NHPs, particularly related to the hypoxia inducible factor (HIF)1α and 6-phosphofructo-2-kinase/fructose-2,6-biphosphatase 3 (PFKFB3) signaling pathways, are unknown. Here, we demonstrate that in NHPs, amylin deposition in heart failure (HF) contributes to cardiac dysfunction via activation of HIF1α and PFKFB3 signaling. This was confirmed in two in vitro cardiomyocyte models. Furthermore, alterations of intracellular $Ca^{2+}$, reactive oxygen species, mitochondrial function, and lactate levels were observed in amylin-treated cells. Our study demonstrates a pathological role for amylin in the activation of HIF1α and PFKFB3 signaling in NHPs with HF, establishing amylin as a promising target for heart disease patients.

---

[1] Department of Translational Safety and Bioanalytical Sciences, Amgen R&D (Shanghai) Co. Ltd., Shanghai, China. [2]These authors contributed equally: Miao Liu, Nan Li, Chun Qu. ✉email: 13661915978@163.com

Type-2 diabetes (T2D) is a chronic metabolic disorder characterized by a progressive defect in insulin secretion from pancreatic β-cells. Obesity primarily affects insulin resistance in target tissues, including the skeletal muscles, liver, and adipose tissue[1,2]. The mechanisms by which T2D and pre-diabetic obesity-associated insulin resistance lead to heart failure (HF) are complex and not well characterized[3–9]. It is known that the pancreatic β-cells compensate for insulin resistance by upregulating insulin secretion[10]. Amylin, also called islet amyloid polypeptide (IAPP), is a hormone co-expressed and co-secreted with insulin by pancreatic β-cells[11,12]. Amylin of humans, non-human primates (NHPs), dogs and cats, but not of rodents (mice and rats), possesses an amyloidogenic promoting region and can form toxic aggregates when overexpressed[12–17]. These small membrane-permeable amylin oligomers cause oxidative[18,19] and inflammatory stress[20,21], contributing to apoptosis in the pancreas[13,22].

Previous studies have reported deposition of aggregated amylin in the hearts of HF patients with obesity or T2D[23–27], in kidneys of patients with diabetic nephropathy[28], and in the brains of patients with Alzheimer's disease[29–33]. Accumulation of aggregated amylin was also detected in the pancreas[16] as well as extra-pancreatic organs, including the heart[23], kidney[34], and brain[30,35] of transgenic rats expressing human amylin in the pancreas (HIP). Amylin incorporation in the heart (as well as in the kidneys and brain) originates in the pancreas as human amylin mRNA has not been detected in the heart[24] or brain[31] of HIP rats and humans. This amylin accumulation may result from blood circulation[27]. Subsequently, HIP rats develop T2D, diastolic heart dysfunction, cardiac hypertrophy, and dilation[23,24]. Recent studies have also shown that accumulation of human amylin in cardiomyocytes leads to cardiomyocyte sarcolemmal $Ca^{2+}$ leakage, independent of diabetic remodeling of the myocardium[27].

Alteration of cardiac metabolism contributes to the development of HF[36–38], which is commonly associated with metabolic inflexibility[37], impaired mitochondrial oxidative metabolism[39], and decreased glucose oxidation[36]. Uncoupling between glucose uptake and oxidation is observed in HF, resulting in an increase in glycolysis and conversion of pyruvate to lactate[36]. These changes then contribute to cardiomyocyte dysfunction[9,37,40]. Hypoxia signaling also plays a critical role in intracellular metabolism and has a considerable impact on cardiac function[41]. Studies have shown that tissue hypoxia develops during cardiac remodeling, and that metabolic reprograming occurs in cardiomyocytes for the maintenance of cardiac function[41–43]. Hypoxia-inducible factor 1α (HIF1α), which regulates the expression of genes involved in glycolysis in cardiomyocytes, is a key regulator of intracellular metabolism[44]. Moreover, 6-phosphofructo-2-kinase/fructose-2,6-biphosphatase 3 (PFKFB3), an induced isoform of phosphofructokinase 2 (PFK2), is known to be upregulated under the hypoxic condition and increases the glycolytic rate in diabetes[45]. In this context, we attempted to determine whether hypoxia signaling pathways are indeed activated in stressed cardiomyocytes.

Although various animal models, including rodents and rabbits, have been used to study the underlying mechanisms of amylin actions, the results obtained from these models cannot be directly translated to humans due to inherent differences such as pharmacology and receptor splice variants that exist between the species[46]. Alternatively, NHPs serve as superior models as their genetic, metabolic, and cardiovascular processes resemble those of humans more closely[47]. As amyloidogenesis of amylin can occur only in humans, NHPs, and cats[17], and as its cardiotoxic effects do not require diabetic remodeling of the myocardium, it is reasonable to assume that amylin may be deposited in the cardiac tissue of NHPs with HF even in the absence of diabetic state. Isolated rat ventricular cardiomyocytes (RVCMs), derived from rat ventricle,

and human induced pluripotent stem cell-cardiomyocytes (hiPSC-CMs) retain their physiological functions and are widely used in in vitro models to study cardiotoxicity[48]. In this study, we aimed to analyze cardiac amylin deposition in NHPs with HF and determine its role in cardiac metabolism, particularly regarding the HIF1α and PFKFB3 signaling pathways. Overall, our findings reveal a pathological role for amylin in myocardial signaling, which may be applicable for the development of HF therapeutic targets.

## Results

**Cardiac functions and metabolic profiles of NHPs.** The cardiac functions and metabolic profiles of the 13 NHPs enrolled in this study were summarized in Tables 1 and 2, respectively. Measurements of electrocardiography (ECG), echocardiography, left ventricular (LV) hemodynamics, and metabolic profiles were conducted right before the animals were euthanized. As listed in the reference standard from the American Society of Echocardiography[49], LV ejection fraction (EF) < 40% is indicative of systolic heart dysfunction. Six animals with EF < 40% were categorized in the HF group in the current study. As compared to those of the healthy control (CTL), both the EF and fractional shortening (which measures the reduction in the length of the end-diastolic diameter that occurs by the end of systole, is a measure of the heart's muscular contractility) in the HF group showed significantly lower percentages, indicating impairment of heart muscular contractility (Table 1). Furthermore, these NHPs with HF had significantly wider QRS complex (a typical parameter for ventricular conduction delay in systolic heart dysfunction), shorter QT(c) interval (a risk factor for ventricular tachyarrhythmias and sudden death), increased LV end-diastolic pressure (LVDP), increased LV end-systolic pressure (LVSP), decreased maximum rate of LV pressure rise (+d$p$/d$t$ max) and fall (–d$p$/d$t$ max), decreased ratio of mitral peak E-wave velocity and mitral peak A-

**Table 1 Comparison of cardiac functions parameters of NHPs.**

| Groups | CTL | HF |
|---|---|---|
| Systolic BP (mmHg) | 119.79 ± 8.89 | 117.48 ± 7.04 |
| Diastolic BP (mmHg) | 75.15 ± 6.32 | 73.93 ± 2.69 |
| Heart beat (bpm) | 181.33 ± 4.42 | 168.38 ± 12.61 |
| PR interval (ms) | 63.09 ± 6.53 | 63.34 ± 16.89 |
| RR interval (ms) | 339.00 ± 8.75 | 335.17 ± 33.11 |
| QRS complex (ms) | 51.53 ± 1.85 | 84.94 ± 6.11*** |
| QT interval (ms) | 287.70 ± 22.24 | 187.60 ± 24.39* |
| QTc interval (ms) | 345.81 ± 21.99 | 242.63 ± 37.79* |
| LVSP (mmHg) | 94.79 ± 1.59 | 110.63 ± 9.44* |
| LVDP (mmHg) | 0.79 ± 0.62 | 11.98 ± 7.17* |
| +d$p$/d$t$ max (mmHg/s) | 4141.67 ± 204.27 | 3173.71 ± 311.99* |
| –d$p$/d$t$ max (mmHg/s) | 3273.58 ± 94.45 | 2357.84 ± 299.59** |
| EF (%) | 68.06 ± 0.63 | 29.05 ± 5.58*** |
| FS (%) | 31.66 ± 0.45 | 16.42 ± 4.96** |
| EDV (mL) | 3.70 ± 0.52 | 13.53 ± 2.12*** |
| ESV (mL) | 1.18 ± 0.16 | 9.87 ± 1.98*** |
| SV (mL) | 2.52 ± 0.36 | 3.66 ± 0.82 |
| LVIDd | 1.52 ± 0.08 | 2.31 ± 0.23** |
| LVIDs | 1.04 ± 0.05 | 1.87 ± 0.19*** |
| MV E/A ratio | 1.56 ± 0.05 | 1.35 ± 0.06* |
| LV mass (g) | 5.14 ± 1.01 | 20.92 ± 5.09** |

Data represent mean ± SEM. *$P$ < 0.05, **$P$ < 0.01, ***$P$ < 0.001 by Student's t test. Control (CTL) group, $n$ = 7; heart failure (HF) group, $n$ = 6. QTc corrected QT, LVSP left ventricular systolic pressure, LVDP left ventricular diastolic pressure, +d$p$/d$t$ max maximum rate of rise of left ventricular pressure, –d$p$/d$t$ max maximum rate of fall of left ventricular pressure, EF ejection fraction, FS fraction shortening, EDV end-diastolic volume, ESV end-systolic volume, LVIDd left ventricular end-diastolic internal diameter, LVIDs left ventricular end-systolic internal diameter, MV E/A ratio the ratio of mitral peak E-wave velocity and mitral peak A-wave velocity, LV mass left ventricular mass.

**Table 2 Comparison of the metabolic profiles of NHPs.**

| Groups | CTL | HF |
|---|---|---|
| Age (years) | 12 ± 2 | 18 ± 1* |
| Body weight (kg) | 4.7 ± 0.51 | 8.3 ± 1.02** |
| BMI (kg/m²) | 23 ± 2 | 47 ± 6** |
| Fasting blood glucose (mg/dL) | 67 ± 6.7 | 93 ± 23 |
| Fasting insulin (μIU/mL) | 21.7 ± 4 | 58.9 ± 27.12 |
| Glycated hemoglobin (HbA1C) (%) | 4 ± 0.09 | 4.35 ± 0.37 |
| Serum albumin(g/L) | 36.38 ± 2.1 | 38.35 ± 1.67 |
| Serum cholesterol (mmol/L) | 2.11 ± 0.09 | 3.80 ± 0.56* |
| Serum triglyceride (mmol/L) | 0.37 ± 0.03 | 0.82 ± 0.14** |
| Serum HDL (mmol/L) | 1.07 ± 0.10 | 1.58 ± 0.33 |
| Serum LDL (mmol/L) | 0.92 ± 0.15 | 2.10 ± 0.89 |

Data represent mean ± SEM. *$P < 0.05$, **$P < 0.01$ by Student's t test. Control (CTL) group, $n = 7$; heart failure (HF) group, $n = 6$. BMI body mass index, HbA1C hemoglobin A1C, HDL high-density lipoprotein, LDL low-density lipoprotein.

wave velocity (MV E/A ratio), increased end-diastolic volume (EDV) and end-systolic volume (ESV), as well as increased left ventricular end-diastolic and end-systolic internal diameter (LVIDd and LVIDs) and LV mass (Table 1). These cardiac changes in the NHPs with HF corresponded to the published pathological symptoms of infiltrative cardiovascular diseases in humans[50]. Table 2 also showed that NHPs with HF had significantly higher body weight (BW) and BMI (> 30 kg/m², considered as obese) than CTL. Serum cholesterol and triglyceride levels were also higher in the NHPs with HF compared to the CTL.

**Myocardial pathology.** Microscopic assessments were conducted on all hearts (Fig. 1a–d, f, g, i, j). No myocardial abnormalities were observed in the CTLs (Fig. 1a). The histopathologic alterations of hearts from NHPs with HF included disorganized arrangement of cardiomyocytes, myocardial interstitial fibrosis (star, Fig. 1b), and hypertrophy of cardiomyocytes with atypical karyomegaly (arrowhead, Fig. 1b). Moreover, infiltration of mixed inflammatory cells comprised primarily of macrophages with other mononuclear cells accounting for a smaller proportion, and neutrophils were rarely observed (arrow, Fig. 1b–d). Infiltrated immune cells and their subtypes were further classified by specific antibodies against CD3 (T cells), CD8 (cytotoxic T cells), CD68 (macrophages), and CD45 (leukocytes) using immunohistochemistry (IHC) staining. Their amounts were significantly higher in the HF group compared with CTL animals (Supplementary Fig. 1). Additionally, randomly distributed, multifocal degeneration and necrosis of cardiomyocytes characterized by loss of striations (square, Fig. 1c, d), arteriopathy with thickening of the tunica media of the vascular smooth muscles and narrowing of the lumen (circle, Fig. 1c), and multifocal arteriosclerosis were also observed in HF group. The microscopic grades were determined using a semi-quantification scoring system as shown in Fig. 1e. Compared with those of CTL NHPs (Fig. 1f, i), larger amounts of fibrotic tissues were detected by Masson's trichrome (star, blue color, Fig. 1g) and Picro Sirius red staining (star, red color, Fig. 1j) in the failing hearts of NHPs. The randomly distributed fibrotic areas were greater in the HF group than the CTL group (Fig. 1h, k, bar graph). Taken together, these pathological findings were consistent with the cardiac dysfunctions assessed using ECG and Echo in the NHPs with HF.

**NHPs with HF presented with cardiac amylin accumulation.** Amylin deposition was confirmed by IHC staining using antiamylin antibody (brown color). Amylin staining in the islets of the pancreas of NHPs was used as a positive control (Fig. 2a and Supplementary Fig. 2a, b). No amylin deposition was observed in the CTL myocardium (Fig. 2b). In contrast, aggregated amylin depositions were observed in the nuclei (Fig. 2c, arrow) and sarcolemma of hypertrophic cardiomyocytes (Fig. 2c, d, arrowhead), infiltrated inflammatory cells (Fig. 2e, simple arrow), and blood cells (Fig. 2f, circle) of NHPs with HF, which is generally observed in infiltrative heart diseases[50]. Figure 2g presented the relative positive signal intensity for amylin in heart samples from NHPs with HF and CTL. We found less amylin depositions in the myocardium of NHPs with HF (Fig. 2c–f) compared to that in the islets of NHPs (Fig. 2a). In addition to the heart and pancreas, we also observed amylin deposits in kidneys of NHPs with HF, but not from CTL (Supplementary Fig. 2c, d). This was consistent with previous studies reporting amylin deposits in the kidneys of HIP rats (with impaired cardiac function) but not from wild-type animals[23]. Treatment of hyperamylinemia dramatically reduced amylin incorporation in the heart and kidneys of HIP rats[34], suggesting amylin may play a role in driving cardio-renal communication.

Less amylin accumulation was expected to be observed in the heart compared to the pancreas since amylin is expressed and produced exclusively in the pancreas[24]. Moreover, myocardial accumulates of amylin is largely facilitated via blood circulation[25,51]. Therefore, we next aimed to investigate the level of amylin in circulation.

**Levels of circulating amylin and its correlations with other HF markers in NHPs.** We used ELISA to assess amylin levels in a parallel set of commercial sera from CTL and HF NHPs (whose heart tissues were not available). Serum amylin level was significantly higher in the HF groups (Fig. 3a). We also assessed the levels of three well-known HF-related biomarkers[52,53]: growth differentiation factor-15 (GDF-15, myocardial inflammation), soluble suppression of tumorigenicity-2 receptor (ST2, ventricular remodeling, and hypertrophy), and cardiac troponin I (cTnI, myocardial necrosis) in these sera. As expected, the levels of all three biomarkers were significantly higher in the sera of NHPs with HF than in that of the CTL (Fig. 3b).

Next, we assessed the relationship between serum amylin levels and those of the three HF biomarkers in NHP sera using correlation estimates with scatterplots. In NHPs with HF, higher ST2 levels correlated significantly with increased amylin accumulation ($r = 0.6887$; $P < 0.01$, Fig. 3c). However, there was no correlation between serum amylin level and cTnI or GDF-15 levels (Fig. 3d, e).

Red blood cells (RBCs) play an important role in the release of oxygen into tissues, and are in intimate contact with blood capillaries. Therefore, previous reports of the presence of amylin deposition in the retinal[54] and brain capillaries[30] of patients with small vessel disease, as well as in the serum and myocardium of NHPs with HF in the present study prompted us to investigate whether the plasma and RBCs of NHPs accumulate amylin in context of HF.

ELISA of matched plasma and RBC lysates from thirteen NHPs (CTL group, $n = 7$; HF group, $n = 6$; whose heart sections were used for histopathological evaluation in Fig. 1 and amylin assessment in Fig. 2) indicated that the amylin level was substantially higher in these two blood components (Fig. 3f, g) than the average level of amylin in CTL NHPs.

Collectively, these data indicate that the circulating amylin level was closely associated with cardiac remodeling and the cardiac hypertrophy marker, ST2.

**HIF1α and PFKFB3 activation in the heart of HF NHPs.** Metabolic alteration is a common feature in the development of

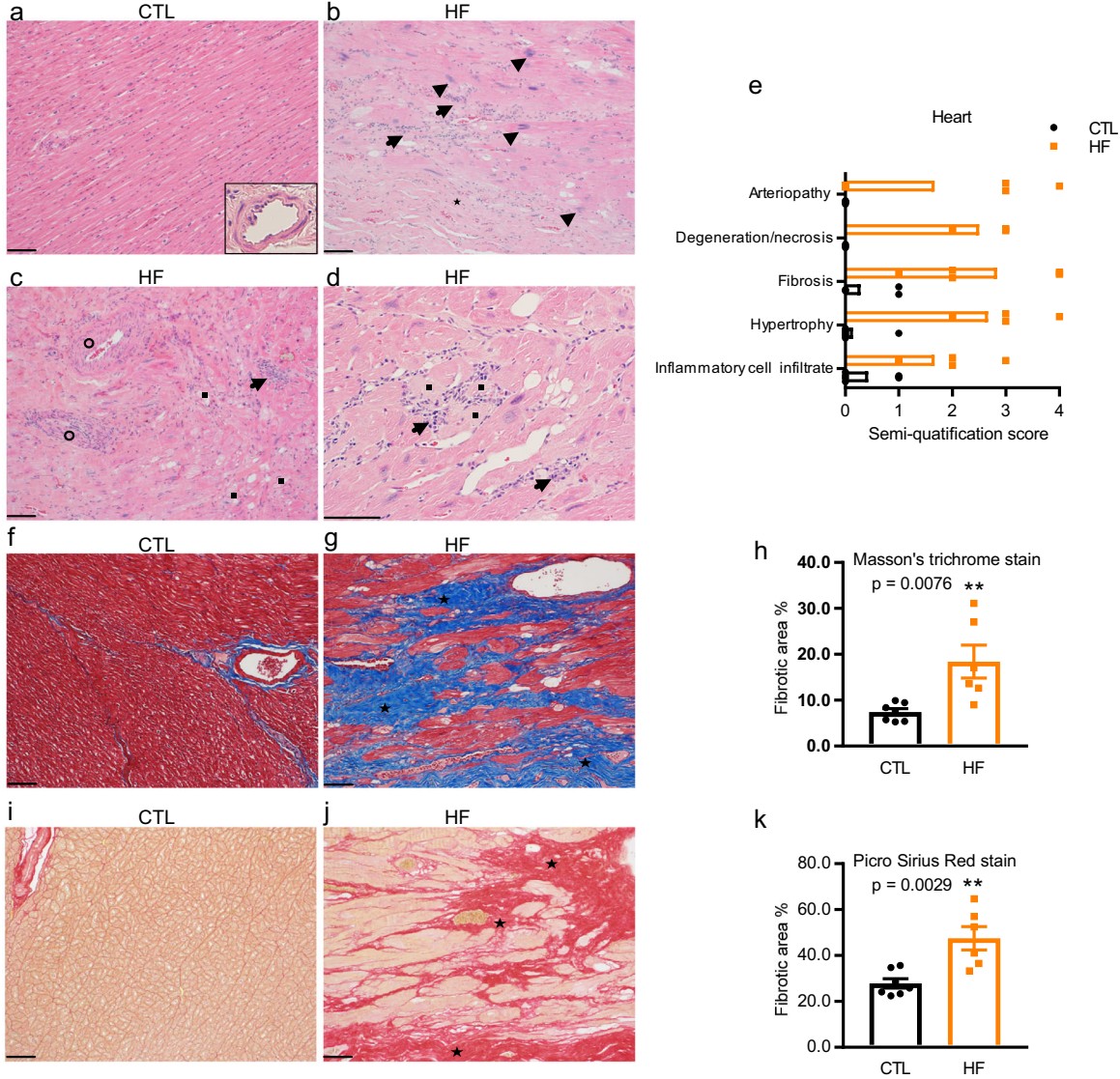

**Fig. 1 Cardiac pathological changes in non-human primates (NHPs) with control (CTL) and heart failure (HF). a–e** Histopathological evaluation of heart tissue from NHPs with CTL ($n = 7$) and HF ($n = 6$). **a** CTL hearts showed no abnormality. **b–d** HF hearts showed myocardial interstitial fibrosis, hypertrophy of cardiomyocytes with atypical karyomegaly, infiltrations of mixed inflammatory cells, and randomly distributed, multifocal degeneration and necrosis of cardiomyocytes. Arteriopathy with thickening of the tunica media of the vascular smooth muscles and narrowing of the lumen was also observed in NHPs with HF. **e** Histopathological grades of CTL and HF were determined using the semi-quantification scoring system (0-not apparent change,1-minimal, 2-moderate, 3-marked, 4-severe). **f–k** Masson's trichrome staining and Picro Sirius red staining of heart tissue from NHPs with CTL ($n = 7$) and HF ($n = 6$). **f, i** No abnormality was observed in the CTL. **g, j** The HF hearts showed diffused blue-stained (**g**, Masson's trichrome stain) and red-stained (**j**, Picro Sirius red stain). **h, k** Quantification of fibrotic area was shown in histogram. Scale bar, 100 μm. Data represent mean ± SEM, **$P < 0.01$ by student's t test.

HF. Molecular oxygen is essential for maintaining normal tissue homeostasis, from metabolic energy production to the regulation of intracellular signal transduction pathways. Under HF conditions, the oxygen tension in cardiomyocytes is negatively regulated by several factors. HF is often associated with cardiomyocyte hypertrophy[43,55–57]. Enlarged cardiomyocytes can reduce the effectiveness of oxygen distribution in the cardiac tissue. As a result of insufficient oxygenation, hypoxia occurs in the heart tissue under pathologically HF-inducing conditions. A recent study showed that mild hypoxia can induce cardiomyocyte hypertrophy via upregulation of the HIF1α-mediated transient receptor potential signaling pathway in neonatal rats[58].

HIF1α is a key transcriptional regulator that acts in response to hypoxia. To investigate whether HIF1α expression correlates with the development of HF, we explored the expression of HIF1α mRNA and protein in NHP hearts. HIF1α mRNA (Fig. 4a) was

significantly elevated in the NHPs with HF. Moreover, immunohistochemical staining showed elevated HIF1α protein levels in the intercalated disks, sarcomeres, intravascular blood cells, and plasma of NHPs with HF (Fig. 4b, c). The elevated HIF1α protein in the myocardium of NHPs with HF was confirmed using western blotting (Fig. 4d, e).

It is well-established that cardiac metabolism is altered in response to pathological cardiac remodeling-associated changes, including hypertrophy and fibrosis. The alteration of cardiac metabolism is characterized by impaired mitochondrial oxidative metabolism[39], increased glycolysis, and decreased glucose oxidation[36]. PFKFB3 is a master regulator of glycolysis that is highly expressed under hypoxic conditions. The *PFKFB3* promoter contains an HIF1α binding site, which recruits HIF1α[59]. To further determine whether the activation of PFKFB3 is associated with HIF1α signaling in HF, we performed immunohistochemical

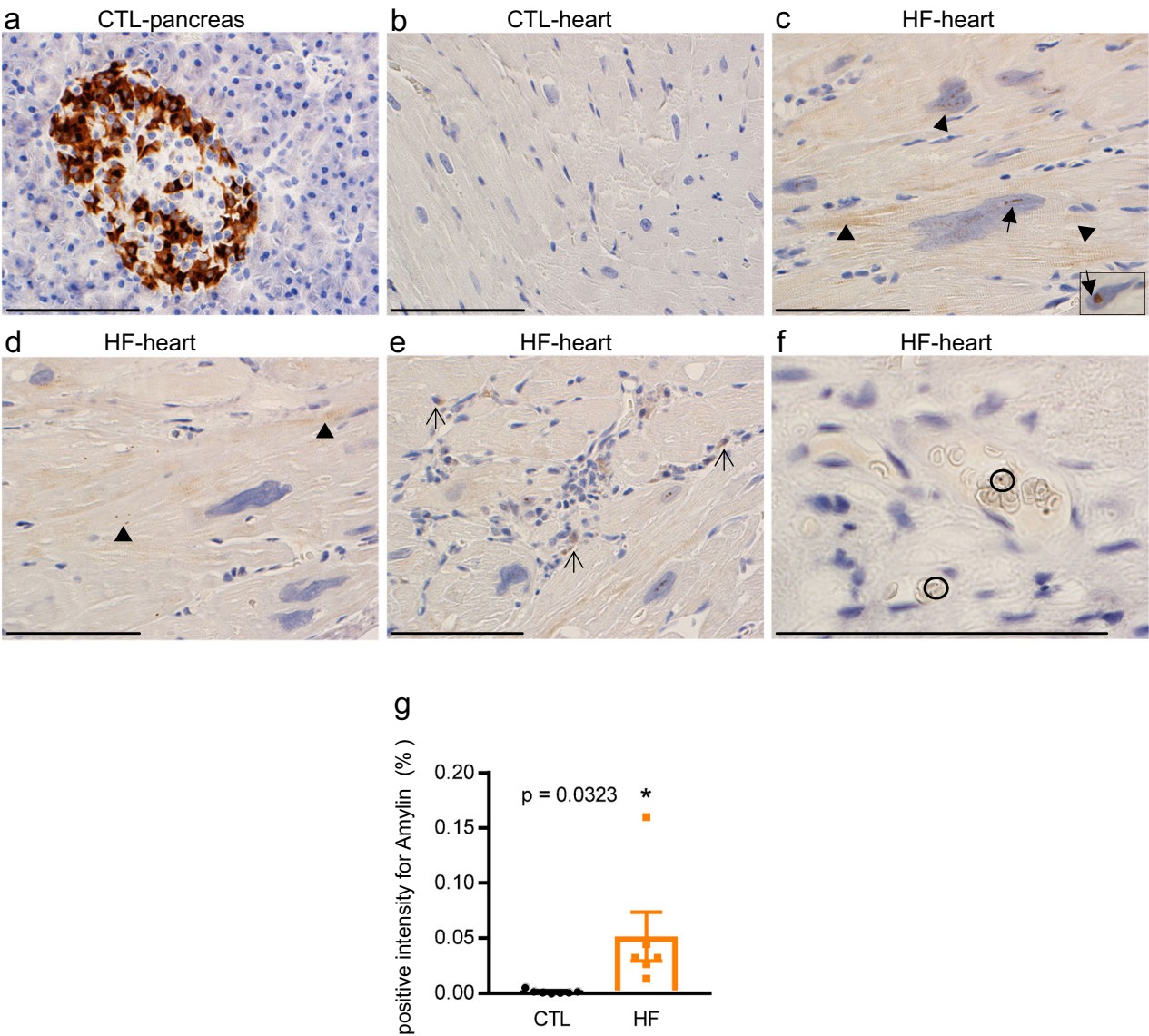

**Fig. 2 Amylin deposition in the heart sections of NHPs with HF. a–f** Amylin staining visualized with DAB chromogen. **a** Positive staining for amylin in pancreas from CTL NHPs. **b** No amylin deposits were observed in the CTL heart ($n = 7$). **c–f** Amylin deposition in the nucleus (**c**), hypertrophic cardiomyocyte sarcolemma (**c** and **d**), inflammatory cells (**e**), and blood cells (**f**) of the HF group ($n = 6$). **g** Bar graph presented the relative positive signal intensity for amylin in a 284 × 214 μm field of view of heart sections from NHPs with CTL ($n = 7$) and HF ($n = 6$). Scale bar, 100 μm. Data represent mean ± SEM, *$P < 0.05$ by student's t test.

staining for PFKFB3 in heart sections from NHPs (Fig. 4f, g) and assessed the mRNA (Fig. 4h) and protein levels (Fig. 4i, j) of PFKFB3, which were also significantly increased in the hearts of NHPs with HF expressing high levels of HIF1α. In addition to increased total expression, phosphorylation of PFKFB3-Ser461 was also significantly enhanced in the HF group compared with CTL NHPs (Fig. 4k, l).

**Cardiac amylin accumulation activates HIF1α and PFKFB3 in hiPSC-CMs and RVCMs.** After establishing that amylin deposition and activation of HIF1α and PFKFB3 occur in the failing hearts of NHPs, we next sought to determine whether cardiac amylin accumulation is involved in the activation of hypoxia (HIF1α) and glycolysis (PFKFB3) in the myocardium. In this study, we utilized an in vitro system hiPSC-CMs to illustrate the role of human amylin in the activation of HIF1α and PFKFB3 signaling pathways. Firstly, the maturation of hiPSC-

CMs was confirmed by assessing the expression of a cardiomyocyte-specific marker, sarcomeric α-actinin (Fig. 5a, green color). High dose of human amylin is known to be cyto-toxic, leading to apoptosis and necrosis. Therefore, we used 6.25 μM (an early cytotoxic concentration, Supplementary Fig. 3) as the concentration for subsequent experiments to evaluate its effect on HIF1α and PFKFB3 signaling pathways.

Incorporation of human amylin into hiPSC-CMs was observed after 2 h of exposure to 6.25 μM human amylin (Fig. 5a, red color, 5b, bar graph). As expected, the immuno-positive signal for HIF1α (Fig. 5c, green color, 5d, bar graph) was stronger in human amylin-stressed hiPSC-CMs than in untreated cardiomyocytes. Western blot analysis of cell lysates from human amylin treated hiPSC-CMs confirmed the increase in HIF1α (Fig. 5e, blots, 5f, bar graph). These results also correlated with the increase in *HIF1α* (Fig. 5g, bar graph) transcripts in cells exposed to 6.25 μM human amylin for 2 h. Collectively, this data suggests that human amylin deposition in cardiomyocytes increased HIF1α expression.

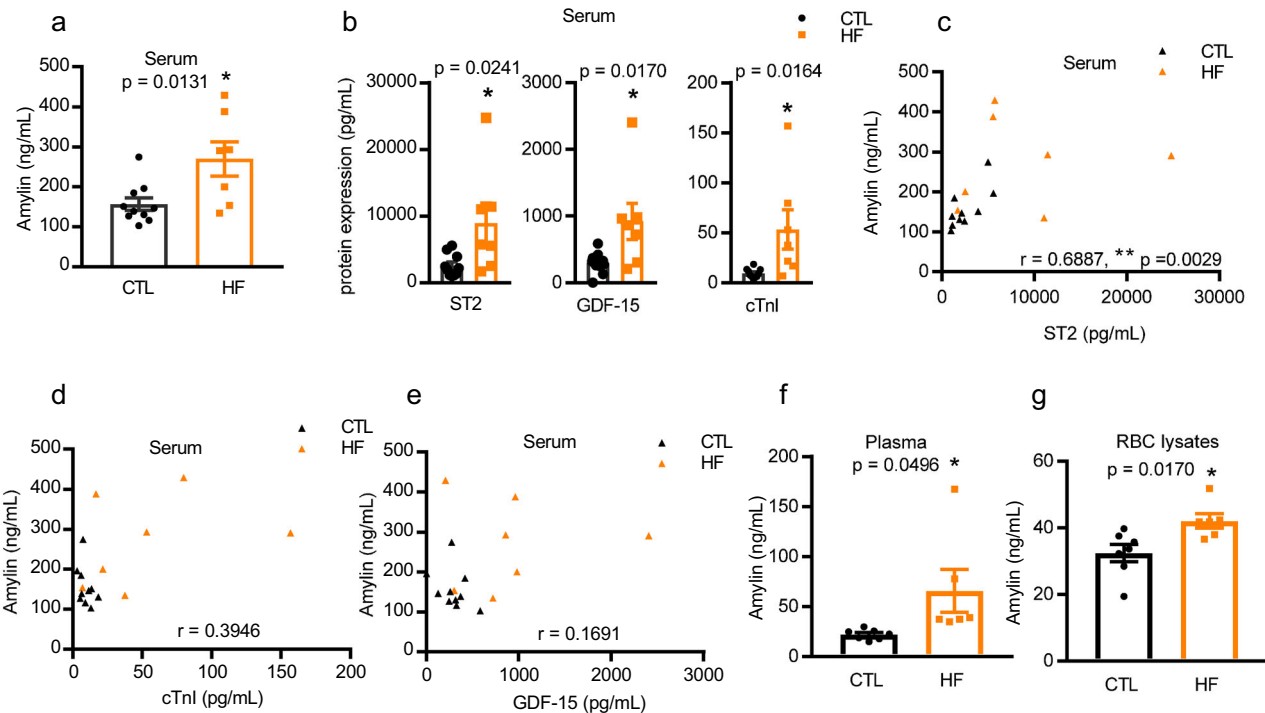

**Fig. 3 Circulating amylin and HF markers levels in NHPs. a** Amylin concentration in serum from a parallel set of NHPs (CTL, $n = 10$; HF, $n = 7$). **b** Protein levels of ST2, GDF-15, and cTnI in serum as in panel (**a**). **c–e** The correlations between serum amylin level and ST2 (**c**), cTnI (**d**), and GDF-15 (**e**), respectively, in serum as in panel (**a**) and (**b**). Amylin levels in matched plasma (**f**) and RBC lysates (**g**) from the same animals whose hearts were used for histopathological evaluation in Fig. 1 and amylin assessment in Fig. 2 (CTL, $n = 7$; HF, $n = 6$). Data represent mean ± SEM. *$P < 0.05$; **$P < 0.01$ by Student's t test. The correlations between serum amylin level and ST2 (**c**), cTnI (**d**), and GDF-15 (**e**), respectively, were analyzed by Spearman nonparametric correlation analysis in GraphPad 7.04 (GraphPad Inc., San Diego, CA), and the values for the Spearman r were indicated on the plots.

After we have demonstrated that the expression of PFKFB3 increased in the hearts of NHPs with HF, we investigated whether PFKFB3 expression increased in hiPSC-CMs exposed to human amylin for 2 h. The protein level of PFKFB3 was significantly upregulated in human amylin-treated cells, as observed using immunofluorescent staining (Fig. 5h, red color, 5i, bar graph) and western blotting (Fig. 5j, blots, 5k, bar graph). Concomitant with this finding, the mRNA level of *PFKFB3* was increased in hiPSC-CMs treated with amylin (Fig. 5l). Moreover, immunofluorescence showed co-localization of amylin with HIF1α and PFKFB3 in hiPSC-CMs incubated with human amylin (Supplementary Fig. 4).

To determine whether human amylin drives the expression of HIF1α and PFKFB3 in cardiomyocytes, we treated hiPSC-CMs with a human amylin antibody for 24 h prior to human amylin exposure. Figure 6a, b shows that compared to the non-treatment group, pre-treatment with the human amylin antibody reduced the upregulation of *HIF1α* and *PFKFB3* mRNA levels induced by human amylin. In addition, we observed decreased HIF1α and PFKFB3 protein levels in hearts from amylin knockout (Amy-KO) mice compared with their wild-type (WT) littermates (Supplementary Fig. 5). These findings indicate that the presence of amylin affected basal cardiac HIF1α and PFKFB3 regulation.

Furthermore, to confirm that the increase in PFKFB3 expression is HIF1α-dependent, we used siRNAs for HIF1α (siHIF1α) to knockdown *HIF1α* in amylin-stressed hiPSC-CMs. As expected, silencing of *HIF1α* reduced *HIF1α* (Fig. 6c) and *PFKFB3* (Fig. 6d) transcript levels induced by amylin. Reduced PFKFB3 expression was also confirmed using PFKFB3 siRNA (siPFKFB3; Fig. 6d). Interestingly, the amylin-induced upregulation of *HIF1α* and *PFKFB3* mRNA levels was significantly suppressed by the HIF1α inhibitor YC-1 (Fig. 6e, f). In addition, we observed slightly deceased amylin deposition in hiPSC-CMs

treated with siRNAs targeting HIF1α (siHIF1α) and PFKFB3 (siPFKFB3), separately, prior to human amylin treatment (Supplementary Fig. 6).

Although hiPSC-CM is a suitable in vitro model for studying the underlying mechanism, the metabolic and functional maturation of these cells is controversial[60–63]. Hence, we used isolated RVCMs, with matured sarcomeric structure and functions, as an additional in vitro model to test our hypothesis. Previous studies have shown that aggregated amylin was incorporated into rat myocardial sarcolemma when RVCMs were incubated with 50 μM human amylin for 2 h[23]. Consistent with the results obtained using hiPSC-CMs, RVCMs acutely exposed to human amylin also showed elevated HIF1α (Fig. 7a–d, green color in a, western blot in c, bar graph in b and d) and PFKFB3 (Fig. 7e–h, red color in e, western blot in g, bar graph in f and h) protein and transcript levels (Fig. 7i, j), respectively.

Taken together, these results support the hypothesis that the HIF1α and PFKFB3 signaling pathway is activated by human amylin, which is also activated in the myocardium of NHPs with HF. A recent study showed that hypoxic activation of HIF1α may occur due to mitochondrial dysfunction in response to increased intracellular $Ca^{2+}$ (ref. [64]). Notably, similar to altered cardiomyocytes, $Ca^{2+}$ cycling was observed in the HIP rats[23] and mitochondrial disarrangement was observed in control RVCMs incubated with exogenous aggregated human amylin[24]. Therefore, we next attempted to investigate the effect of human amylin toxicity on $Ca^{2+}$ and mitochondrial functions.

**Amylin accumulation in cardiomyocytes induced altered $Ca^{2+}$ level, mitochondrial function, oxidative stress, and lactate production in hiPSC-CMs.** Mitochondrial dysfunction may

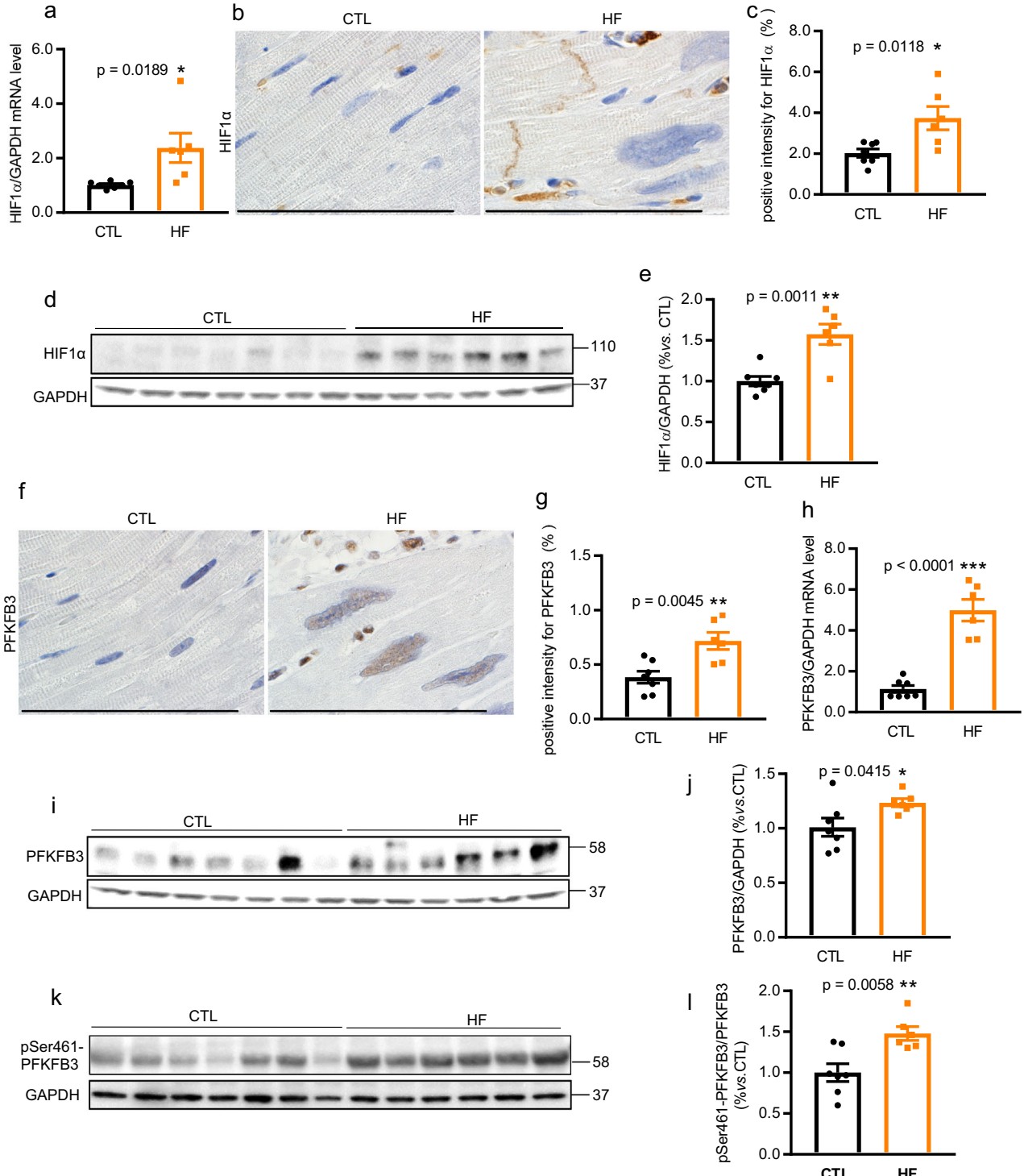

**Fig. 4 Activation of HIF1α, PFKFB3, and PFKFB3 phosphorylation at Ser461 in the myocardium of NHPs with HF.** Expression of HIF1α (**a–e**), PFKFB3 (**f–j**), and pSer461-PFKFB3 (**k–l**) in hearts from CTL ($n = 7$) and HF ($n = 6$) NHPs. qRT-PCR for HIF1α mRNA (**a**) and PFKFB3 (**h**) in hearts from NHPs with HF ($n = 6$) and CTL ($n = 7$). Immunohistochemistry of HIF1α (**b**) and PFKFB3 (**f**) (brown color) in heart sections from NHPs with HF and CTL as used in panel (**a**). Quantifications of the relative positive signal intensity for HIF1α (**c**) and PFKFB3 (**g**) in a 284 × 214 μm field of view of heart sections from NHPs as used in panels (**a**). Western blot analysis with anti-HIF1α (**d**), anti-PFKFB3 (**i**), and anti-pSer461-PFKFB3 (**k**) antibodies on heart lysates as used in panels (**a**). Quantifications of protein expression in western blot were shown in bar graphs (**e**, **j**, and **l**). Scale bar, 100 μm. Data represent mean ± SEM. *$P < 0.05$, **$P < 0.01$, ***$P < 0.001$ by Student's t test.

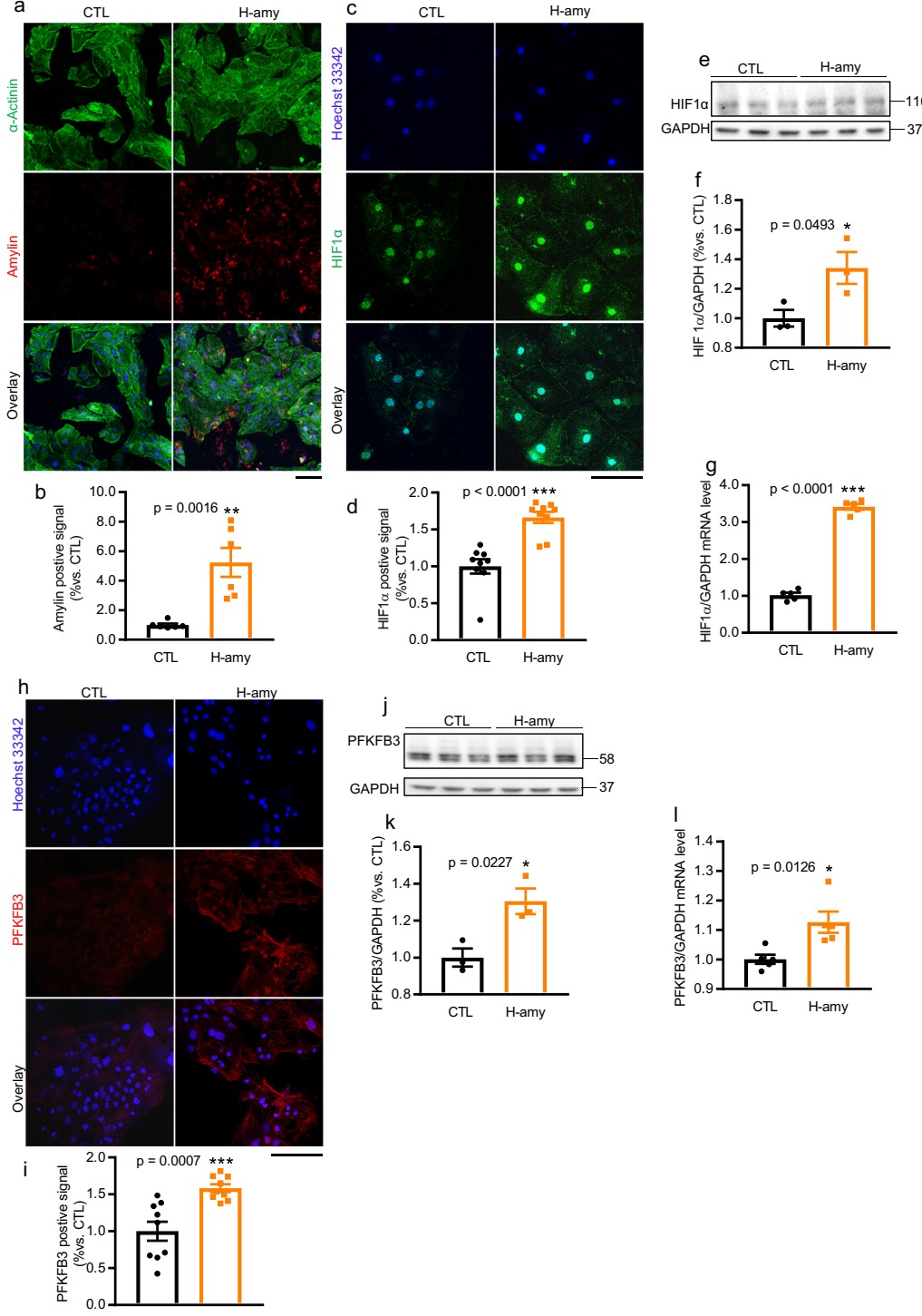

**Fig. 5 Cardiac amylin accumulation activated HIF1α and PFKFB3 in hiPSC-CMs. a** Immunostaining of α-actinin (**a**, green color, mouse anti-α-Actinin antibody), positive signal of human amylin (**a**, red color, rabbit anti-human amylin antibody) and Hoechst 33342 (blue color) in cells treated with H-amy and **b** quantification of amylin positive signal was shown in bar graph (*n* = 6/group). **c–l** Expression of HIF1α (**c–g**) and PFKFB3 (**h–l**) in hiPSC-CMs treated with CTL and H-amy. Positive signal of HIF1α (**c**, green color, mouse anti-HIF1α antibody), PFKFB3 (**h**, red color, rabbit anti-PFKFB3 antibody) and Hoechst 33342 (**c** and **h**, blue color) levels were quantified as shown in bar graph (**d**, HIF1α; **i**, PFKFB3, *n* = 9/group). Western blot analysis of HIF1α (**e**, **f**) and PFKFB3 (**j**, **k**) in hiPSC-CMs incubated under the two conditions described in panel (**c**) (*n* = 3/group). qRT-PCR of HIF1α (**g**) and PFKFB3 (**l**) in myocytes incubated with CTL and H-amy (*n* = 5/group). Scale bar, 100 μm. Data represent mean ± SEM. *$P < 0.05$, **$P < 0.01$, ***$P < 0.01$ by Student's t test.

account for one of the mechanisms underlying cardiac remodeling in heart disease. Hypoxia can reduce electron transport and lead to mitochondrial inner membrane depolarization and subsequent generation of reactive oxygen species (ROS)[57].

Amylin aggregation is a potent generator of oxidative stress[18]. Previous studies have shown that incorporation of aggregated amylin in the sarcolemma induces sarcolemmal $Ca^{2+}$ leakage, leading to increased cytosolic $Ca^{2+}$ and mitochondrial

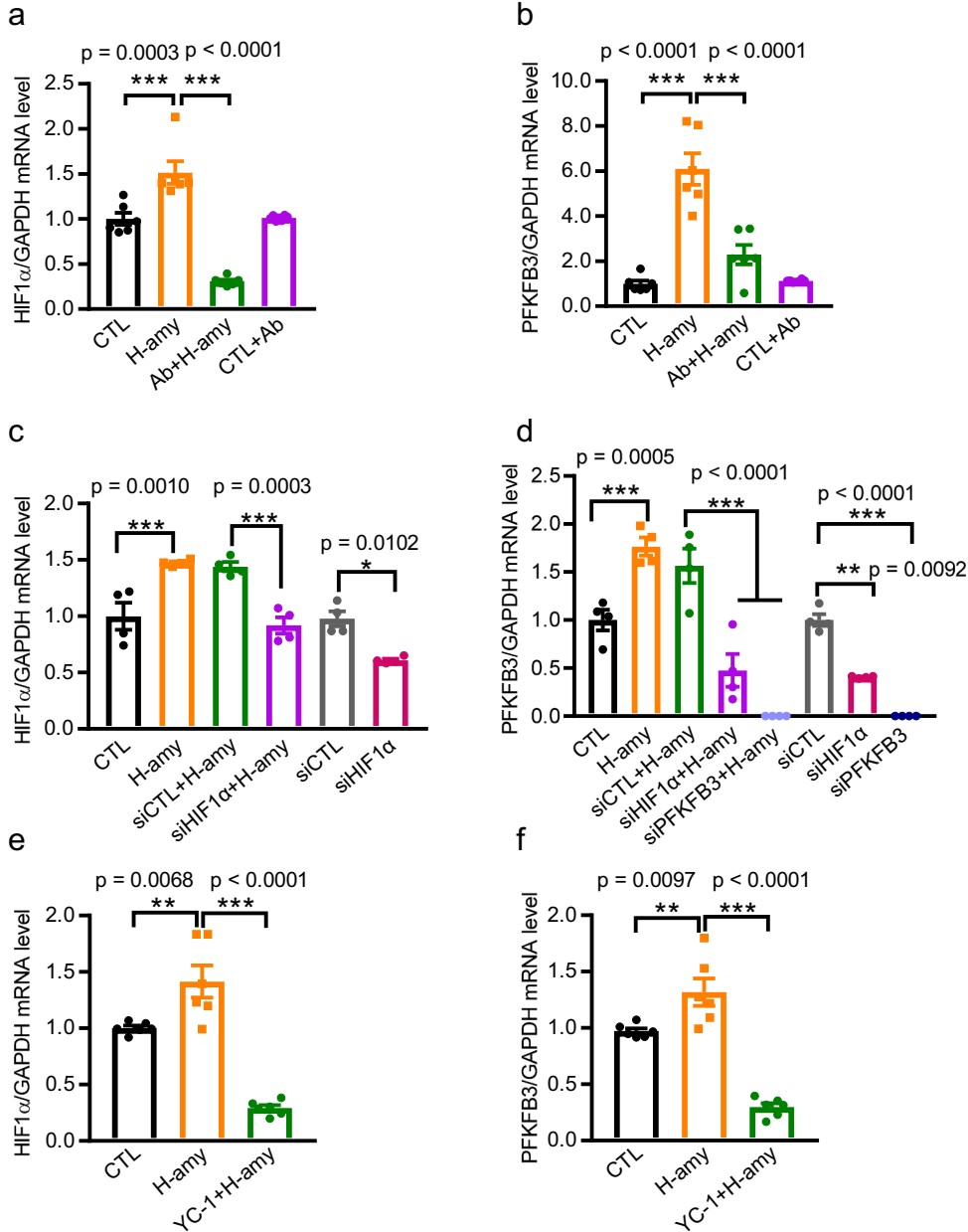

**Fig. 6 Amylin drives the expression of HIF1α and PFKFB3 in hiPSC-CMs. a, b** Effect of pre-treatment of hiPSC-CMs with a human amylin antibody for 24 h on amylin-induced HIF1α (**a**) and PFKFB3 (**b**) mRNA levels ($n = 6$/group). **c** Effect of pre-treatment of hiPSC-CMs with siRNA for HIF1α (siHIF1α) or for vehicle control (siCTL) on amylin-induced HIF1α ($n = 4$/group). **d** Effect of pretreatment of hiPSC-CMs with siHIF1α, siRNA for PFKFB3 (siPFKFB3), or siCTL on amylin-induced PFKFB3 ($n = 4$/group). **e, f** Effect of HIF1α inhibitor YC-1 on the upregulation of HIF1α (**e**) and PFKFB3 (**f**) induced by amylin ($n = 6$/group). Data represent mean ± SEM. *$P < 0.05$, **$P < 0.01$, ***$P < 0.001$ by One-way ANOVA with Tukey's post-test.

dysfunction[23]. Therefore, we examined the steady-state intracellular $Ca^{2+}$ level, ROS production, mitochondrial membrane potential (MMP), and cellular adenosine triphosphate (ATP) levels in hiPSC-CMs incubated with aggregated human amylin. The levels of steady-state intracellular $Ca^{2+}$ (Fig. 8a) and ROS production (Fig. 8b) in hiPSC-CMs increased after incubation with human amylin, which correlated with the results of a previous study showing that incubation of isolated RVCMs with aggregated amylin increases $Ca^{2+}$ transient amplitude[23] and ROS production[24]. In addition, $Ca^{2+}$ transient amplitude and $Ca^{2+}$ handling proteins were measured in hiPSC-CMs exposed to human amylin for 2 h (Supplementary Fig. 7). The results showed that 6.25 μM of human amylin increased $Ca^{2+}$ transient amplitude at spontaneous beating (SB) and paced at 1, 1.5, and 2 Hz,

respectively, although not significantly (Supplementary Fig. 7a–c). Also, illustrative isochronal maps (Supplementary Fig. 7d) showed shortened calcium activation time and CaD90 (the duration of calcium transient at 90% decay) and augmented calcium amplitude in hiPSC-CMs exposed with amylin, indicating $Ca^{2+}$ mishandling in these cells. However, protein expression levels of sarcoplasmic reticulum $Ca^{2+}$-ATPase (SERCA), phospholamban (PLB) (endogenous SERCA inhibitor), and $Na^+/Ca^{2+}$ exchanger (NCX) remained unaltered in amylin-stressed cells (Supplementary Fig. 7e). Consistent with the results of a previous study showing sarcolemmal damage and mitochondrial disarrangement in isolated cardiomyocytes treated with human amylin, we observed a significant loss of MMP (Fig. 8c) and impaired ATP production (Fig. 8d) in amylin-stressed cells. Thus, human

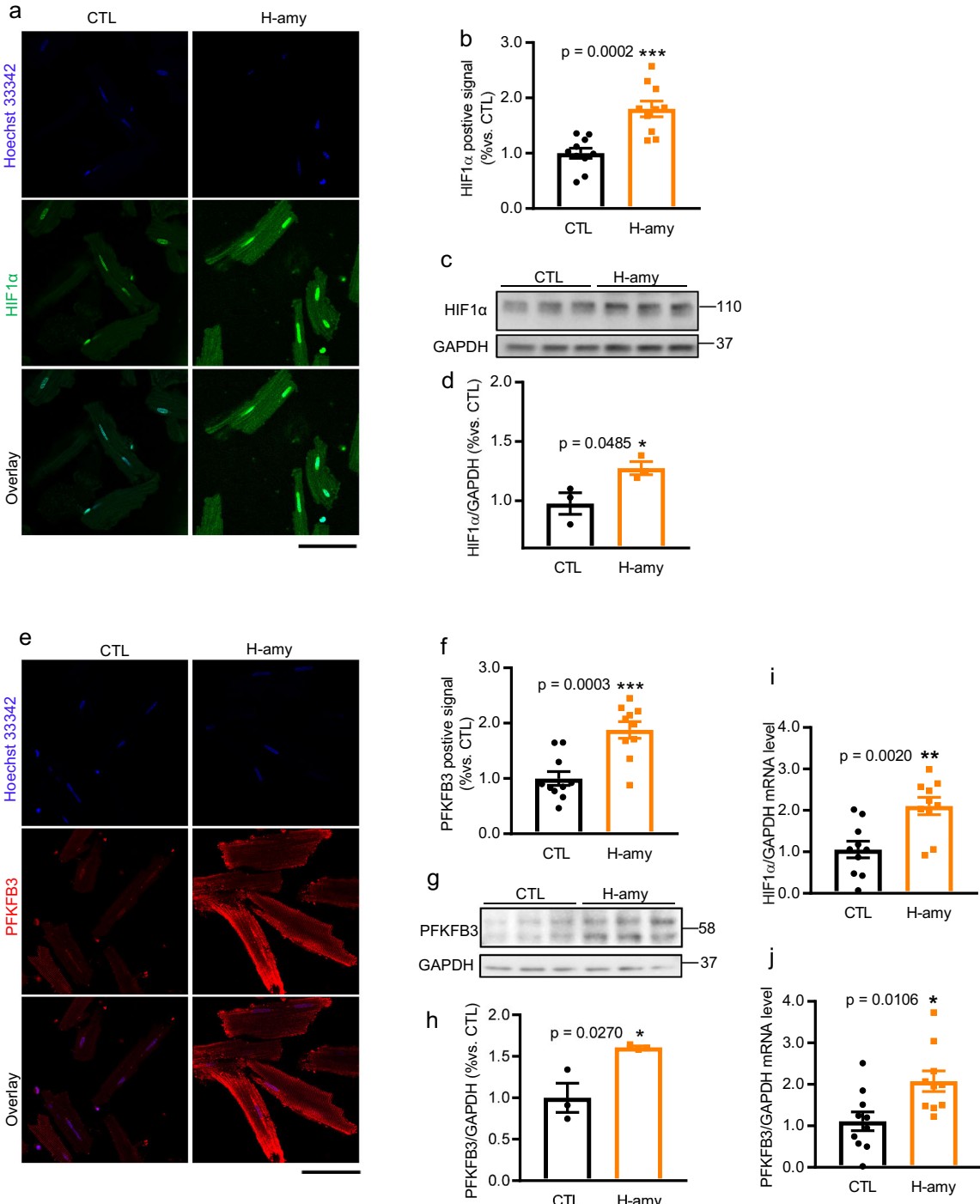

**Fig. 7 Cardiac amylin accumulation activated HIF1α and PFKFB3 in isolated RVCMs. a–j** Increased expression of HIF1α (**a–d**, **i**) and PFKFB3 (**e–h**, **j**) in RVCMs treated with or without H-amy for 2 h. **a**, **b** Immunofluorescence staining (**a**, green color, mouse anti-HIF1α antibody; blue color, Hoechst 33342) in cells treated under the above conditions and **b** quantification of HIF1α positive signal was shown in bar graph (*n* = 10/group). **c**, **d** Western blot analysis (blot, **c** and bar graph, **d**) of HIF1α (mouse anti-HIF1α antibody) in RVCMs treated as in panel (**a**) (*n* = 3/group). **e**, **f** Immunofluorescence staining (**e**, red color, rabbit anti-PFKFB3 antibody; blue color, Hoechst 33342) in cells under the above conditions and **f** quantification of PFKFB3 positive signal was shown in bar graph (*n* = 10/group). **g**, **h** Western blot analysis (blot, **g** and bar graph, **h**) of PFKFB3 (rabbit anti-PFKFB3 antibody) in RVCMs treated as in panel (**a**) (*n* = 3/group). **i**, **j** Increased mRNA levels of HIF1α (**i**) and PFKFB3 (**j**) in RVCMs incubated with amylin (*n* = 10/group). Scale bar, 100 μm. Data represent mean ± SEM. *P < 0.05, **P < 0.01, ***P < 0.001 by Student's t test.

amylin toxicity increased intracellular Ca²⁺ level and ROS production, and impaired MMP, which favors the glycolysis flux with the diversion of pyruvate to lactate[39]. Next, we assessed the levels of the glycolytic end-product, lactate, to evaluate the effect of human amylin on glycolysis. Figure 8e showed that the

production of lactate was significantly increased in cells exposed to human amylin, while the HIF1α inhibitor YC-1, suppressed this increase in human amylin-stressed cells. Based on these observations, we concluded that the incorporation of human amylin induces metabolic changes in intracellular Ca²⁺ level,

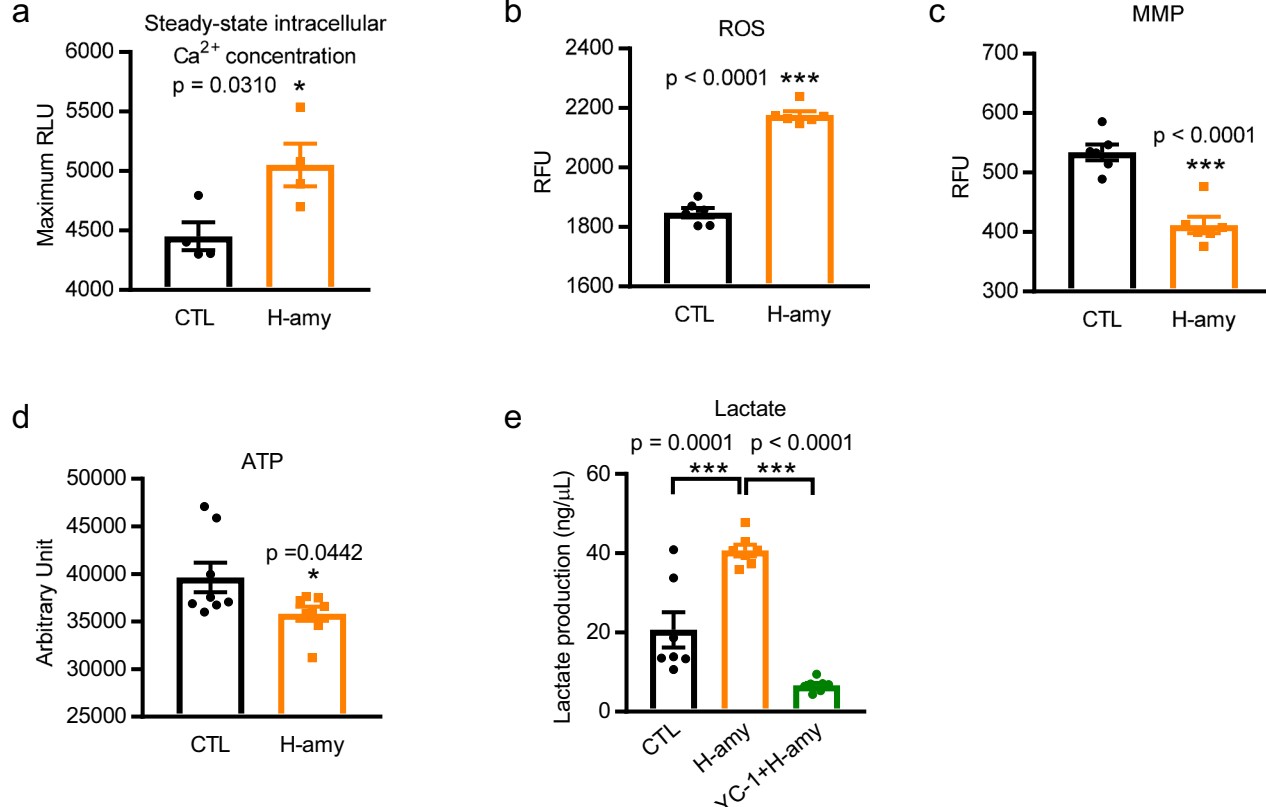

**Fig. 8 Cardiac amylin accumulation induced changes in $Ca^{2+}$ level, oxidative stress, mitochondrial function and lactate production in hiPSC-CMs. a** Steady-state intracellular $Ca^{2+}$ concentration was measured in hiPSC-CMs incubated with or without amylin for 2 h ($n = 4$/group). **b** ROS production was compared in hiPSC-CMs treated as in panel (**a**) ($n = 6$/group). **c** Mitochondrial membrane potential (MMP) ($n = 6$/group) and **d** ATP level ($n = 8$/group) were measured in hiPSC-CMs treated with amylin for 2 h. **e** Effect of pre-treatment with HIF1α inhibitor YC-1 on lactate production induced by amylin ($n = 7$/group). Data represent mean ± SEM. *$P < 0.05$; ***$P < 0.001$ by Student's t test (**a–d**) and One-way ANOVA with Tukey's post-test (**e**).

mitochondrial function, oxidative state, and glycolysis in hiPSC-CMs.

## Discussion

Similar to humans, NHPs spontaneously or following treatment with a high-fat diet, develop cardiovascular diseases related to obesity and T2D, thereby acting as ideal models for investigating the molecular pathogenesis of these diseases[47,65,66]. As shown previously, aggregated amylin plays important detrimental roles in the pancreas, heart, kidney, and brain of humans and HIP rats[13,19,22–24,28–30,51]. In particular, amylin in NHPs possesses an amyloidogenic region, similar to that observed in humans[17]; hence, the role of amylin in NHP hearts warrants further investigation. In this study, we first observed that NHPs with HF have amylin deposition in cardiomyocytes (Fig. 2). Cardiac amylin deposition primarily occurs in the nuclei and sarcolemma of the hypertrophic cardiomyocytes, infiltrated inflammatory cells, and blood cells. In contrast, no amylin deposition or structural abnormalities were observed in the CTL heart sections. These NHPs with HF also showed abnormalities not only in cardiac structures (as shown in Fig. 1), but also pathological signs such as infiltrated immune cells (Supplementary Fig. 1) consistent with the infiltrative cardiovascular diseases of the whole heart[50], characterized by significant reduction in EF and fractional shortening (systolic dysfunction), wider QRS, shortened QT(c) interval, increased LVDP, LVSP, EDV, ESV, and LV mass, and decreased +d$p$/d$t$ max, −d$p$/d$t$ max (Table 1). It is known that high-fat diet negatively affects multiple organs and tissues such as the aorta[67], heart[68], lungs[69], livers[70],

brain[71], kidneys[72], and pancreas[73]. Consistent with the previous findings, we observed lipid accumulation in the aorta, arteriopathy in the aorta, brain, lung as well as other major organs, vacuolation in the liver, hyperplasia in the pancreatic islets, mononuclear cells infiltration, and glomerulopathy in the kidneys in some of the HF NHPs treated with high-fat diet. Analysis of the entire metabolic profiles of these animals revealed a significant increase in BW, BMI, serum cholesterol, and serum triglyceride in the HF group (Table 2), which are potential risk factors for HF in humans. We also observed that NHPs with HF were significantly older than the control NHPs (Table 2 and Supplementary Table 1), a finding consistent with what has been reported that age is a major determinant of the risk for HF and overall cardiovascular disease in humans[74,75]. Sex differences exist in HF and it impacts almost every aspect of HF from epidemiology and risk factors to pathophysiology[76]. Men are predisposed to HF with reduced EF, whereas more women have HF with preserved EF[77]. Our results (Supplementary Table 1) indicated that differences may exist on weight between sexes (although the sample size is small) in NHPs. Indeed, it has been reported that cardiac amylin level was higher in males Amy-KO mice infused with human amylin, compared to females[27], and female HIP rats developed diabetes later in life compared to males[30], indicating a sex-dependent effect in amylin-induced pathology. The previous data[78] showed greater pancreatic amylin deposition in men compared to women, most likely due to increased insulin resistance in men[79,80]. What's more, sex differences have been reported existing in cardiomyocyte ion channels[81–84], intracellular $Ca^{2+}$ handling[27,85,86], contractile

functions[86,87], and cardiac metabolism[88–90], which were typically linked to human amylin induced effects. However, the complicated molecular mechanisms underlying HF are poorly understood, particularly in NHPs. Previous studies have reported that amylin may contribute to fibrosis, which includes disruption of basement membrane structure[30], regulation of extracellular matrix metabolism[91], and alteration of fibroblast activity[92]. Thus, the present results suggest that cardiac amylin accumulation is linked to pathological cardiac hypertrophy, which may accelerate the development of systolic dysfunction in HF.

Previous studies have reported the absence of *amylin* mRNA in the heart[24] and brain[31], while demonstrating that the pancreas is the only organ that generates amylin[23]. Thus, amylin in the heart must originate in the pancreas. Therefore, we assessed the circulation level of amylin in serum, plasma, and RBCs from CTL NHPs and NHPs with HF. Similar to humans, NHPs with HF showed increased circulation of amylin (Fig. 3a, f, g). This is likely due to the high-fat diet leading to amylin release into the blood[93,94]. When over-secreted, amylin forms toxic oligomers, contributing to β-cell dysfunction, as evidenced by impaired glucose tolerance in HF NHPs in this study (Supplementary Fig. 8). Furthermore, the circulating levels of three physiologically distinct biomarkers, GDF15 (inflammation), ST2 (ventricular remodeling and hypertrophy), and cTnI (myocardial necrosis), were significantly high in the sera of NHPs with HF (Fig. 3b). Interestingly, we observed a highly significant positive correlation between serum amylin level and that of the ventricular remodeling and hypertrophy marker, ST2 (Fig. 3c), but not with cTnI and GDF15 (Fig. 3d, e). It is reported that increased ST2 level was associated with diastolic dysfunction[95]. Hence, our present data suggest that amylin oligomer accumulation may accelerate the onset of diastolic dysfunction. This concept is supported by our finding of significantly increased LVDP and EDV (Table 1), characteristics of diastolic function, in NHPs with HF. Although we observed an obvious decrease in the MV E/A ratio (the ratio of mitral peak E-wave velocity and mitral peak A-wave velocity) in NHPs with HF (Table 1), an important parameter for diastolic function, this change does not appear particularly relevant to this study, because they remain within the normal reference range for humans and other animals[96]. Besides, elevated LV mass (Table 1), an indicator of hypertrophy, is also evidenced to be a strong predictor for diastolic dysfunction[97]. This observation is consistent with the results of a previous study showing that the accumulation of cardiac amylin is linked to pathological cardiac hypertrophy and diastolic dysfunction where amylin deposit in the nucleus is capable of inducing hypertrophic transcriptional effects, such as activation of $Ca^{2+}$/calmodulin-dependent protein kinase II (CaMKII)-histone deacetylase (HDAC) and calcineurin-nuclear factor of activated T cells (NFAT) hypertrophic pathways[23]. Collectively, these data suggest that amylin plays an important pathogenic role in the development of HF in NHPs and humans.

Furthermore, the hearts of NHPs with HF positive for amylin deposition showed overexpression of HIF1α (Fig. 4a–e), as suggested by previous studies[41,51]. The results obtained from our study using amylin stressed hiPSC-CMs and isolated RVCMs (Figs. 5–7) showed that amylin formed aggregates in a time-dependent manner (Supplementary Fig. 9). Cytotoxic amylin is readily incorporated into the plasma membrane[16,18,98] and RBC membrane[25,51], leading to cardiomyocyte dysfunction and hypoxia in humans and HIP rats. These enlarged cardiomyocytes exhibit deficiency of oxygen distribution, leading to cardiac hypoxia. Altered cardiac metabolism with increased glycolysis and impaired mitochondrial oxidative state is often observed during cardiac hypertrophy and fibrosis. Our data show increased expression and production of PFKFB3 (a key regulator of glycolysis) in NHPs with HF (Fig. 4f–j) as well as in

cardiomyocytes incubated in vitro with human amylin (Figs. 5–7). Interspecies differences in physiology and genetics would account for some of the differences in results such as PFKFB3 staining in NHPs (Fig. 4f) and hiPSC-CMs (Fig. 5h) shows a nuclear position which is not the case in RVCMs (Fig. 7e). In addition, phosphorylation at Ser461 of PFKFB3 (Fig. 4i, j) was also highly induced, which could further increase kinase activity and enhance glycolysis. Besides, we observed significantly increased levels of protein kinase C (PKC) in HF NHPs. Meanwhile, no protein expression alterations were found for adenosine monophosphate (AMP)-activated protein kinase (AMPK) and protein kinase A (PKA) (Supplementary Fig. 10). Therefore, increased expression of PKC may contribute to the phosphorylation of PFKFB3-Ser461 in HF NHPs, consistent with a previous finding that PKC regulates the PFKFB3 isoenzyme by covalent modification of its C-terminal domain[99].

Myocardial infarction (MI) is one the most common cause of HF, defined as heart muscle necrosis secondary to prolonged lack of oxygen and nutrient supply (ischemia)[100]. HIF1α was reported to increase in ischemic heart tissues[101] and cardiac hypertrophy[44] in humans. Meanwhile, PFKFB3, a target gene of HIF1α, has low basal expression levels but is strongly induced by hypoxia upon myocardial ischemia[102]. Interestingly, significant amylin accumulation was observed in the human ischemic failing heart[23]. In our study, we found elevated levels of serum cTnI (Fig. 3), the preferred biomarker for myocardial injury and MI[100], circulating amylin (Fig. 3), and cardiac HIF1α/PFKFB3 (Fig. 4) in NHPs with HF. Further study is required to establish a causal link between cardiac amylin and HIF1α/PFKFB3 regarding the hypoxia and HF induced by MI. A recent study reported that genetically modified cardiac-specific PFKFB mutants regulate myocardial metabolism and cardiac remodeling[103]. In addition, PFKFB3 regulates not only glucose metabolism but also mitochondrial metabolism, glycerolipid synthesis and the pentose phosphate pathway[45]. PFKFB3 allosterically activates its downstream isoenzyme phosphofructokinase 1 (PFK1), the rate-limiting enzyme of glycolysis[103,104], through its product fructose 2,6 biphosphate (F2,6BP), a potent activator of PFK1. PFK1 functions as a gate-keeper to glycolysis and its activity is tightly controlled by AMP, adenosine diphosphate (ADP), ATP, and citrate[105]. In the failing heart, increased intracellular free AMP and ADP in the cardiomyocytes consequently transduce signaling through AMPK[106], leading to the enhanced synthesis of F2,6BP, an adaptive response to cardiac pressure overload whereas the ATP production is impaired. Thus, the acceleration of glycolytic flux in HF could be partly attributed to an activation of F2,6BP and PFK-1 by both an increase of AMP, an activator of PFK-1, and a decrease of ATP, an inhibitor of the enzyme. Moreover, enhanced PFKFB3 activity affects autophagy, insulin signaling[107], and p38/MAPK signaling[108] in the heart. Of note, in this study, we confirmed that PFKFB3 upregulation was HIF1α-dependent (Fig. 6d). In addition, HIF1α and PFKFB3 participate in mediating human amylin deposition as silencing of either gene effectively suppressed human amylin accumulation (Supplementary Fig. 6). This was in line with previous findings that silencing PFKFB3 suppressed human amylin induced increased glycolytic flux and intracellular $Ca^{2+}$ levels[45]. Hence, modulation of HIF1α and PFKFB3 signaling may help develop novel treatments for HF.

To determine whether human amylin drives the expression of HIF1α and PFKFB3, we measured HIF1α and PFKFB3 levels in hiPSC-CMs treated with a human amylin antibody for 24 h prior to human amylin exposure (Fig. 6a, b) and also in Amy-KO mice (Supplementary Fig. 5). The decreased HIF1α and PFKFB3 expression reported in previous studies indicated that the presence of amylin might affect cardiac HIF1α and PFKFB3 regulation. This was supported by the finding that nuclear HIF1α and PFKFB3 levels were both increased in islets from T2D

patients and HIP rats[45], which could be attributed partly to the nuclear amylin deposition. Indeed, previous investigations showed that compared to WT littermates, Amy-KO mice showed improved glucose intolerance[109], which can be caused by hypoxia[110], and treatment of HIP rats with an amylin lowering compound reversed HIF1α activation in kidneys[51].

Key aspects of cardiomyocyte functions require tight regulation of the subcellular $Ca^{2+}$ level, as sustained abnormal cytosolic $Ca^{2+}$ signaling may contribute to cardiac dysfunction and remodeling. Figure 8a shows an increase in intracellular $Ca^{2+}$ level in hiPSC-CMs exposed to human amylin for 2 h. This agrees with previously published data showing that amylin toxicity is mediated by an increase of intracellular $Ca^{2+}$ level[45]. Elevated $Ca^{2+}$ transient amplitude was also observed in hiPSC-CMs treated with amylin at SB and paced at 1, 1.5, and 2 Hz, respectively (Supplementary Fig. 7a–d). However, no changes were observed in SERCA, PLB, or NCX protein amounts in amylin treated cells (Supplementary Fig. 7e), although these $Ca^{2+}$ handling proteins are known as contributing factors in response to amylin stress. We postulate that amylin oligomers could acutely elevate $Ca^{2+}$ transients, while altered expressions of SERCA, PLB, and NCX may require longer-term effects, as NCX expression was significantly decreased in NHPs with HF (Supplementary Fig. 11). Further experiments are needed to study the direct connection between the activities of $Ca^{2+}$ handling proteins and amylin. Previous findings indicate that binding of amylin to calcitonin gene-related peptide (CGRP) receptors serves as an important mechanism for amylin internalization in pancreatic β-cells and neurons[111,112]. However, this does not appear to apply to cardiomyocytes as another study reported that blocking CGRP receptors did not reduce amylin-mediated sarcolemmal $Ca^{2+}$ leakage in isolated cardiomyocytes[27] suggesting that other mechanisms, e.g., sarcolemmal injury, contribute to amylin-mediated myocyte $Ca^{2+}$ dysregulation[26]. Studies have also reported that amylin receptors, comprised of heterodimers of calcitonin receptors (CTR) and one of three receptor activity–modifying proteins (RAMP 1–3), are widely expressed in the central nervous system of NHPs[113,114]. Additionally, mRNA expression levels of RAMP1, RAMP2, and RAMP3 are elevated in different rat heart dysfunction models[115,116]. Given that RAMPs form a key part of amylin receptors, we infer that these amylin receptors may also contribute to $Ca^{2+}$ signaling. However, further investigations are required to understand the mechanisms underlying the increase in intracellular $Ca^{2+}$ levels.

Mitochondria is the most important intracellular source of ROS, and can also induce cellular oxidative damage, meanwhile, MMP and ATP play important roles in the regulation of ROS production. The incorporation of amylin in the sarcolemma of isolated cardiomyocytes causes mitochondrial disarrangement and oxidative stress[24]. Consistent with previous findings, we observed significantly increased ROS production as well as reduced MMP levels and ATP production in hiPSC-CMs treated with amylin (Fig. 8b–d).

The normal healthy heart maintains metabolic flexibility by utilizing different energy substrates including fatty acids, carbohydrates, ketones, and amino acids at different rates to support its contractile function. Reduced cardiac function in HF is accompanied by altered cardiac energy metabolism primarily due to impaired mitochondrial ATP production and metabolic flexibility. Increased glycolysis and ketone oxidation rates, as well as downregulated glucose oxidation, are common characteristics of the failing heart[117,118]. Recent studies reported that amylin induces the glycolysis and pentose phosphate pathways by reducing glucose oxidation via the TCA cycle and pyruvate anaplerotic reactions in β-cells[45]. Moreover, cardiac amylin accumulation in HF accelerates cardiac hypertrophy and

remodeling[23] partly because amylin aggregates can alter mitochondrial disarrangement and mitochondrial function in cardiomyocytes[24]. The primary metabolic changes induced by the activation of the HIF1α and PFKFB3 pathway under human amylin toxicity stress included the diversion of glycolysis to lactate production rather than pyruvate oxidation, as is observed in the β-cells of patients with T2D and in our present results (Fig. 8e). To understand the extent of the metabolic alteration caused by human amylin, we suppressed accelerated glycolysis by pre-treatment of hiPSC-CMs with HIF1α inhibitor YC-1 (Fig. 8e). Interestingly, YC-1 not only is a HIF1α inhibitor but also intensifies the antioxidant properties of nitric oxide by inducing soluble guanylyl cyclase activation[119], which may play a protective role in amylin induced ROS production. We found the elevated level of glycolytic end product lactate in amylin stressed hiPSC-CMs, consistent with the upregulated lactate dehydrogenase (LDH) gene expression observed in HIP rat pancreatic islets[45]. It has been reported that human amylin evoked LDH release and enhanced LDH activity in rat pancreatic insulinoma beta-cells, human islets cells, and human brain vascular pericytes[33,111,120]. Therefore, we expect a similar effect of human amylin on LDH in hiPSC-CMs and isolated RVCMs. Pyruvate kinase M2 (PKM2), regulating the final rate-limiting step of glycolysis, is known to reduce pyruvate kinase activity and promote the glycolytic pathway in the failing heart and the activation of HIF1α could lead to an induction of PKM2[121]. The upregulated PKM2 in HF is most likely due to the lower enzymatic activity of PKM2 that disfavors oxidative phosphorylation, a maladaptation to hypoxia. PKM2 can then be modified by signaling proteins and posttranslational modifications to adjust its enzymatic activity to favor higher proliferation or energy production as needed[122]. Based on these findings, we postulate that there might be a dysregulation of PKM2 expression and activity in amylin stressed cardiomyocytes as increased PKM2 gene expression was also observed in pancreatic islets from HIP rats[45].

In conclusion, these studies provide a potential link between human amylin aggregation and the alterations in cardiomyocyte functions associated with HF. We further determined that human amylin stress activates the HIF1α and PFKFB3 pathways in the cardiomyocytes of individuals with HF. The consequent metabolic remodeling included mitochondrial dysfunction and increased glycolysis in response to human amylin stress, which could be a potential therapeutic target for HF. Hence, the potential of circulating amylin as a biomarker should be further investigated.

## Methods

**Animals**. Fresh and fixed hearts from adult cynomolgus monkeys (*Macaca fascicularis*), including systolic HF and CTL animals, were obtained from vendors (Wuxi AppTec and Kunming Biomedical International) at the time of necropsy in accordance with relevant regulations. All animals were housed in an Association for Assessment and Accreditation of Laboratory Animal Care (AAALAC)—accredited facility under light-, temperature-, and humidity-controlled conditions, with three meals per day, and water provided *ad libitum*. CTL animals were fed a normal calorie diet (3.81 kcal/g of total energy, 33% of calories from protein, 14% of calories from fat, 53% of calories from carbohydrate). Meanwhile, NHPs with HF were fed the vendor (Kunming Biomedical International)'s proprietary high-fat diet (4.15 kcal/g of total energy, 12% of calories from protein, 32% of calories from fat, 56% of calories from carbohydrate, with 1.20 mg/kcal cholesterol) for 2 years before developing HF. Metabolic profiles and cardiac function ([ECG], echocardiography, and [LV] hemodynamics) were assessed before the animals were sacrificed by vendors. The NHP heart tissues were divided into the systolic HF ($n = 6$, monkeys with [EF] < 40%) and a non-HF ($n = 7$; controls, CTL groups). Monkeys in the HF group were obese except for one female animal and generally had a body mass index (BMI) > 30. Age, gender, BW, and BMI data for the 13 NHPs used in this study were included in Supplementary Table 1. Fresh hearts from Amy-KO mice were obtained from Shanghai Model Organisms, LTD. The Amy-KO mouse was generated via the CRISPR/Cas9 gene editing technique targeting mouse *IAPP* exon 3 in the C57BL/6 J strain, resulting in a deletion of *IAPP* exon 3 coding sequence (gRNA1: 5′-CAGTGTACATAGTCAATGAC-3′; gRNA2:

5′-ATTGTGCATTCTCACTGAGG-3′). Totally 5 Amy-KO mice were assessed in the study (4 males and 1 female) at 2 months of age. Age- and sex-matched WT littermates served as the control group.

**Isolation of RVCMs**. Fresh *Sprague Dawley* rat hearts were provided (2-month-old, male, around 300 g) by Wuxi AppTec in accordance with all the relevant regulations. Immediately after euthanasia, the hearts were collected and placed on a Langendorff perfusion apparatus and perfused with 1 mg/mL collagenase as described previously[23,25]. The LV tissue was cut into small pieces, filtered, and placed in standard Tyrode's solution containing 140 mM NaCl, 4 mM KCl, 1 mM MgCl$_2$, 10 mM glucose, 5 mM HEPES, and 1 mM CaCl$_2$ (pH = 7.4). All experiments were performed at room temperature.

**hiPSC-CM culture**. hiPSC-CMs from Help Stem Cell Innovation were cultured as described previously[123]. Beating hiPSC-CMs were visible after 8 days of differentiation. hiPSC-CMs expressing cardiac troponin T (cTnT), α-actinin, and myosin heavy chain (MHC) were selected for subsequent experiments. Cells were sequentially cultured for 3 weeks to obtain stable beats per minute (BPM) of ~40 to 60 prior to use.

**Treatment of isolated RVCMs with recombinant human amylin**. The amylin oligomerization reaction was prepared in saline at 37 °C with 50 μM recombinant human amylin (AS-60254-1, Anaspec, Fremont, CA). Adult rat cardiomyocytes were then incubated with preformed amylin oligomers (50 μM) for 2 h at room temperature.

**Flow cytometry for apoptosis assessment in hiPSC-CMs**. hiPSC-CMs were treated with preformed amylin oligomers (6.25 μM, 12.5 μM, and 25 μM) at 37 °C for 2 h. After washing, cells were stained with annexin V and propidium iodide (PI) for 15 min at room temperature and analyzed on a BD LSRFortessa cell analyzer. The data were analyzed with the BD FACSDiva software.

**Treatment of hiPSC-CMs with recombinant human amylin**. hiPSC-CMs were incubated with preformed amylin oligomers (6.25 μM) for 2 h at 37 °C.

**Immunofluorescence**. Isolated RVCMs and hiPSC-CMs washed with phosphate-buffered saline (PBS) were fixed and permeabilized as described previously[23]. The primary antibodies used were mouse anti-α-actinin (sarcomeric) antibody (1:800, AF7811, Sigma-Aldrich, St. Louis, MO), mouse anti-HIF1α antibody (1:200, ab16066, Abcam, Cambridge, UK), goat anti-HIF1α antibody (1:20, AF1935, R&D Systems, Minneapolis, MN), rabbit anti-PFKFB3 antibody (1:100, ab181861, Abcam, Cambridge, UK), rabbit anti-amylin antibody (1:400, T4157, Peninsula Laboratories, San Carlos, CA), and mouse anti-amylin antibody (1:50, sc377530, Santa Cruz, Dallas, TX). The secondary antibodies included goat anti-mouse IgG labeled with Alexa Fluor 488 (1:500, ab150113, Abcam, Cambridge, UK), donkey anti-rabbit IgG labeled with Alexa Fluor 647 (1:500, ab150075, Abcam, Cambridge, UK), donkey anti-goat IgG labeled with Alex Fluor 594 (1:250, A-11058, Thermo Fisher, Waltham, MA), and donkey anti-rabbit IgG labeled with Alexa Fluor 594 (1:400, a32754, Thermo Fisher, Waltham, MA). Nuclei were stained with Hoechst 33342 (1:10000, H3570, Thermo Fisher, Waltham, MA) for 10 min at room temperature. Cells were analyzed under a confocal microscope (Leica TCS SP8, Germany).

**Enzyme-linked immunosorbent assay and Luminex assay**. ELISA for human amylin (EIA-AMY-5, Raybiotech, GA) detection was performed according to the manufacturer's protocols. Luminex assays for detecting ST2, GDF-15, and cTnI (R&D Systems, Minneapolis, MN) were performed according to the manufacturer's protocols.

**Immunoblotting**. Western blot analysis was performed on heart homogenates from NHPs and Amy-KO mice, and cell lysates from isolated RVCMs and hiPSC-CMs. Primary antibodies were mouse anti-HIF1α (1:1000, ab16066, Abcam, Cambridge, UK), rabbit anti-PFKFB3 (1:1000, ab181861, Abcam, Cambridge, UK), rabbit anti-phospho461-PFKFB3 antibody (1:1000, ab202291, Abcam, Cambridge, UK), rabbit anti-AMPKα antibody (1:1000, 2532S, Cell Signaling Technology, Danvers, MA), rabbit anti-PKCα antibody (1:1000, 2056S, Cell Signaling Technology, Danvers, MA), rabbit anti-PKA C-α antibody (1:1000, 4782S, Cell Signaling Technology, Danvers, MA), mouse anti-SERCA (1:1000, MA3-919, Thermo Fisher, Waltham, MA), mouse anti-NCX (1:1000, MA3-926, Thermo Fisher, Waltham, MA), rabbit anti-phospholamban (1:1000, PA5-82945, Thermo Fisher, Waltham, MA), and mouse anti-GAPDH (1:10000, MA515738, Thermo Fisher, Waltham, MA) antibodies as described previously[23].

**Immunohistochemistry (IHC)**. Paraffin-embedded heart tissue sections (4 μm) were incubated with anti-amylin (1 μg/mL, sc377530, Santa Cruz, Dallas, TX), anti-HIF1α (4 μg/mL, ab16066, Abcam, Cambridge, UK), anti-PFKFB3 (10 μg/mL, ab181861, Abcam, Cambridge, UK), anti-CD3 (0.3 μg/mL, CME324C, Biocare

Medical, Pacheco, CA), anti-CD8 (4 μg/mL, ab178089, Abcam, Cambridge, UK), anti-CD45 (1 μg/mL, M0701, Dako, Santa Clara, CA), and anti-CD68 (0.1 μg/mL, CM033B, Biocare Medical, Pacheco, CA), as well as related isotype controls. After washing, the tissue sections were subsequently incubated with horseradish peroxidase (HRP)-labeled goat anti-rabbit or anti-mouse IgG (Dako EnVision + System) for 30 min and visualized with the DAB + substrate chromogen system (Dako, Santa Clara, CA) for the identification of immunoreactive sites. The sections were then counterstained with Gill's hematoxylin (Sigma-Aldrich, St. Louis, MO), dehydrated with ethanol and xylene, and coverslipped for analysis.

**Histopathological evaluation**. Tissue sections were stained with hematoxylin & eosin (H&E) (hematoxylin, GHS132-1L; eosin Y solution, HT110132-1L; Sigma-Aldrich, St. Louis, MO) for routine light microscopy evaluation. The Masson's trichrome stain kit (ab150686, Abcam, Cambridge, UK) to detect cardiac fibrosis was used according to the manufacturer's instructions. Picro Sirius red staining (Direct red 80, 65548, Sigma-Aldrich, St. Louis, MO) and Picric acid (10015714, Xilong, China) was also applied to evaluate the fibrotic area, appearing as red color. Histopathological scores were graded as 0 "-" (no apparent change), 1 (minimal), 2 (moderate), 3 (marked), and 4 (severe) based on the increasing extent and/or complexity of morphological changes[124,125]. More details for grading standards are described in Supplementary Table 2.

The stained tissue sections were captured under an Olympus BX53 digital microscope with the cellSens digital imaging software (Olympus, Tokyo, Japan), scanned on a Leica Aperio AT2 scanner (200x magnification; Leica Biosystems), and analyzed with the Halo software v3.0.311.217 (Indica Labs) with Area Quantification (v2.1.3.0; Indica Labs) and CytoNuclear (v2.0.5.0; Indica Labs) algorithm.

**Quantitative reverse transcription-polymerase chain reaction (qRT-PCR)**. The expression levels of HIF1α and PFKFB3 in NHP hearts, rat cardiomyocytes, and hiPSC-CMs were assessed using the related qRT-PCR kit. The specific primers are listed in Supplementary Table 3.

**Measurement of mitochondrial membrane potential (MMP)**. hiPSC-CMs were incubated with human amylin as described above. After washing, the cells were further incubated with 25 nM MitoTracker® (MitoTracker™ Red CMXRos, Thermo Fisher, Waltham, MA) according to the manufacturer's instructions and analyzed on a microplate reader (SpectraMax M5, Molecular Devices, USA).

**Detection of cellular ATP (adenosine triphosphate) levels**. hiPSC-CMs were incubated with human amylin as described above. After washing, the cells were lysed, and the supernatant was collected for assessing ATP level according to the manufacturer's instructions (S0027, Beyotime, China). Luminescence was detected by a multifunctional microplate reader (SpectraMax M5, Molecular Devices, USA) and the results were shown in an arbitrary unit.

**Detection of reactive oxygen species (ROS)**. hiPSC-CMs were incubated with human amylin as described above and treated with 10 μM 2′,7′-dichlorofluorescein diacetate (H2DCF-DA, Thermo Fisher, Waltham, MA) for 30 min at 37 °C. ROS production was measured per the manufacturer's instructions.

**Measurement of intracellular Ca$^{2+}$ flux using fluorescence imaging plate reader (FLIPR)**. hiPSC-CMs were cultured in a 384-well plate and treated with human amylin as described above. Five percent Calcium 6 FLIPR dye (FLIPR Calcium 6 Assay Kit, Molecular Devices, San Jose, CA) was loaded in each well and incubated for 2 h at 37 °C. Intracellular Ca$^{2+}$ flux was measured on a FlexStation 3 reader (FLIPR$^{Tetra}$, Molecular Devices).

**Measurement of Ca$^{2+}$ transients**. hiPSC-CMs were cultured in a 24-well plate and treated with human amylin as described previously. Cells were loaded with 1 mg/mL Rhod-2 AM (R1244, Thermo Fisher, Waltham, MA) at 37 °C for 30 min. Rhod-2 was excited at 530 ± 25 nm and fluorescence was read at 590 nm. Data collected with the Rhod-2 by optical mapping system OMS-PCIE-2002 (Mapping Lab, UK) were expressed as $\Delta F = F - F_0$, where $F_0$ is the fluorescence signal in resting myocytes. Ca$^{2+}$ transients were recorded at spontaneously beating and stimulation with external electrodes at 1, 1.5, and 2 Hz, respectively. Cells were perfused with perfusion solution containing 128.2 mM NaCl, 4.7 mM KCl, 1.05 mM MgCl$_2$, 11.1 mM glucose, 1.19 mM NaH$_2$PO$_4$, 20 mM NaHCO$_3$, and 1.3 mM CaCl$_2$ (pH = 7.4, equilibrated with 5% CO$_2$ and 95% O$_2$ before use).

**Lactate measurements**. Media from cultured hiPSC-CMs were sampled from each experiment, and the lactate levels were analyzed by a colorimetric test (MAK064-1KT, Sigma-Aldrich, St. Louis, MO) according to the manufacturer's instructions.

**Antibody treatment**. hiPSC-CMs were pre-treated with 62.5 μM anti-amylin antibody (T4157, Peninsula Laboratories, San Carlos, CA) for 2 h and acutely exposed to 6.25 μM amylin for 2 h at 37 °C as described above.

**siRNA treatment**. siRNAs against *HIF1α* (siRNA ID: s6539, 4390824) and *PFKFB3* (siRNA ID: s10359, 4390824), respectively, and a negative siRNA control (4390843) were purchased from Thermo Fisher (Waltham, MA). hiPSC-CMs were pre-treated with the above siRNAs for 72 h and incubated with human amylin as described previously.

**Drug**. YC-1 (Y102, Sigma-Aldrich, St. Louis, MO) (10 μM) was used to block HIF1α activity. hiPSC-CMs were pre-treated with YC-1 for 24 h at 37 °C and exposed to human amylin as described above.

**Statistics and reproducibility**. All the data were presented as mean ± SEM. Data analysis was done with GraphPad Prism 7.04. Statistical analyses were assessed with student's t test or One-way ANOVA with Tukey's post-test or Two-way ANOVA. $p \leq 0.05$ was considered as statistically significant. The Spearman nonparametric correlation analysis for Fig. 3c–e was performed in GraphPad (GraphPad Inc., San Diego, CA), and the values for the Spearman $r$ were indicated on the plots.

**Study approval**. All the experiments, which were performed in accordance with relevant guidelines and regulations, were approved by Amgen Institutional Animal Care and Use Committee.

**Reporting summary**. Further information on research design is available in the Nature Research Reporting Summary linked to this article.

## Data availability statement

All source data underlying the graphs and charts presented in the main and supplementary figures are presented in Supplementary Data 1. Full, uncropped blots are shown in Supplementary Fig. 12. Additional data and information about this study are available from the corresponding author upon request.

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

## Acknowledgements
This study was funded by Amgen Inc.

## Author contributions
M.L. and L.H. designed the study. M.L. wrote the manuscript. M.L., N.L., C.Q., Y.G. and L.W. performed the experiments and statistical analysis. M.L., N.L., C.Q., Y.G., L.W. and L.H. revised the manuscript. M.L., N.L. and C.Q. contribute equally. All authors read and approved the final manuscript.

## Competing interests
The authors declare no competing interests.
