## [Peer Review File · Communications Biology]

Reviewers' comments:

Reviewer #1 (Remarks to the Author):

The authors goal was to address the question what is drives the accumulation of Amylin during heart failure in non-human primates and how this amylin may regulate cardiac metabolism and function. The authors showed that non-human primates (NHP) with heart failure (HF) had increase cardiac and circulating Amylin levels in the setting of changes consistent with systolic HF which contributed to increases in the expression of the metabolic proteins, HIF1 α and PFKFB3 and which may contribute to changes in Ca²⁺ levels and parameters of mitochondrial function. Although the findings are of particular interest, a number of concerns exist, which reduce the overall enthusiasm for the manuscript in its current form, which would need to be addressed.

Major comments

[1] The NHP used in this study are not appropriately age-matched as the NHP with HF are significantly older than the control NHPs. What is the reason for this difference? It should be addressed. The authors have shown that the NHP with HF had significantly higher body weights and BMI -could this be due in part to increase in age especially since severe systolic heart failure often associated with a decreased in body weight. What is contributing cause to the HF observed in the NHP in this study?

[2] It is unclear from the manuscript text as to what sex the NHP used in this study were. This data needs to be reported as there are documented differences in HF with respect to sex and it has been shown by Liu et al that cardiac Amylin accumulation is greater in males. In addition, the age, sex, and weights of the Sprague-Dawley rats used for RVCM studies need to be described.

[3] The severity scoring system shown in Figure 1 looks to be very subjective as it seems it would be hard to make distinctions between the middle scores (1-3). In addition, the criteria for judging each score is not discussed in detail, so it is difficult to interpret these data. It would be good to include further detail of the scoring system and to provide additional confirmatory assessment methods. Additional assessment could be in the form of staining with picosirius red, PTAH, CD68, WGA, Troponin I and desmin.

In order to fully confirm infiltrative cells in the heart tissue, it would be best to complement the H&E staining with staining for immune cell populations such as macrophages and neutrophils. This is important since macrophages have been reported to have similar changes in PFKFB3 and metabolism changes when activated (similar to Amylin) and also important as it has been shown that Amylin contributes to systemic inflammation. Furthermore, is anything known as to whether Amylin contributes to fibrosis, disrupts the ECM/ alters fibroblast activity?

[4] Are there any gravimetric data available regarding the heart size, lung edema, spleen size, liver size etc. upon extraction? If these data are available it would be good to include. Is it known if Amylin accumulates in other organs at baseline and during HF? Could Amylin be driving some organ-organ communication? What is the signal driving Amylin release from the pancreas? These should be considered and discussed.

[5] Although changes in PFKFB3 expression are shown to be increased it would be helpful to examine changes in its phosphorylation at Ser461 and changes in its activity as even though its expression has changed it may not correlate with activity. This is important since in Figure 4 PFKFB3 mRNA expression appears to be significantly increased in NHP with HF; however, this does not seem to be the case for the protein levels. It would be beneficial to explain why this may be the case especially as PFKFB3 is known to undergo posttranslational modification.

If you decreased Amylin levels during HF in the NHP, does this reverse the HF pathology and also reverse the changes in PFKFB3 and HIF1 α expression? This could be tested in Amylin KO mice +/- HF (Liu et al) Does Amylin deposition still occur in conditions of reduced expression PFKFB3 and HIF1 α ? In addition, since AMPK also phosphorylates PFKFB3 at Ser 461 in addition to PKA and PKC it may beneficial to look at these in addition to downstream targets of PFKFB3 and glycolytic flux. In the same respect, it would be helpful to examine changes in mitochondrial function/metabolism, and oxidative stress. In the same respect, it would be beneficial to look at downstream targets of HIF1 α (e.g. VEGF) and potential changes in posttranslational modifications. Furthermore, the current data would be strengthened with the inclusion of additional co-staining of Amylin with both PFKFB3 and HIF1 α .

[6] Since Amylin is known to position itself in the sarcolemma- is there any information of which proteins it may associate with there and any additional effects it may mediate especially as it can induce sarcolemmal Ca²⁺ leak. Only cytosolic Ca²⁺ levels were measured in the hiPSC-CMs, comprehensive Ca²⁺ transient analysis in addition to changes in levels of Ca²⁺ handling proteins would need to be performed.

[7] Additional discussion is needed to address the metabolic changes that are known to occur in the heart during baseline and HF and the impact of Amylin. Do HIF1 α and PFKFB3 expression change in heart during HF in humans- this should be discussed. It has been shown by Gibb et al that if you increase or decrease cardiac PFKFB3 expression remodeling occurs in addition to changes in cardiac metabolism. In addition, If PFKFB3 activity is enhanced this would also have implications for autophagic, insulin signaling, and p38/MAPK signaling in the heart. PFKFB3 not only regulates glucose metabolism but also regulates mitochondrial metabolism, glycerolipid synthesis and the pentose phosphate pathway. It may be beneficial to interrogate and discuss further the latter given the fact that Montemurro et al have shown that enhanced PFKFB3 increased pyruvate to lactate and stimulated the pentose phosphate pathway in the beta cell Amylin (IAPP) model.

Minor comments

[1] Several grammatical and spelling errors exist in the manuscript. For example, in figure legends 1 and 2 scale bar is misspelt. Please address.

[2] Some of the immunoblots are over cropped and hard to see (e.g. Fig 5 and 7) and it would be helpful to see more of the blot to fully appreciate the data.

[3] The authors discussed using commercial sera from control NHPs and those with HF- were these sera obtained from the NHPs from which the other data were obtained? Or are these a parallel set of NHPs? Please clarify.

[4] The HF image for Figure 4e appears to be cross-sectional. It would be better to include an image from the same orientation as the control image

[5] Please include quantification for the staining in Figures 2, 5, and 7

[6] The HIF1 α inhibitor, YC-1 used in the cell studies not only inhibits HIF1 α but is an activator of soluble guanylyl cyclase. The potential implications of this should be discussed.

[7] The NHP with HF have wider QRS interval consistent with systolic heart failure; however, it is surprising this is the only EKG change observed. This should be discussed

[8] What is the expression of Amylin receptor in NHP? Does this change between control and HF NHP? This should be discussed.

Reviewer #2 (Remarks to the Author):

The report by Liu and colleagues investigated a mechanism by which amylin aggregates in cardiomyocytes (CMs) could induce heart failure (HF). For their studies, the authors analyzed tissue from non-human primates with or without HF, and used two in vitro methods to determine if amylin treatment upregulates HIF1 α pathways in CMs. In general the manuscript was well-written and easy to follow. While this connection among HF, amylin aggregates and HIF1 α appears novel, I have several questions/comments on the methods and analyses used, clarification of which will make the findings more compelling.

Major comments:

1. Please provide evidence of specificity for all antibodies used. How were the antibodies validated? Were no-primary antibody controls used? Higher mag photos would allow the reader to better visualize the positive staining.

2. I do not find the details on the human amylin used for incubation studies. There is reference to

the procedure ("as previously described"), but I don't find the initial description. How do you know the effects you see (e.g. on HIF1a and PFKFB3) is caused by aggregated amylin rather than the unaggregated form? What is the time course of human aggregation in culture? How can you verify that aggregates are indeed present in your experiments? What is known about the presence of amylin receptors in the hiPSC-CMs?

3. Provide more information on the NHPs. Diet, housing conditions, sex? Was there something specific that predisposed these specific NHPs to HF? While it doesn't appear statistically significant, is the difference in age between the control and HF group relevant?

Minor comments:

1. Briefly introduce the pluripotent stem cell-CMs and isolated rat ventricular CMs as a model in introduction.

2. For the presentation of results, always have CTRL data in the first position (discussed first, pictured in the first/top/left position.) Notably in Fig. 1.

3. Fig. 1: Show higher magnification photos of critical pathologies as insets.

4. Fig. 2: Was amylin in the HF islet different than control?
As above, how were these antibodies validated?

5. Fig. 3: Why are the group numbers different from the Tables?
Would be helpful if the points in the scatter plots were labeled CTRL or HF.

6. Fig. 4: Why are there so many data points in a and d?

7. Fig. 6: Label each individual graph a through f.
As in Figure 4, there are inconsistencies in sample size ($n = 3$) and what is depicted graphically. Please make clearer how the individual data points were derived.

8. Fig. 7: Please label individual graphs/panels for ease of reading. As in Fig. 6, there's inconsistency in sample size ($n = 3$) and what is depicted graphically. Please make clearer how the individual data points were derived.

9. Define MMP at first use.

10. Fig. 8: Define group sizes.

Reviewers' comments:

Reviewer #1 (Remarks to the Author):

The authors goal was to address the question what is drives the accumulation of Amylin during heart failure in non-human primates and how this amylin may regulate cardiac metabolism and function. The authors showed that non-human primates (NHP) with heart failure ¹ had increase cardiac and circulating Amylin levels in the setting of changes consistent with systolic HF which contributed to increases in the expression of the metabolic proteins, HIF1 α and PFKFB3 and which may contribute to changes in Ca²⁺ levels and parameters of mitochondrial function. Although the findings are of particular interest, a number of concerns exist, which reduce the overall enthusiasm for the manuscript in its current form, which would need to be addressed.

Major comments

[1] The NHP used in this study are not appropriately age-matched as the NHP with HF are significantly older than the control NHPs. What is the reason for this difference? It should be addressed. The authors have shown that the NHP with HF had significantly higher body weights and BMI -could this be due in part to increase in age especially since severe systolic heart failure often associated with a decreased in body weight. What is contributing cause to the HF observed in the NHP in this study?

[2] It is unclear from the manuscript text as to what sex the NHP used in this study were. This data needs to be reported as there are documented differences in HF with respect to sex and it has been shown by Liu et al that cardiac Amylin accumulation is greater in males. In addition, the age, sex, and weights of the Sprague-Dawley rats used for RVCM studies need to be described.

[3] The severity scoring system shown in Figure 1 looks to be very subjective as it seems it would be hard to make distinctions between the middle scores (1-3). In addition, the criteria for judging each score is not discussed in detail, so it is difficult to interpret these data. It would be good to include further detail of the scoring system and to provide additional confirmatory assessment methods. Additional assessment could be in the form of staining with picrosirius red, PTAH, CD68, WGA, Troponin I and desmin.

In order to fully confirm infiltrative cells in the heart tissue, it would be best to complement the H&E staining with staining for immune cell populations such as macrophages and neutrophils. This is important since macrophages have been reported to have similar changes in PFKFB3 and metabolism changes when activated (similar to Amylin) and also important as it has been shown that Amylin contributes to systemic inflammation. Furthermore, is anything known as to whether Amylin contributes to fibrosis, disrupts the ECM/ alters fibroblast activity?

[4] Are there any gravimetric data available regarding the heart size, lung edema, spleen size, liver size etc. upon extraction? If these data are available it would be good to include. Is it known if Amylin accumulates in other organs at baseline and during HF? Could Amylin be driving some organ-organ communication? What is the signal driving Amylin release from the pancreas? These should be considered and discussed.

[5] Although changes in PFKFB3 expression are shown to be increased it would be helpful

to examine changes in its phosphorylation at Ser461 and changes in its activity as even though its expression has changed it may not correlate with activity. This is important since in Figure 4 PFKFB3 mRNA expression appears to be significantly increased in NHP with HF; however, this does not seem to be the case for the protein levels. It would be beneficial to explain why this may be the case especially as PFKFB3 is known to undergo posttranslational modification.

If you decreased Amylin levels during HF in the NHP, does this reverse the HF pathology and also reverse the changes in PFKFB3 and HIF1 α expression? This could be tested in Amylin KO mice +/- HF (Liu et al) Does Amylin deposition still occur in conditions of reduced expression PFKFB3 and HIF1 α ? In addition, since AMPK also phosphorylates PFKFB3 at Ser 461 in addition to PKA and PKC it may be beneficial to look at these in addition to downstream targets of PFKFB3 and glycolytic flux. In the same respect, it would be helpful to examine changes in mitochondrial function/metabolism, and oxidative stress. In the same respect, it would be beneficial to look at downstream targets of HIF1 α (e.g. VEGF) and potential changes in posttranslational modifications. Furthermore, the current data would be strengthened with the inclusion of additional co-staining of Amylin with both PFKFB3 and HIF1 α .

[6] Since Amylin is known to position itself in the sarcolemma- is there any information of which proteins it may associate with there and any additional effects it may mediate especially as it can induce sarcolemmal Ca²⁺ leak. Only cytosolic Ca²⁺ levels were measured in the hiPSC-CMs, comprehensive Ca²⁺ transient analysis in addition to changes in levels of Ca²⁺ handling proteins would need to be performed.

[7] Additional discussion is needed to address the metabolic changes that are known to occur in the heart during baseline and HF and the impact of Amylin. Do HIF1 α and PFKFB3 expression change in heart during HF in humans- this should be discussed. It has been shown by Gibb et al that if you increase or decrease cardiac PFKFB3 expression remodeling occurs in addition to changes in cardiac metabolism. In addition, If PFKFB3 activity is enhanced this would also have implications for autophagic, insulin signaling, and p38/MAPK signaling in the heart. PFKFB3 not only regulates glucose metabolism but also regulates mitochondrial metabolism, glycerolipid synthesis and the pentose phosphate

pathway. It may be beneficial to interrogate and discuss further the latter given the fact that Montenmuro et al have shown that enhanced PFKFB3 increased pyruvate to lactate and stimulated the pentose phosphate pathway in the beta cell Amylin (IAPP) model.

Minor comments

[1] Several grammatical and spelling errors exist in the manuscript. For example, in figure legends 1 and 2 scale bar is misspelt. Please address.

[2] Some of the immunoblots are over cropped and hard to see (e.g. Fig 5 and 7) and it would be helpful to see more of the blot to fully appreciate the data.

[3] The authors discussed using commercial sera from control NHPs and those with HF were

these sera obtained from the NHPs from which the other data were obtained? Or are these a parallel set of NHPs? Please clarify.

[4] The HF image for Figure 4e appears to be cross-sectional. It would be better to include an image from the same orientation as the control image

[5] Please include quantification for the staining in Figures 2, 5, and 7

[6] The HIF1 α inhibitor, YC-1 used in the cell studies not only inhibits HIF1 α but is an activator of soluble guanylyl cyclase. The potential implications of this should be discussed.

[7] The NHP with HF have wider QRS interval consistent with systolic heart failure; however, it is surprising this is the only EKG change observed. This should be discussed

[8] What is the expression of Amylin receptor in NHP? Does this change between control and HF NHP? This should be discussed.

Reviewer #2 (Remarks to the Author):

The report by Liu and colleagues investigated a mechanism by which amylin aggregates in cardiomyocytes (CMs) could induce heart failure ¹. For their studies, the authors analyzed tissue from non-human primates with or without HF, and used two in vitro methods to determine if amylin treatment upregulates HIF1 α pathways in CMs. In general the manuscript was well-written and easy to follow. While this connection among HF, amylin aggregates and HIF1 α appears novel, I have several questions/comments on the methods and analyses used, clarification of which will make the findings more compelling.

Major comments:

1. Please provide evidence of specificity for all antibodies used. How were the antibodies validated? Were no-primary antibody controls used? Higher mag photos would allow the reader to better visualize the positive staining.
2. I do not find the details on the human amylin used for incubation studies. There is reference to the procedure ("as previously described"), but I don't find the initial description. How do you know the effects you see (e.g. on HIF1 α and PFKFB3) is caused by aggregated amylin rather than the unaggregated form? What is the time course of human aggregation in culture? How can you verify that aggregates are indeed present in your experiments? What is known about the presence of amylin receptors in the hiPSCCMs?
3. Provide more information on the NHPs. Diet, housing conditions, sex? Was there something specific that predisposed these specific NHPs to HF? While it doesn't appear statistically significant, is the difference in age between the control and HF group relevant?

Minor comments:

1. Briefly introduce the pluripotent stem cell-CMs and isolated rat ventricular CMs as a model in introduction.
2. For the presentation of results, always have CTRL data in the first position (discussed

first, pictured in the first/top/left position.) Notably in Fig. 1.

3. Fig. 1: Show higher magnification photos of critical pathologies as insets.

4. Fig. 2: Was amylin in the HF islet different than control?

As above, how were these antibodies validated?

5. Fig. 3: Why are the group numbers different from the Tables?

Would be helpful if the points in the scatter plots were labeled CTRL or HF.

6. Fig. 4: Why are there so many data points in a and d?

7. Fig. 6: Label each individual graph a through f.

As in Figure 4, there are inconsistencies in sample size ($n = 3$) and what is depicted graphically. Please make clearer how the individual data points were derived.

8. Fig. 7: Please label individual graphs/panels for ease of reading. As in Fig. 6, there's inconsistency in sample size ($n = 3$) and what is depicted graphically. Please make clearer how the individual data points were derived.

9. Define MMP at first use.

10. Fig. 8: Define group sizes.

Response to Reviewer #1

We are very grateful to this reviewer for very informative and constructive comments. We have revised and improved our manuscript accordingly. Our responses to the specific comments are as follows.

The authors goal was to address the question what is drives the accumulation of Amylin during heart failure in non-human primates and how this amylin may regulate cardiac metabolism and function. The authors showed that non-human primates (NHP) with heart failure ¹ had increase cardiac and circulating Amylin levels in the setting of changes consistent with systolic HF which contributed to increases in the expression of the metabolic proteins, HIF1 α and PFKFB3 and which may contribute to changes in Ca²⁺ levels and parameters of mitochondrial function. Although the findings are of particular interest, a number of concerns exist, which reduce the overall enthusiasm for the manuscript in its current form, which would need to be addressed.

Major comments

[1] The NHP used in this study are not appropriately age-matched as the NHP with HF are significantly older than the control NHPs. What is the reason for this difference? It should be addressed. The authors have shown that the NHP with HF had significantly higher body weights and BMI -could this be due in part to increase in age especially since severe systolic heart failure often associated with a decreased in body weight. What is contributing cause to the HF observed in the NHP in this study?

Response:

We appreciate these comments and certainly agree that aging is a potential risk factor for the whole-body system, especially the heart. In our study, we postulated no direct link between cardiac amylin deposition or cardiac impairment and age, as neither amylin deposition nor marked myocardial abnormality was observed in hearts from both young (less than 10 years old, n = 3) and aged (more than 14 years old, n = 4) controls among the healthy NHPs (revised Fig. 1a), although healthy controls were relatively younger compared with the HF group. In addition, tightly age-matched tissues from healthy NHPs (without other metabolic complications) are very difficult to obtain. Therefore, we used the relatively age-matched NHPs in this study to report our findings. The information about age, gender, body weight and BMI for all NHPs whose heart tissues were used in this study are listed in Supplementary Table 1.

We have added “Age, gender, body weight and BMI data for the 13 NHPs used in this study are included in Supplementary Table 1.” in method section (page 15, line 396-397).

We believe NHPs with HF showing significantly higher body weights and BMIs is mainly because these animals were fed high fat diet (total energy: 4.15 kcal/g with 1.20 mg/kcal cholesterol) for 2 years. This may not due to an increase in age because the body weight and BMI of the aged healthy controls (more than 14 years old) are similar to that of the younger healthy animals (less than 10 years old). High fat diet could be the contributing cause to the HF observed in the NHP in this study. We have added sentences in the method part of the revised

manuscript (page 15, line 390-392).

Inserted sentences (page 15, line 390-392):

“CTL animals were fed a normal calorie diet. Meanwhile, NHPs with HF were fed the vendor’s proprietary high fat diet for 2 years before developing HF.”

[2] It is unclear from the manuscript text as to what sex the NHP used in this study were. This data needs to be reported as there are documented differences in HF with respect to sex and it has been shown by Liu et al that cardiac Amylin accumulation is greater in males. In addition, the age, sex, and weights of the Sprague-Dawley rats used for RVCN studies need to be described.

Response:

We thank the reviewer for highlighting this point. We have added gender information for all the NHPs used in this study in Supplementary Table 1. No difference was observed in cardiac amylin accumulation between female NHPs and males NHPs in the HF group, although there was only one female animal in the HF group. We could include more NHPs to compare cardiac amylin accumulation in both genders in further studies. We have added information about age, sex, and weights of Sprague-Dawley rats used in RVCN studies as “(2-month-old, male, around 300 grams)” in the method section (page 15, line 402-403).

[3] The severity scoring system shown in Figure 1 looks to be very subjective as it seems it would be hard to make distinctions between the middle scores (1-3). In addition, the criteria for judging each score is not discussed in detail, so it is difficult to interpret these data. It would be good to include further detail of the scoring system and to provide additional confirmatory assessment methods. Additional assessment could be in the form of staining with picrosirius red, PTAH, CD68, WGA, Troponin I and desmin.

Response:

We appreciate the reviewer’s comments and apologize for the confusion. We added the detailed grading standards for histopathological findings in Supplementary Table 2 and made related changes in the methods section (page 17-18, line 466-468)

Modified sentences (page 17-18, line 466-468):

“Histopathological scores were graded as 0 “-” (no apparent change), 1 (minimal), 2 (moderate), 3 (marked), and 4 (severe) based on the increasing extent and/or complexity of morphological changes. More details for grading standards are described in Supplementary Table 2.”

As the reviewer suggested, we have added additional assessments such as Picro Sirius red staining to confirm the fibrotic area in the failing hearts of NHPs (new Fig. 1i, j and k). The randomly distributed fibrotic areas quantified by both Masson’s trichrome and Picro Sirius red staining were significantly greater in the HF groups compared with the CTL group (new Fig. 1h and k, bar graph). We have also made the corresponding changes in the results section (Page 5, line 116-120), methods section (page 17, line 464-466; page 18, line 472):

Modified sentences (Page 5, line 116-120):

“Compared with those of CTL NHPs (Fig. 1f and i), larger amounts of fibrotic tissues were detected by Masson’s trichrome (star, blue color, Fig. 1g) and Picro Sirius red staining (star, red color, Fig. 1j) in the failing hearts of NHPs. The randomly distributed fibrotic areas were greater in the HF group than the CTL group (Fig. 1h and k, bar graph).”

Inserted sentences (page 17, line 464-466):

“Picro Sirius red staining (Direct red 80, 65548, Sigma-Aldrich, St. Louis, MO and Picric acid (10015714, Xilong, China) was also applied to evaluate the fibrotic area, appearing as red color”.

Inserted sentences (page 18, line 472):

“with Area Quantification (v2.1.3.0; Indica Labs) and the CytoNuclear (v2.0.5.0; Indica Labs) algorithm.”

We appreciate the reviewer’s suggestions to assess cardiac Troponin I (TnI) and desmin levels. We have carried out immunohistochemical staining of TnI and desmin in heart sections from CTL and HF NHPs as shown below in Figure A and Figure B, respectively. We observed significantly lower TnI expression in HF versus CTL animals (Figure A) indicating loss of TnI in the injured myocardium of HF NHPs. The expression of desmin (an intermediate filament protein) showed a slightly decrease in HF animals compared with the CTL (Figure B). These findings suggest that there might be an alteration in the intermediate filament network.

Figure A. Expression of Troponin I (TnI) in heart sections from CTL (n = 7) and HF (n = 6) NHPs by immunohistochemistry. The percentages of positive tissue areas are shown on the right panel. Scale bar, 100 μ m. Data represent mean \pm SEM. **P<0.01 by Student’s t-test.

Figure B. Expression of desmin in heart sections from CTL (n = 7) and HF (n = 6) NHPs by immunohistochemistry. The percentages of positive tissue areas are shown on the right panel.

Scale bar, 100 μm . Data represent mean \pm SEM.

In order to fully confirm infiltrative cells in the heart tissue, it would be best to complement the H&E staining with staining for immune cell populations such as macrophages and neutrophils. This is important since macrophages have been reported to have similar changes in PFKFB3 and metabolism changes when activated (similar to Amylin) and also important as it has been shown that Amylin contributes to systemic inflammation.

Response:

We thank the reviewer for this valuable suggestion. To fully confirm infiltrated cells in the heart tissue, we stained for immune cell populations with specific antibodies: CD3 (T cells), CD8 (cytotoxic T cells), CD68 (macrophages), and CD45 (leukocytes) in heart sections from NHPs by immunohistochemistry (Supplementary Fig. 1). The amounts of all immune cell subtypes were significantly higher in the HF group compared with CTL animals. We have added these data as Supplementary Fig. 1 and related sentences in the results section (page 5, line 108-111), discussion section (page 11, line 282) and method section (page 17, line 451-454).

Inserted sentences (page 5, line 108-111):

“Infiltrated immune cells and their subtypes were further classified by specific antibodies against: CD3 (T cells), CD8 (cytotoxic T cells), CD68 (macrophages), and CD45 (leukocytes) using immunohistochemistry (IHC) staining. Their amounts were significantly higher in the HF group compared with CTL animals (Supplementary Fig. 1).”

Inserted sentence (page 11, line 282):

“such as infiltrated immune cells (Supplementary Fig. 1)”

Inserted sentences (page 17, line 451-454):

“anti-CD3 (0.3 $\mu\text{g}/\text{mL}$, CME324C, Biocare Medical, Pacheco, CA), anti-CD8 (4 $\mu\text{g}/\text{mL}$, ab178089, Abcam, Cambridge, UK), anti-CD45 (1 $\mu\text{g}/\text{mL}$, M0701, Dako, Santa Clara, CA), and anti-CD68 (0.1 $\mu\text{g}/\text{mL}$, CM033B, Biocare Medical, Pacheco, CA), as well as related isotype controls.”

Furthermore, is anything known as to whether Amylin contributes to fibrosis, disrupts the ECM/ alters fibroblast activity?

Response:

We appreciate the reviewer’s question. A recent study found that brain vascular amylin deposition cause the loss of endothelial cell coverage with the degradation of basement membrane structure and endothelium (Ly H. et al., *Ann Neurol.* 2017, 82(2): 208-222). Another study reported that the level of brain natriuretic peptide (BNP), a well-known marker of cardiac hypertrophy and fibrosis, is significantly elevated in hearts from human amylin transgenic rats vs. healthy animals (Despa S. et al., *Circ Res.* 2012, 110(4): 598-608). These results suggest that amylin may contribute to the regulation of fibrotic processes. Furthermore, amylin modulates extracellular matrix metabolism in human nucleus pulposus cells, playing important roles in maintaining the proper structure and tissue homeostasis of disc extracellular matrix

(Wu X. et al., Cell Death Discov. 2017,3:16107). Amylin can also alter fibroblast activity as mouse embryonic NIH-3T3 fibroblasts treated with human amylin exhibit significant reductions in fibroblast viability and proliferation (Bolarinwa O, et al., Sci Rep. 2020.10(1):95). Following the reviewer's suggestion, we have added related sentences in the discussion section (page 11, line 288-290).

Inserted sentence (page 11, line 288-290):

“Previous studies have reported that amylin may contribute to fibrosis, which includes disruption of basement membrane structure, regulation of extracellular matrix metabolism, and alteration of fibroblast activity. Thus,”

[4] Are there any gravimetric data available regarding the heart size, lung edema, spleen size, liver size etc. upon extraction? If these data are available it would be good to include.

Response:

We appreciate these comments and understand the reviewer's concern. We agree with the reviewer that gravimetric data for each organ are very important. Unfortunately, NHPs tissues are commercially acquired from vendors and gravimetric data are not available in their records. The vendors explained that they have to collect these NHPs tissues as formalin fixed paraffin embedded (FFPE), optimal cutting temperature (OCT) and snap frozen specimens during necropsy. Such procedures should be completed in a very short time to avoid protein/DNA/RNA degradation.

Is it known if Amylin accumulates in other organs at baseline and during HF? Could Amylin be driving some organ-organ communication? What is the signal driving Amylin release from the pancreas? These should be considered and discussed.

Response:

Thanks for your informative comments. We have assessed amylin levels in the pancreas (Fig. 2a, g, Supplementary Fig. 2a-b) and kidneys (Supplementary Fig. 2c-d) from NHPs with CTL and HF, in addition to the heart. Pancreatic amylin levels were relatively higher in the HF group compared with CTL animals, although not significant (Supplementary Fig. 2a-b). No amylin deposition was observed in CTL kidneys (n = 7); meanwhile, amylin deposition was found mostly in kidney nucleus in HF NHPs (n = 6) as shown in Supplementary Fig. 2d. This was consistent with previous studies reporting amylin deposits in the kidneys from human amylin transgenic (HIP) rats with cardiac diastolic dysfunction and hypertrophy, but not in wild type rat kidneys (Despa S. et al., Circ Res. 2012, 110(4): 598-608). Treatment of hyperamylinemia dramatically reduced amylin incorporation in the heart and kidneys in HIP rats (Srodulski S et al., Circulation. 2013, 130: A13963). These data suggest that amylin may play a role in driving cardio-renal communication. According to the reviewer's comments, we have added some sentences in the results section (page 6, line 134-138).

Inserted sentences (page 6, line 134-138):

“In addition to the heart and pancreas, we also observed amylin deposits in kidneys of NHPs with HF, but not from CTL (Supplementary Fig. 2c and d). This was consistent with previous studies reporting amylin deposits in the kidneys of HIP rats (with impaired cardiac function)

but not from wild type animals. Treatment of hyperamylinemia dramatically reduced amylin incorporation in the heart and kidneys of HIP rats, suggesting amylin may play a role in driving cardio-renal communication.”

What is the signal driving Amylin release from the pancreas? These should be considered and discussed.

Response:

We thank the reviewer for this insightful comment. Amylin is a 37-amino acid peptide hormone co-stored and co-secreted with insulin by pancreatic β -cells. It is released in response to glucose, lipids, amino acids (e.g., arginine) or the combination thereof. Under normal conditions, monomeric amylin is soluble and functions to control blood glucose levels by inhibiting food intake and slowing gastric emptying (Kiriya, Y., & Nochi, H., 2018. *Cells*, 7, 95; Ogawa A, et al., *J Clin Invest*. 1990;85(3): 973-976; Qi D, et al., *Am J Physiol Endocrinol Metab*. 2010;298(1): E99-E107). However, patients with obesity and pre-diabetes usually show peripheral insulin resistance which triggers β -cells to produce more insulin in parallel with increased synthesis of amylin. When over-secreted, human amylin forms toxic oligomers in the secretory vesicles of β -cells and amyloid fibrils extracellularly in the islet, contributing to β -cell apoptosis and full development of diabetes (Seaquist ER, et al., 1996, *J Clin Invest* 97:455-460; Paulsson, JF, et al., *Diabetologia* 2006, 49, 1237-1246; Despa, S, et al., *J Am Heart Assoc*.2014;3:e001015).

In this study, average BMI in HF NHPs (fed high fat diet for 2 years) was higher than $>30 \text{ kg/m}^2$ (considered to indicate obesity) (Table 2). Intravenous glucose tolerance test (IVGTT) results in these animals (Supplementary Fig. 8) showed impaired glucose tolerance in HF NHPs, although no significant change was observed in insulin response. High fat diet, either directly or indirectly, causes increased insulin release from islet beta cells and primary hyperinsulinemia. Therefore, it is likely that high fat diet is the direct cause of amylin release from pancreatic islet, eventually leading to cardiac amylin accumulation. In accordance with the reviewer’s comments, we have added some sentences in the discussion section (page 12, line 297-299).

Inserted sentences (page 12, line 297-299):

“This is likely due to the high fat diet leading to amylin release into the blood. When over-secreted, amylin forms toxic oligomers, contributing to β -cell dysfunction, as evidenced by impaired glucose tolerance in HF NHPs in this study (Supplementary Fig. 8).”

[5] Although changes in PFKFB3 expression are shown to be increased it would be helpful to examine changes in its phosphorylation at Ser461 and changes in its activity as even though its expression has changed it may not correlate with activity. This is important since in Figure 4 PFKFB3 mRNA expression appears to be significantly increased in NHP with HF; however, this does not seem to be the case for the protein levels. It would be beneficial to explain why this may be the case especially as PFKFB3 is known to undergo posttranslational modification.

Response:

We appreciate the reviewer for pointing this out. We have assessed the level of phosphorylation

of PFKFB3-Ser461 in NHPs. As the reviewer expected, in addition to increased total expression, phosphorylation of PFKFB3-Ser461 was also significantly increased in the HF group compared with the CTL NHPs (new Fig. 4i and j). We have added related sentences in the results section (page 8, line 187-188) and the discussion section (page 12, line 316-317).

Inserted sentences (page 8, line 187-188):

“In addition to increased total expression, phosphorylation of PFKFB3-Ser461 was also significantly enhanced in the HF group compared with CTL NHPs (Fig. 4i and j).”

Inserted sentence (page 12, line 316-317):

“In addition, phosphorylation at Ser461 of PFKFB3 (Fig. 4i and j) was also highly induced, which could further increase kinase activity and enhance glycolysis.”

If you decreased Amylin levels during HF in the NHP, does this reverse the HF pathology and also reverse the changes in PFKFB3 and HIF1 α expression? This could be tested in Amylin KO mice +/- HF (Liu et al)

Response:

We appreciate the reviewer for this intriguing comment. A previous study has confirmed that lowering human amylin incorporation in cardiac myocytes and blood cells attenuates cardiac hypertrophy and left ventricular dilation in human amylin overexpressing (HIP) rats. (Despa, J Am Heart Assoc.2014;3:e001015). Therefore, we postulated that HF pathology in NHPs could be reversed with decreased amylin levels during HF. We also evaluated the levels of PFKFB3 and HIF1 α in heart samples from Amylin KO and WT mice, and found decreased PFKFB3 and HIF1 α protein levels in Amylin KO mice +/- compared with WT animals (Supplementary Fig. 5). These data are consistent with previous findings that compared with WT littermates, Amy-KO mice had improved glucose intolerance (Gebre-Medhin S, et al., Biochem Biophys Res Commun. 1998,250(2):271-277), and glucose intolerance can be caused by hypoxia (Oltmanns KM, et al., Am J Respir Crit Care Med. 2004,169(11):1231-1237); meanwhile, treatment of HIP rats with an amylin lowering compound reverses HIF1 α activation in kidneys (Verma N, et al., Kidney Int. 2020, 97(1):143-155). Based on the above findings, we hypothesized that decreasing amylin levels during HF in NHPs could also reverse the changes in PFKFB3 and HIF1 α expressions. However, since Amylin KO mice +/- HF model is currently unavailable in our group, we were could not assess HF pathology, and PFKFB3 and HIF1 α expressions directly in this model. We have inserted sentences in the results section (page 9, line 216-219), discussion section (page 13, line 329-335) and method section (page 15, line 398-401).

Inserted sentences (page 9, line 216-219):

“In addition, we observed decreased HIF1 α and PFKFB3 protein levels in amylin knockout (Amy-KO) mice compared with their wild type (WT) littermates (Supplementary Fig. 5). These findings indicate that the presence of amylin affected basal cardiac HIF1 α and PFKFB3 regulation.”

Inserted sentences (page 13, line 329-335):

“To determine whether human amylin drives the expression of HIF1 α and PFKFB3, we

measured HIF1 α and PFKFB3 levels in hiPSC-CMs treated with a human amylin antibody for 24 h prior to human amylin exposure (Fig. 6a and b) and also in Amy-KO mice (Supplementary Fig. 5). The decreased HIF1 α and PFKFB3 expression reported in previous studies indicated that the presence of amylin might affect cardiac HIF1 α and PFKFB3 regulation. Indeed, previous investigations showed that compared to WT littermates, Amy-KO mice showed improved glucose intolerance, which can be caused by hypoxia, and treatment of HIP rats with an amylin lowering compound reversed HIF1 α activation in kidneys.”

Inserted sentences (page 15, line 398-401):

“Fresh hearts from Amy-KO mice were obtained from Shanghai Model Organisms, LTD., where the Amy-KO mouse was generated via the CRISPR/Cas9 gene editing technique targeting mouse IAPP exon 3 in the C57BL/6J strain. Totally 5 Amy-KO mice were assessed in the study (4 males and 1 female) at 2 months of age. Age- and sex-matched WT littermates served as the control group.”

Does Amylin deposition still occur in conditions of reduced expression PFKFB3 and HIF1 α ?

Response:

Thank you for this valuable question. To address this question, we assessed amylin deposition in hiPSC-CMs treated with siRNAs targeting HIF1 α (siHIF1 α) and PFKFB3 (siPFKFB3), separately, prior to human amylin exposure. We found slightly decreased amylin deposition in both treatment groups (Supplementary Fig. 6). This was in line with the literature as silencing PFKFB3 suppresses human amylin induced glycolytic flux and increased intracellular Ca²⁺ levels (Montemurro, C, et al., Nat Commun, 2019, 10, 2679. Following the reviewer’s advice, we have added related sentences in the results section (page 9, line 225-227) and the discussion section (page 12-13, line 324-328).

Inserted sentence (page 9, line 225-227):

“In addition, we observed slightly decreased amylin deposition in hiPSC-CMs treated with siRNAs targeting HIF1 α (siHIF1 α) and PFKFB3 (siPFKFB3), separately, prior to human amylin treatment (Supplementary Fig. 6).”

Inserted sentences (page 12-13, line 324-328):

“Of note, in this study, we confirmed that PFKFB3 upregulation was HIF1 α -dependent (Fig. 6d). In addition, HIF1 α and PFKFB3 participate in mediating human amylin deposition as silencing of either gene effectively suppressed human amylin accumulation (Supplementary Fig. 6). This was in line with previous finding that silencing PFKFB3 suppressed human amylin induced increased glycolytic flux and intracellular Ca²⁺ levels. Hence, modulation of HIF1 α and PFKFB3 signaling may help develop novel treatments for HF.”

In addition, since AMPK also phosphorylates PFKFB3 at Ser 461 in addition to PKA and PKC it may be beneficial to look at these in addition to downstream targets of PFKFB3 and glycolytic flux.

Response:

We thank the reviewer for this insightful comment. Following your suggestion, we have assessed the protein levels of AMPK, PKA, and PKC in heart homogenates from CTL (n = 7) and HF (n = 6) NHPs. We observed significantly increased levels of PKC in HF NHPs. Meanwhile, no protein expression alterations were found for AMPK and PKA as shown in below. Therefore, increased expression of PKC may contribute to the phosphorylation of PFKFB3-Ser461 in HF NHPs.

AMPK, PKA and PKC expression levels in heart samples from CTL and HF NHPs. (a) Representative image for Western blot analysis with rabbit anti-AMPK α antibody (2532s, Cell Signaling Technology), rabbit anti-PKC α antibody (2056S, Cell Signaling Technology), and rabbit anti-PKA C- α antibody (4782S, Cell Signaling Technology) respectively, for assessing heart homogenates from CTL (n = 7) and HF (n = 6) NHPs. (b) Quantifications of AMPK, PKC and PKA expression levels in CTL and HF NHPs. Data represent mean \pm SEM. *P<0.05 by Student's t-test.

In the same respect, it would be helpful to examine changes in mitochondrial function/metabolism, and oxidative stress.

Response:

Per the reviewer's advice, we added data about ATP production in hiPSC-CMs treated with human amylin (Fig. 8d). Consistent with our finding that loss of mitochondrial membrane potential in amylin stressed hiPSC-CMs, we observed significantly decreased ATP levels in amylin treated cells. We made related changes in the results section (page 10, line 250-251; page 10, line 262) and discussion section (page 14, line 357; page 14, line 360).

Modified sentence (page 10, line 250-251):

“mitochondrial membrane potential (MMP), and cellular ATP levels in hiPSC-CMs incubated with aggregated human amylin”

Modified sentence (page 10, line 262):

“loss of MMP (Fig. 8c) and impaired ATP production (Fig. 8d) in amylin-stressed cells”

Modified sentences (page 14, line 357):

“meanwhile, MMP and ATP play important roles in the regulation of ROS.”

Modified sentences (page 14, line 360):

“as well as reduced MMP levels and ATP production in hiPSC-CMs”

Usually, ROS production is dramatically increased under oxidative stress. Therefore, we

believe ROS is a reliable marker of oxidative stress. The data depicting ROS production in response to amylin have been shown in Fig. 8b.

In the same respect, it would be beneficial to look at downstream targets of HIF1 α (e.g. VEGF) and potential changes in posttranslational modifications.

Response:

We understand the reviewer's concern and have assessed the mRNA and protein levels of VEGFA (one of the main members of the VEGF family) in both NHPs and hiPSC-CMs treated with amylin as shown below in Figure C. However, no significant changes were found in heart samples from HF NHPs compared with CTL group (below Figure C a-c). What's more, no VEGFA mRNA and protein changes were found in hiPSC-CMs treated with amylin (below Figure C d-e), although VEGF is one of the well-known targets of HIF1 α . These results suggest that VEGF is very likely regulated under another mechanism other than HIF1 α in this study.

Figure C:

Figure C. VEGFA levels in hearts from CTL and HF NHPs (a-c) and in hiPSC-CMs treated with human amylin (d-e). (a) Immunohistochemistry for VEGFA detection (brown color, sc-

7269, Santa Cruz, CA) in heart sections from CTL and HF NHPs. The percentages of positive tissue areas are shown on the right. (b) Western blot analysis with anti-VEGFA antibody (b, sc-7269, Santa Cruz, CA) for assessing heart homogenates from CTL (n = 7) and HF (n = 6) NHPs. (c) RT-PCR for detecting VEGFA mRNA (primers are listed in below) in NHPs as in panels (a) and (b). (d) Western blot analysis with anti-VEGFA in the lysates from hiPSC-CMs treated with human amylin (n=3/group). (e) RT-PCR for VEGFA mRNA detection (primers listed in the table below) in hiPSC-CMs as in panel (d). Scale bar, 100 μ m. Data represent mean \pm SEM.

Gene name	Forward primer sequence	Reverse primer sequence
GAPDH (NHPs)	TTTTCTCTTGCATCGCCAGC	ATGACGAGCTTCCCGTTCTC
VEGFA (NHPs)	GCTCAGAGCGGAGAAAGCAT	AGCTGCGCTGATAGACATCC
GAPDH (Human)	GCACCGTCAAGGCTGAGAAC	AGGGATCTCGCTCCTGGAA
VEGFA (Human)	TCACCAAGGCCAGCACATAG	ACAGGGACGGGATTTCTTGC

Furthermore, the current data would be strengthened with the inclusion of additional co-staining of Amylin with both PFKFB3 and HIF1 α .

Response:

Primary antibody	Secondary antibody
Mouse anti-Amylin antibody (1:50, sc377530, Santa Cruz)	Goat anti-mouse IgG secondary antibody labeled with Alex Fluor 488 (1:500, ab150113, Abcam)
Goat anti-HIF1 α antibody (1:20, AF1935, R&D)	Donkey anti-Goat IgG Secondary Antibody labeled with Alexa Fluor 594 (1:250, a11058, Thermo Fisher)
Rabbit anti-PFKFB3 antibody (1:100, ab181861, Abcam)	Donkey anti-rabbit IgG secondary antibody labeled with Alex Fluor 647 (1:500, ab150075, Abcam)

Per the reviewer’s suggestion, we have included the data for co-staining of amylin with both PFKFB3 and HIF1 α in hiPSC-CMs in Supplementary Fig. 4. The antibodies used in this experiment are listed below:

In order to co-stain amylin with both HIF1 α and PFKFB3, different antibodies were used for HIF1 α (Goat anti-HIF1 α , 1:20, AF1935, R&D) and human amylin (1:50, sc377530, Santa Cruz) from those in main Fig. 5 and Fig. 7 (mouse anti-HIF1 α , 1:200, ab16066, Abcam) and human amylin (rabbit anti-human amylin, 1:400, T4157, Peninsula Laboratories). We observed amylin co-localized both with HIF1 α and PFKFB3 in hiPSC-CMs treated with amylin (Supplementary Fig. 4). We have added related sentences in results section (page 9, line 211-212).

Inserted sentences (page 9, line 211-212):

“Moreover, immunofluorescence showed co-localization of amylin with HIF1 α and PFKFB3 in hiPSC-CMs incubated with human amylin (Supplementary Fig. 4).”

[6] Since Amylin is known to position itself in the sarcolemma- is there any information of which proteins it may associate with there and any additional effects it may mediate especially as it can induce sarcolemmal Ca²⁺ leak. Only cytosolic Ca²⁺ levels were measured in the hiPSC-CMs, comprehensive Ca²⁺ transient analysis in addition to

changes in levels of Ca²⁺ handling proteins would need to be performed.

Response:

We appreciate these comments and certainly agree that the proposed experiments would provide valuable information. Previous work found that binding of amylin to calcitonin gene-related peptide (CGRP) receptors is an important mechanism of amylin internalization in pancreatic β -cells and neurons (Tripathi S. et al. PLoS One, 2013, 8, e73080; Jhamandas JH, et al., J Neurosci, 2004, 24, 5579-5584). However, this doesn't seem to be the case for cardiomyocytes as another study reported that blocking CGRP receptors does not reduce amylin-mediated sarcolemmal Ca²⁺ leak in isolated cardiac myocytes (Liu M. et al., Biochim Biophys Acta Mol Basis Dis, 2018, 1864, 1923-1930). This suggests that other mechanisms contribute to amylin-mediated myocyte Ca²⁺ dysregulation, such as sarcolemmal injury. This was supported by a recent work demonstrating that oligomerized amylin is prone to interact with lipid peroxidation end products such as 4-hydroxy-2-nonenal (4-HNE) and malondialdehyde (MDA) to form toxic 4-HNE/MDA-amylin adducts. These adducts, in turn, may accelerate myocardial amylin deposition in the sarcolemma and cause sarcolemmal Ca²⁺ leak (Liu M. et al., Diabetes, 2016, 65:2772-2783). We have included new data for comprehensive Ca²⁺ transient analysis and the levels of Ca²⁺ handling proteins in the revised manuscript (Supplementary Fig. 7). The results showed that 6.25 μ M human amylin increased Ca²⁺ transient amplitude at spontaneous beating (SB) and paced at 1, 1.5 and 2 Hz, respectively, although not significantly. As the reviewer suggested, we have measured the protein expression levels of sarcoplasmic reticulum Ca²⁺-ATPase (SERCA), phospholamban (PLB) (endogenous SERCA inhibitor), and Na⁺/Ca²⁺ exchanger (NCX, the main Ca²⁺ extrusion pathway) in hiPSC-CMs treated with 6.25 μ M human amylin for 2h. However, SERCA, PLB, and NCX protein amounts were unaltered in amylin stressed cells. We infer that amylin oligomers could acutely elevate intracellular Ca²⁺ and Ca²⁺ transients, but altered functions of SERCA, PLB, and NCX may be longer-term effects, as we found that NCX expression was significantly decreased in NHPs with HF (Supplementary Fig. 10). Per the reviewer's suggestion, we have added related sentences in the results section (Page 10, line 254-260) and the discussion section (page 13-14, line 340-354).

Inserted sentences (Page 10, line 254-260):

“In addition, Ca²⁺ transient amplitude and Ca²⁺ handling proteins were measured in hiPSC-CMs exposed to human amylin for 2 h (Supplementary Fig. 7). The results showed that 6.25 μ M of human amylin increased Ca²⁺ transient amplitude at spontaneous beating (SB) and paced at 1, 1.5 and 2 Hz, respectively, although not significantly (Supplementary Fig. 7c). However, protein expression levels of sarcoplasmic reticulum Ca²⁺-ATPase (SERCA), phospholamban (PLB) (endogenous SERCA inhibitor), and Na⁺/Ca²⁺ exchanger (NCX) remained unaltered in amylin-stressed cells (Supplementary Fig. 7e).”

Inserted sentences (page 13-14, line 340-354):

“Elevated Ca²⁺ transient amplitude was also observed in hiPSC-CMs treated with amylin at SB and paced at 1, 1.5 and 2 Hz, respectively (Supplementary Fig. 7c). However, no changes were observed in SERCA, PLB or NCX protein amounts in amylin treated cells (Supplementary Fig. 7e). We postulate that amylin oligomers could acutely elevate Ca²⁺ transients, while altered

functions of SERCA, PLB, and NCX may require longer-term effects, as NCX expression was significantly decreased in NHPs with HF (Supplementary Fig. 10). Previous findings indicate that binding of amylin to calcitonin gene-related peptide (CGRP) receptors serves as an important mechanism for amylin internalization in pancreatic β -cells and neurons. However, this does not appear to apply to cardiomyocytes as another study reported that blocking CGRP receptors did not reduce amylin-mediated sarcolemmal Ca^{2+} leakage in isolated cardiomyocytes suggesting that other mechanisms, e. g., sarcolemmal injury, contribute to amylin-mediated myocyte Ca^{2+} dysregulation. Studies have also reported that amylin receptors, comprised of heterodimers of calcitonin receptors (CTR) and one of three receptor activity-modifying proteins (RAMP1-3), are widely expressed in the central nervous system of NHPs. Additionally, mRNA expression levels of RAMP1, RAMP2 and RAMP3 are elevated in different rat heart dysfunction models. Given that RAMPs form a key part of amylin receptors, we infer that these amylin receptors may also contribute to Ca^{2+} signaling.”

[7] Additional discussion is needed to address the metabolic changes that are known to occur in the heart during baseline and HF and the impact of Amylin. Do HIF1 α and PFKFB3 expression change in heart during HF in humans- this should be discussed. It has been shown by Gibb et al that if you increase or decrease cardiac PFKFB3 expression remodeling occurs in addition to changes in cardiac metabolism. In addition, If PFKFB3 activity is enhanced this would also have implications for autophagic, insulin signaling, and p38/MAPK signaling in the heart. PFKFB3 not only regulates glucose metabolism but also regulates mitochondrial metabolism, glycerolipid synthesis and the pentose phosphate pathway. It may be beneficial to interrogate and discuss further the latter given the fact that Montemurro et al have shown that enhanced PFKFB3 increased pyruvate to lactate and stimulated the pentose phosphate pathway in the beta cell Amylin (IAPP) model.

Response:

We appreciate your authoritative advice and certainly agree that further discussion would provide valuable information. The normal healthy heart keeps metabolic flexibility with utilization of different energy substrates including fatty acids, carbohydrates, ketones and amino acids at different rates to maintain its contractile function. In HF, reduced cardiac function is accompanied by altered cardiac energy metabolism due to impaired mitochondrial ATP production and metabolic flexibility. The failing heart is generally considered to have increased glycolysis, depressed glucose oxidation and increased ketone body oxidation (Rosano GM & Vitale, C. Card Fail Rev. 2018, 4(2): 99-103; Karwi QG, et al., Front Cardiovasc Med. 2018, 5:68). Recent studies reported that amylin increases glycolysis and pentose phosphate pathways by reducing glucose oxidation through the TCA cycle and pyruvate anaplerotic reactions in β -cells (Montemurro C. et al., Nat Commun. 2019,10(1):2679). It is also known that cardiac amylin accumulation in HF accelerates cardiac hypertrophy and remodeling (Despa S. et al., Circ Res. 2012, 110(4): 598-608) partly due to amylin aggregates that can alter mitochondrial disarrangement and mitochondrial function in cardiomyocytes (Despa S. et al., J Am Heart Assoc. 2014, 3:e001015). This is consistent with our finding that the level of the glycolytic end-product, lactate, is significantly increased in amylin treated cells which indicates an advantage for glycolysis flux with the diversion of pyruvate to lactate.

HIF1 α was reported to increase in peri-infarct areas of human ischemic heart tissues (Lee, S. H. et al. *N Engl J Med*, 2000, 342, 626-633) and in human cardiac hypertrophy (Krishnan J, et al., *Cell Metab*, 2009, 9(6):512-24). PFKFB3, a target gene of HIF1 α , has low basal expression levels but is strongly induced by hypoxia upon myocardial ischemia (Minchenko O, et al., *FEBS Lett.*, 2003, 554, 264-270). A recent study found that genetically modified cardiac-specific PFKFB mutants regulate myocardial metabolism and cardiac remodeling (Gibb AA. et al., *Circulation*, 2017, 136, 2144-2157). In addition, PFKFB3 regulates not only glucose metabolism but also mitochondrial metabolism, glycerolipid synthesis and the pentose phosphate pathway (Minchenko O, et al., *FEBS Lett.*, 2003, 554, 264-270). What's more, enhanced PFKFB3 activity also affects autophagy, insulin signaling (Bockus LB, et al., *J Am Heart Assoc.* 2017, e007159), and p38/MAPK signaling (Marsin AS, et al., *Curr Biol*, 2000,10:1247-1255) in the heart. Of note, in this study, amylin deposition was decreased by silencing HIF1 α and PFKFB3 in hiPSC-CMs (Supplementary Fig. 5). Hence, modulation of HIF1 α and PFKFB3 signaling may help develop novel treatments for HF.

Per the reviewer's suggestion, we have inserted related sentences in the discussion section (page12-13, line 318-328 and page 14, line 362-370).

Inserted sentences (page12-13, line 318-328):

"HIF1 α was reported to increase in ischemic heart tissues and cardiac hypertrophy in humans. Meanwhile, PFKFB3, a target gene of HIF1 α , has low basal expression levels but is strongly induced by hypoxia upon myocardial ischemia. A recent study reported that genetically modified cardiac-specific PFKFB mutants regulate myocardial metabolism and cardiac remodeling. In addition, PFKFB3 regulates not only glucose metabolism but also mitochondrial metabolism, glycerolipid synthesis and the pentose phosphate pathway. Moreover, enhanced PFKFB3 activity affects autophagy, insulin signaling, and p38/MAPK signaling in the heart. Of note, in this study, we confirmed that PFKFB3 upregulation was HIF1 α -dependent (Fig. 6d). In addition, HIF1 α and PFKFB3 participate in mediating human amylin deposition as silencing of either gene effectively suppressed human amylin accumulation (Supplementary Fig. 6). This was in line with previous findings that silencing PFKFB3 suppressed human amylin induced increased glycolytic flux and intracellular Ca²⁺ levels. Hence, modulation of HIF1 α and PFKFB3 signaling may help develop novel treatments for HF."

Inserted sentences (page 14, line 362-370):

"The normal healthy heart maintains metabolic flexibility by utilizing different energy substrates including fatty acids, carbohydrates, ketones and amino acids at different rates to support its contractile function. Reduced cardiac function in HF is accompanied by altered cardiac energy metabolism primarily due to impaired mitochondrial ATP production and metabolic flexibility. Increased glycolysis and ketone oxidation rates, as well as downregulated glucose oxidation are common characteristics of the failing heart. Recent studies reported that amylin induces the glycolysis and pentose phosphate pathways by reducing glucose oxidation via the TCA cycle and pyruvate anaplerotic reactions in β -cells. Moreover, cardiac amylin accumulation in HF accelerates cardiac hypertrophy and remodeling partly because amylin aggregates can alter mitochondrial disarrangement and mitochondrial function in

cardiomyocytes.”

Minor comments

[1] Several grammatical and spelling errors exist in the manuscript. For example, in figure legends 1 and 2 scale bar is misspelt. Please address.

Response:

Thank you for pointing this out. We have corrected these errors.

[2] Some of the immunoblots are over cropped and hard to see (e.g. Fig 5 and 7) and it would be helpful to see more of the blot to fully appreciate the data.

Response:

Thanks for your suggestion. We have replaced the indicated immunoblots in revised manuscript in Fig. 5. and Fig. 7.

[3] The authors discussed using commercial sera from control NHPs and those with HF were these sera obtained from the NHPs from which the other data were obtained? Or are these a parallel set of NHPs? Please clarify.

Response:

We appreciate your questions, and have modified the related sentences in the results section (page 6, line 144-145; page 7, line 159-160) as well as in the legend of Fig. 3.

Modified sentence (page 6, line 144-145):

“amylin levels in a parallel set of commercial sera from CTL and HF NHPs (whose heart tissues were not available).”

Modified sentence (page 7, line 159-160):

“ELISA of matched plasma and RBC lysates from thirteen NHPs (CTL group, n = 7; HF group, n = 6, whose heart sections were used for histopathological evaluation in Fig. 1 and amylin assessment in Fig. 2)”

Modified sentence in the legend of Fig. 3:

“Fig. 3. Circulating amylin and HF markers levels in NHPs. (a) Amylin concentration in serum from a parallel set of NHPs (CTL, n = 10; HF, n = 7). (b) Protein levels of ST2, GDF-15, and cTnI in serum as in panel (a). (c-e) The correlations between serum amylin level and ST2 (c), cTnI (d), and GDF-15 (e), respectively, in serum as in panel (a) and (b). Amylin levels in matched plasma (f) and RBC lysates (g) from the same animals whose heart were used for histopathological evaluation in Fig. 1 and amylin assessment in Fig. 2 (CTL, n = 7; HF, n = 6). Data represent mean ± SEM. *P<0.05; **P<0.01 by Student’s t test. The correlations between serum amylin level and ST2 (c), cTnI (d), and GDF-15 (e), respectively, was analyzed by Spearman nonparametric correlation analysis in GraphPad 7.04 (GraphPad Inc., San Diego, CA), and the values for the Spearman r are indicated on the plots.”

[4] The HF image for Figure 4e appears to be cross-sectional. It would be better to include an image from the same orientation as the control image

Response:

Thanks for your suggestion. We have included a new image with the same orientation as the control image in the revised manuscript (Fig. 4f).

[5] Please include quantification for the staining in Figures 2, 5, and 7

Response:

We appreciate your advice. We have included the quantification for staining in Fig. 2j and k, Fig. 5b, 5d, 5i and Fig. 7b, 7f.

[6] The HIF1 α inhibitor, YC-1 used in the cell studies not only inhibits HIF1 α but is an activator of soluble guanylyl cyclase. The potential implications of this should be discussed.

Response:

Thanks for your insightful suggestion. We have included the potential implications of YC-1 in the discussion section (page 14, line 375-377).

Inserted sentences (page 14, line 375-377):

“Interestingly, YC-1 not only is a HIF1 α inhibitor but also intensifies the antioxidant properties of nitric oxide by inducing soluble guanylyl cyclase activation, which may play a protective role in amylin induced ROS production.”

[7] The NHP with HF have wider QRS interval consistent with systolic heart failure; however, it is surprising this is the only EKG change observed. This should be discussed

Response:

We thank the reviewer for pointing this out and the opportunity to correct any miscalculated statistics in Table 1. We also observed significantly shorter QT and QTc intervals in addition to wider QRS interval in HF NHPs. We have made changes in the results section (page 5, line 93-94) and the discussion section (page 11, line 284).

Inserted sentences (page 5, line 93-94):

“shorter QT(c) interval (a risk factor for ventricular tachyarrhythmias and sudden death),”

Inserted sentences (page 11, line 284):

“shortened QT(c) interval,”

[8] What is the expression of Amylin receptor in NHP? Does this change between control and HF NHP? This should be discussed.

Response:

We thank the reviewer for this valuable question. Functional amylin receptors are comprised of heterodimers of calcitonin receptors (CTR) and one of three receptor activity-modifying proteins (RAMP 1-3) which gives rise to at least six different subtypes of amylin receptors. Amylin receptors are widely expressed in the central nervous system of NHPs (Paxinos G. et al., J Chem Neuroanat, 2004,27:217-236; Eftekhari S. et al., J Comp Neurol,2016,524:90-118). Previous studies found that RAMP1 and RAMP3 mRNA expression levels are elevated in the atria and ventricles 6 months post-surgery in a rat HF model (Cueille C, Biochem Biophys Res

Commun, 2002,294:340-346). Increased RAMP2 mRNA levels in the heart was also observed in rat model of isoproterenol-induced myocardial hypertrophy and ischemia (Qi YF, et al., Peptides,2003,24:463-468). Given that RAMPs form a key part of amylin receptors, we postulate that the expression levels of amylin receptors are very likely altered in HF NHPs. We have added related sentences in the discussion section (page 13-14, line 349-354).

Inserted sentences (page 13-14, line 349-354):

“Studies have also reported that amylin receptors, comprised of heterodimers of calcitonin receptors (CTR) and one of three receptor activity–modifying proteins (RAMP 1-3), are widely expressed in the central nervous system of NHPs. Additionally, mRNA expression levels of RAMP1, RAMP2 and RAMP3 are elevated in different rat heart dysfunction models. Given that RAMPs form a key part of amylin receptors, we infer that these amylin receptors may also contribute to Ca²⁺ signaling.”

Response to Reviewer #2

We sincerely appreciate this reviewer for the insightful and constructive comments. We have revised and improved our manuscript according to your suggestions. Our responses to the specific comments are found below.

Reviewer #2 (Remarks to the Author):

The report by Liu and colleagues investigated a mechanism by which amylin aggregates in cardiomyocytes (CMs) could induce heart failure¹. For their studies, the authors analyzed tissue from non-human primates with or without HF, and used two in vitro methods to determine if amylin treatment upregulates HIF1a pathways in CMs. In general the manuscript was well-written and easy to follow. While this connection among HF, amylin aggregates and HIF1a appears novel, I have several questions/comments on the methods and analyses used, clarification of which will make the findings more compelling.

Major comments:

1. Please provide evidence of specificity for all antibodies used. How were the antibodies validated? Were no-primary antibody controls used? Higher mag photos would allow the reader to better visualize the positive staining.

Response:

We thank this reviewer for the insightful comments. We have included the data about specificities of the antibodies used in Figure D in below.

Figure D:

Specificity test for rabbit anti-amylin antibody (T4157, Peninsula Laboratories)

rabbit anti-amylin antibody (T4157, Peninsula Laboratories)

a) hiPSC-CMs were stained with rabbit anti-amylin antibody (T4157, Peninsula Laboratories) and donkey anti-rabbit IgG labelled with Alex Fluor 488 (A32790, Thermo Fisher).

b) hiPSC-CMs were stained with rabbit anti-amylin antibody (T4157, Peninsula Laboratories) that was pre-incubated with amylin peptide (AS-60254-1, Anaspec) at 4 °C overnight and donkey anti-rabbit IgG labelled with Alex Fluor 488 (A32790, Thermo Fisher),

c) hiPSC-CMs were stained only with donkey anti-rabbit IgG labelled with Alex Fluor 488 (A32790, Thermo Fisher),

Specificity test for mouse anti-amylin antibody (sc-377530, Santa Cruz)

- a) IHC staining for mouse anti-amylin antibody (sc-377530, Santa Cruz) in HEK-293 cells transfected with human amylin plasmid. Scale bar, 100 μ m.
 b) IHC staining for mouse anti-amylin antibody (sc-377530, Santa Cruz) in HEK-293 cells transfected with empty vector. Scale bar, 100 μ m.
 c) Western blot analysis of mouse anti-amylin antibody (sc-377530, Santa Cruz) in non-transfected (A) and human amylin transfected (B) HEK293T whole cell lysates (This image is downloaded from it's official website).

Specificity test for rabbit anti-PFKFB3 antibody (ab181861, Abcam) and rabbit anti-phospho461-PFKFB3 antibody (ab202291, Abcam)

- a) IHC staining for rabbit anti-PFKFB3 antibody (ab181861, Abcam) in HEK-293 cells transfected with PFKFB3 plasmid. Scale bar, 100 μ m.
 b) IHC staining for rabbit anti-PFKFB3 antibody (ab181861, Abcam) in HEK-293 cells transfected with empty plasmid. Scale bar, 100 μ m.
 c) Western blot analysis of PFKFB3 (ab181861, Abcam) and phospho461-PFKFB3 (ab202291, Abcam) in HEK-293 cells transfected with empty vector (lane 2), with PFKFB3 (WT) plasmid (lane 3), with PFKFB3 S461A mutant plasmid (lane 4).
 d) Image downloaded from those antibodies' official website. Western blot for PFKFB3 (ab181861, Abcam) and phospho461-PFKFB3 (ab202291, Abcam) in HEK-293 cells transfected with empty vector (lane 1), with PFKFB3 (WT) plasmid (lane 2), with PFKFB3 (WT) plasmid then treated with alkaline phosphatase for 1 hour (lane 3), with PFKFB3 S461A mutant plasmid (lane 4).
 e) hiPSC-CMs were stained with rabbit anti-PFKFB3 antibody (ab181861, Abcam) and donkey anti-rabbit IgG labelled with Alex Fluor 488 (A32790, Thermo Fisher).
 f) hiPSC-CMs were stained with rabbit anti-PFKFB3 antibody (ab181861, Abcam) that was pre-incubated with recombinant human PFKFB3 peptide (ab151856, Abcam) at 4 °C overnight and donkey anti-rabbit IgG labelled with Alex Fluor 488 (A32790, Thermo Fisher).
 g) hiPSC-CMs were stained only with donkey anti-rabbit IgG labelled with Alex Fluor 488 (A32790, Thermo Fisher),

Specificity test for mouse anti-HIF1 α antibody (ab16066, Abcam) and goat anti-HIF1 α antibody (AF1935, R&D systems)

mouse anti-HIF1 α antibody (ab16066, Abcam)

goat anti-HIF1 α antibody (AF1935, R&D systems)

- a) IHC staining for mouse anti-HIF1 α antibody (ab16066, Abcam) in NHPs heart section. Scale bar, 100 μ m.
b) IHC staining for mouse anti-HIF1 α antibody (ab16066, Abcam) that was pre-incubated recombinant Human HIF-1 alpha protein (ab154478) at 4 $^{\circ}$ C overnight in NHPs heart section
c) Western blot analysis of mouse anti-HIF1 α antibody (ab16066, Abcam) in hiPSC-CMs transfected with si-vehicle control ((4390843, Thermo Fisher, lane 1) and siHIF1 α (siRNA ID: s6539, 4390824, Thermo Fisher, lane 2).
e) hiPSC-CMs were stained with goat anti-HIF1 α antibody (AF1935, R&D systems) and donkey anti-goat IgG labelled with Alex Fluor 594 (A-11058, Thermo Fisher)
f) hiPSC-CMs were stained with goat anti-HIF1 α antibody (AF1935, R&D systems) that was pre-incubated with recombinant Human HIF-1 alpha protein (ab154478)) at 4 $^{\circ}$ C overnight and donkey anti-goat IgG labelled with Alex Fluor 594 (A-11058, Thermo Fisher)
g) hiPSC-CMs were stained only with donkey anti-goat IgG labelled with Alex Fluor 594 (A-11058, Thermo Fisher)

Specificity test for mouse anti- α -actinin (sarcomeric) antibody (AF7811, Sigma-Aldrich)

mouse anti- α -actinin (sarcomeric) antibody (AF7811, Sigma-Aldrich)

Lane 1: hiPSC-CMs were transfected si-vehicle control,
Lane 2: hiPSC-CMs were transfected siActinin

- a) hiPSC-CMs were stained with mouse anti- α -actinin (sarcomeric) antibody (AF7811, Sigma-Aldrich) and goat anti-mouse IgG labelled with Alex Fluor 488 (ab150113, Abcam, Cambridge, UK)
b) hiPSC-CMs were stained with mouse anti- α -actinin (sarcomeric) antibody (AF7811, Sigma-Aldrich) that was pre-incubated with α -actinin peptide (AT01, Cytoskeleton, Inc) at 4 $^{\circ}$ C overnight and goat anti-mouse IgG labelled with Alex Fluor 488 (ab150113, Abcam, Cambridge, UK)
c) hiPSC-CMs were stained only with goat anti-mouse IgG labelled with Alex Fluor 488 (ab150113, Abcam, Cambridge, UK)
d) Western blot analysis of mouse anti- α -actinin (sarcomeric) antibody (AF7811, Sigma-Aldrich) in hiPSC-CMs transfected with si-vehicle control ((4390843, Thermo Fisher, lane 1) and siActinin (siRNA ID: s968, 4392420, Thermo Fisher, lane 2).

Specificity test for mouse anti-SERCA2 ATPase antibody (MA3-919, Thermo Fisher)

This SERCA2 ATPase Antibody (MA3-919) antibody was verified by Knockdown to ensure that the antibody binds to the antigen stated. (a) Western blot analysis of for mouse anti-SERCA antibody (MA3-919, Thermo Fisher) in A549 cells with nontransfected (lane 1) transfected with scrambled siRNA (lane 2) and transfected with SERCA2 ATPase siRNA (lane 3). (b) SERCA2 ATPase expression was reduced in A549 transfected with SERCA2 ATPase siRNA (These images were downloaded form its official website).

Specificity test for mouse anti-Sodium/Calcium Exchanger (NCX) antibody (MA3-919, Thermo Fisher)

C) Western blots testing for mouse anti-NCX (MA3-919) in different transfected and non-transfected HEK293 cell lines using anti-NCX mouse monoclonal antibody (MA3-926). The positions of the 120 and 160 kDa protein bands associated with the NCX1 expression are marked on the left side. D) Confocal fluorescence images of different HEK293 cell lines (newly developed and non-transfected) immunostained with mouse anti-NCX (MA3-919) and FITC-labelled secondary antibody. (These images were downloaded form its official website).

Specificity test for rabbit Phospholamban (PLB) antibody (PA5-82945, Thermo Fisher)

This Antibody was verified by Relative expression to ensure that the antibody binds to the antigen stated. a) Relative expression in different tissues in IHC: Detection of differential expression levels of Phospholamban demonstrates antibody specificity. Immunohistochemical analysis of Phospholamban using anti-Phospholamban Polyclonal Antibody (Product #PA5-82945), shows significant staining of Phospholamban in heart muscle and shows minimal or weak staining in skin tissues. b) The relative expression levels of Phospholamban within each tissue is shown using RNA-Seq. (These images were downloaded form its official website).

Specificities of anti-CD3, anti-CD8, anti-CD45, and anti-CD68 antibodies verified by their respective isotype controls.

We have improved the quality of all images for better visualization of positive staining in the revised manuscript.

2. I do not find the details on the human amylin used for incubation studies. There is reference to the procedure (“as previously described”), but I don’t find the initial description.

Response:

We appreciate the reviewer's comments, and have added the information about human amylin (AS-60254-1, Anaspec, Fremont, CA) in the methods section. The amylin oligomerization reaction was prepared in saline at 37 °C with 50 μM recombinant human amylin (AS-60254-1, Anaspec, Fremont, CA). Adult rat cardiomyocytes were then incubated with preformed amylin oligomers (50 μM) for 2 h at room temperature. Since hiPSC-CMs are rather fragile, and high amylin concentration is very cytotoxic in these cells (apoptosis induced by amylin oligomers was evaluated by flow cytometry as shown in Supplementary Fig. 3), we use 6.25 μM amylin oligomers to treat hiPSC-CMs. We have added related sentences in the method section (page 16, line 412-420).

Inserted sentences (page 16, line 412-420):

“Treatment of isolated RVCs with recombinant human amylin. The amylin oligomerization reaction was prepared in saline at 37 °C with 50 μM recombinant human amylin (AS-60254-1, Anaspec, Fremont, CA). Adult rat cardiomyocytes were then incubated with preformed amylin oligomers (50 μM) for 2 h at room temperature.

Flow cytometry for apoptosis assessment in hiPSC-CMs. hiPSC-CMs were treated with preformed amylin oligomers (6.25 μM, 12.5 μM, and 25 μM) at 37 °C for 2 h. After washing, cells were stained with annexin V and propidium iodide (PI) for 15 min at room temperature and analyzed on a BD LSRFortessa cell analyzer. The data were analyzed with the BD FACSDiva software.

Treatment of hiPSC-CMs with recombinant human amylin. hiPSC-CMs were incubated with preformed amylin oligomers (6.25 μM) for 2 h at 37 °C.”

How do you know the effects you see (e.g. on HIF1a and PFKFB3) is caused by aggregated amylin rather than the unaggregated form?

Response:

Thanks for your insightful questions. A previous study has demonstrated amylin oligomer formation in serum by electron microscopy. The electron microscopy images (Figure E) below show that 50 μM human amylin incubated in serum for 1 h forms oligomers and protofibrils (arrow) but not rat amylin (Despa S. et al., *Circ Res.* 2012,110(4): 598-608. Supplementary Materials).

Figure E:

Therefore, we prepared the amylin oligomerization reaction in saline at 37 °C using 50 μM recombinant human amylin (AS-60254-1, Anaspec, Fremont, CA). We used 50 μM preformed amylin oligomers for RVCs. Since hiPSC-CMs are rather fragile, and high amylin concentration is very cytotoxic for these cells (Supplementary Fig. 3), hiPSC-CMs were

incubated with preformed amylin oligomers (diluted to 6.25 μ M) for 2 h at 37 °C.

What is the time course of human aggregation in culture? How can you verify that aggregates are indeed present in your experiments?

Response:

We thank the reviewer for highlighting this important point. In order to evaluate the time course of human aggregation in culture and verify the amylin aggregates, hiPSC-CMs were treated with 6.25 μ M human amylin for 0 h, 1 h, 2 h, 3 h and 4 h, respectively. Amylin aggregates (stained with rabbit anti-amylin antibody, red) and α -actinin (a specific protein in hiPSC-CMs sarcomeres, stained with mouse anti- α -actinin antibody, green) were co-stained (Supplementary Fig. 9a). Western blot analysis with an anti-amylin antibody for assessing lysates from hiPSC-CMs treated as above is shown in Supplementary Fig. 9b. The results showed amylin aggregates were formed time-dependently. The level and size distribution of amylin aggregates were assessed by western blot as shown in molecular weight bands corresponding to amylin trimers (12 kDa), tetramers (16 kDa), hexamers (24 kDa), octamers (32 kDa), 12-mers (48 kDa), 16-mers (64 kDa) and 20-mers (80 kDa). We have added related sentences in the discussion section (page 12, line 309-310).

Inserted sentences (page 12, line 309-310):

“showed that amylin formed aggregates in a time-dependent manner (Supplementary Fig. 9).”

What is known about the presence of amylin receptors in the hiPSCCMs?

Response:

We thank the reviewer for this valuable question. Functional amylin receptors are composed of heterodimers of calcitonin receptors (CTR) and one of three receptor activity-modifying proteins (RAMP 1-3), which gives rise to at least six different subtypes of amylin receptors. However, no study evaluating the presence of amylin receptors in hiPSC-CMs has been reported. We believe this would be a very interesting topic for a further study in our laboratory. In addition, amylin receptors are widely expressed in the central nervous system of NHPs. (Paxinos G. et al., *J Chem Neuroanat*, 2004,27:217-236; Eftekhari S. et al., *J Comp Neurol*,2016,524:90-118). Previous findings demonstrated that RAMP1 and RAMP3 mRNA expression levels were elevated in the atria and ventricles 6 months post-surgery in a rat HF model (Cueille C, *Biochem Biophys Res Commun*, 2002,294:340-346). Increased RAMP2 mRNA levels in the heart were also observed in a rat model of isoproterenol-induced myocardial hypertrophy and ischemia (Qi YF, et al., *Peptides*,2003,24:463-468). Given that RAMPs form a key part of amylin receptors, we postulate that the expression levels of amylin receptors are very likely altered in HF NHPs. We have added related sentences in the discussion section (page 13-14, line 349-354).

Inserted sentences (page 13-14, line 349-354):

“Studies have also reported that amylin receptors, comprised of heterodimers of calcitonin receptors (CTR) and one of three receptor activity-modifying proteins (RAMP1-3), are widely expressed in the central nervous system of NHPs. Additionally, mRNA expression levels of RAMP1, RAMP2 and RAMP3 are elevated in different rat heart dysfunction models. Given that RAMPs form a key part of amylin receptors, we infer that these amylin receptors may also

contribute to Ca²⁺ signaling.”

3. Provide more information on the NHPs. Diet, housing conditions, sex? Was there something specific that predisposed these specific NHPs to HF? While it doesn't appear statistically significant, is the difference in age between the control and HF group relevant?

Response:

We appreciate these comments and have added the diet and housing conditions in the method section (page 15, line 388-392).

Inserted sentences (page 15, line 388-392):

“All animals were housed in an Association for Assessment and Accreditation of Laboratory Animal Care (AAALAC)–accredited facility under light-, temperature- and humidity-controlled conditions, with 3 meals per day, and water provided *ad libitum*. CTL animals were fed a normal calorie diet. Meanwhile, NHPs with HF were fed the vendor's proprietary high fat diet for 2 years before developing HF.”

The information about age, gender, body weight and BMI for all NHPs whose heart tissues were used in this study are listed in Supplementary Table 1. We have added “Age, gender, body weight and BMI data for the 13 NHPs used in this study are included in Supplementary Table 1.” in the method section (page 15, line 396-397).

We believe NHPs with HF showing significantly higher body weights and BMIs are mainly because these animals were fed high fat diet (total energy: 4.15 kcal/g with 1.20 mg/kcal cholesterol) for 2 years. This may not be due to an increase in age because the body weights and BMIs of aged healthy controls (more than 14 years old) were similar to those of younger counterparts (less than 10 years old) (Supplementary Table 1). High fat diet could be a contributing cause to the HF observed in the NHP group in this study.

We certainly agree that aging is potential risk factor for the whole-body system, especially the heart. In our study, we postulate no direct link between cardiac amylin deposition or cardiac impairment and age, as neither amylin deposition nor marked myocardial abnormality was observed in the hearts from both young (less than 10 years old, n = 3) and aged controls (more than 14 years old, n = 4) among the healthy NHPs (revised Fig. 1a), although healthy controls were relatively younger compared with the HF group.

Minor comments:

1. Briefly introduce the pluripotent stem cell-CMs and isolated rat ventricular CMs as a model in introduction.

Response:

Thanks for your suggestion. We have added sentences about RVCMs and hiPSC-CMs in the introduction section (page 4, line 74-76).

Inserted sentences (page 4, line 74-76):

“Isolated rat ventricular cardiomyocytes (RVCMs), derived from rat ventricle, and human induced pluripotent stem cell-cardiomyocytes (hiPSC-CMs) retain their physiological functions and are widely used in *in vitro* models to study cardiotoxicity”.

2. For the presentation of results, always have CTRL data in the first position (discussed first, pictured in the first/top/left position.) Notably in Fig. 1.

Response:

We appreciate this comment and agree with the reviewer that this would be the best way for the reader to understanding the content. We have made the related changes in Fig. 1 and Fig. 2 of the revised manuscript.

3. Fig. 1: Show higher magnification photos of critical pathologies as insets.

Response:

We appreciate this comment and understand the reviewer's concern. We have provided images with higher resolutions in Fig. 1 and Fig. 2 of the revised manuscript.

4. Fig. 2: Was amylin in the HF islet different than control?

As above, how were these antibodies validated?

Response:

Thanks for your insightful comments. We have assessed amylin levels in pancreas samples from CTL and HF NHPs. Pancreatic amylin levels in the HF group were relatively higher compared with those of the CTL group, although not significantly (Supplementary Fig. 2a-b). The specificity of mouse anti-amylin antibody has been tested (Figure D, page 23).

Figure D:

Specificity test for mouse anti-amylin antibody (sc-377530, Santa Cruz)

- a) IHC staining for mouse anti-amylin antibody (sc-377530, Santa Cruz) in HEK-293 cells transfected with human amylin plasmid. Scale bar, 100 μ m.
- b) IHC staining for mouse anti-amylin antibody (sc-377530, Santa Cruz) in HEK-293 cells transfected with empty vector. Scale bar, 100 μ m.
- c) Western blot analysis of mouse anti-amylin antibody (sc-377530, Santa Cruz) in non-transfected (A) and human amylin transfected (B) HEK293T whole cell lysates (This image is downloaded from it's official website).

5. Fig. 3: Why are the group numbers different from the Tables?

Would be helpful if the points in the scatter plots were labeled CTRL or HF.

Response:

We appreciate your question and have modified the related sentences in the results section (page 6, line 144-145; page 7, line 159-160) as well as in the legend of Fig. 3.

Modified sentence (page 6, line 144-145):

“amylin levels in a parallel set of commercial sera from CTL and HF NHPs (whose heart tissues were not available).”

Modified sentence (page 7, line 159-160):

“ELISA of matched plasma and RBC lysates from thirteen NHPs (CTL group, n = 7; HF group,

n = 6, whose heart sections were used for histopathological evaluation in Fig. 1 and amylin assessment in Fig. 2)”

Modified sentence in the legend of Fig. 3:

“Fig. 3. Circulating amylin and HF markers levels in NHPs. (a) Amylin concentration in serum from a parallel set of NHPs (CTL, n = 10; HF, n = 7). (b) Protein levels of ST2, GDF-15, and cTnI in serum as in panel (a). (c-e) The correlations between serum amylin level and ST2 (c), cTnI (d), and GDF-15 (e), respectively, in serum as in panel (a) and (b). Amylin levels in matched plasma (f) and RBC lysates (g) from the same animals whose heart were used for histopathological evaluation in Fig. 1 and amylin assessment in Fig. 2 (CTL, n = 7; HF, n = 6). Data represent mean \pm SEM. *P<0.05; **P<0.01 by Student’s t test. The correlations between serum amylin level and ST2 (c), cTnI (d), and GDF-15 (e), respectively, was analyzed by Spearman nonparametric correlation analysis in GraphPad 7.04 (GraphPad Inc., San Diego, CA), and the values for the Spearman r are indicated on the plots.”

We have also labeled the HF samples in the scatter plots in Fig. 3c-e in the revised Fig. 3.

6. Fig. 4: Why are there so many data points in a and d?

Response:

Thanks for your question. In the bar graphs a and d, we have accidentally put all triplicate readings of RT-PCR data. We have calculated the average readings and corrected the graphs in Fig. 4 a and Fig. 4d (new Fig. 4e) in the revised manuscript.

7. Fig. 6: Label each individual graph a through f.

As in Figure 4, there are inconsistencies in sample size (n = 3) and what is depicted graphically. Please make clearer how the individual data points were derived.

Response:

Thanks for your advice. Per your advice, we have labeled individual graphs a through f in Fig. 6. We have also modified the legend for Fig. 6 and defined sample size in each panel.

8. Fig. 7: Please label individual graphs/panels for ease of reading. As in Fig. 6, there’s inconsistency in sample size (n = 3) and what is depicted graphically. Please make clearer how the individual data points were derived.

Response:

Thanks for your advice. Per your advice, we have labeled individual graphs a through j in Fig. 7. We have also modified the legend for Fig. 7 and defined sample size in each panel.

9. Define MMP at first use.

Response:

We appreciate your comment. We have defined mitochondrial membrane potential (MMP) at first use on page 10, line 251.

10. Fig. 8: Define group sizes.

Response:

We appreciate your comment. We have defined group sizes in each figure legend.

Reviewers' comments:

Reviewer #1 (Remarks to the Author):

The authors goal was to address the question what is drives the accumulation of Amylin during heart failure in non-human primates and how this amylin may regulate cardiac metabolism and function. The authors showed that non-human primates (NHP) with heart failure (HF) had increase cardiac and circulating Amylin levels in the setting of changes consistent with systolic HF which contributed to increases in the expression of the metabolic proteins, HIF1 α and PFKFB3 and which may contribute to changes in Ca²⁺ levels and parameters of mitochondrial function. The authors have been responsive to the previous review; however, a few concerns exist, which would need to be addressed.

Major comments

[1] Although additional information was included in supplementary table 1 regarding the NHP used in this study, the differences in age between control and HF should be addressed in the discussion as the NHP with HF are significantly older than the control NHPs. Normal diet and high fat diet compositions should be included (i.e. fat, carbohydrate, protein contents, etc.). Were all of the HF NHP in both the histological studies and echocardiography studies given high fat diet for 2 years? Did the HFD NHP have any additional systemic pathology? In addition, the newly included NHP demographic data suggest that differences may exist regarding weight between sexes (although the sample size is small). Therefore, sex differences cannot be ruled out with regard to amylin accumulation and its metabolic effects during HF and needs to be discussed as there are documented differences in HF with respect to sex and it has been shown by Liu et al that cardiac Amylin accumulation is greater in males.

[3] Since HF and changes in metabolism will impact the activity/ expression of other PFK isoenzymes, these potential changes should be discussed. Given the changes in lactate in response to amylin treatment, are there changes in Ldh activity/ expression and potential changes in PKM2 activity? Does amylin accumulation in nucleus impact any additional nuclear driven processes. These should also be discussed. An additional important discussion topic would be examination of cardiac amylin levels and HIF1/PFKFB3 with regard to the hypoxia and HF induced by myocardial infarction

[4] Do you have data examining diastolic function in the NHP with HF? Since increased ST2 levels are observed in NHP with HF and increased ST2 levels have been associated with diastolic dysfunction. This should be discussed. In addition, in lieu of actual heart weights, do you have LV corrected mass from the echocardiographic studies?

Minor comments

[1] Line 69 – briefly elaborate on the inherent differences in amylin actions between species. In addition, need to discuss the potential differences between human, NHP, and rat experiments since for example PFKFB3 staining in IPSC cells shows a nuclear position which is not the case in rat cardiomyocytes.

[2] Supplementary Fig 1 – include CTL images.

[3] With regard to mRNA expression data, please reference the housekeeping gene used for normalization in the y-axis.

[4] Figure 2a and Supplemental figure 2a are the same image with different magnifications but scale bar is the same. This should be indicated, or alternative example images used. In addition, supp fig 2 would benefit from a higher degree of magnification to appreciate the changes. Further, what is being quantified in Figure 2 panels J and K. Please clarify.

[5] Line 185- should read "protein levels (Fig 4g and h)". Please adjust.

[6] Figures 4f and supplemental figure 7d are included but are not discussed in the manuscript text.

[7] For phosphorylation of PFKFB3, it would be beneficial to present phospho-to-total ratio.

[8] Line 206- should this be PFKFB3 rather than PFKFB

[9] Provide additional information on the Amy-KO mice. Is this a cardiomyocyte KO or global KO?

Global KO could have compensatory or additional non-cardiac effects.

[10] Figure 8a – does not depict Ca²⁺ flux changes rather steady state concentrations, please relabel.

[11] Line 435- All the staining figures suggest DAPI was used as the nuclear stain but methods state that Hoechst 33342 was used. Please correct.

[12] Line 427 – please include the dilution of phospho-PFKFB3 used

[13] Lines 513-515- details of the non-parametric correlation analysis performed in figure 3 should be added to statistics section.

[14] Figure 3B- Panel for cTnI does not include data points on the bar chart for the CTL. Please address.

[15] Figure 4- show image quantification and either provide higher magnification or highlight specific mentioned changes in the image.

[16] Figure 6- was the following condition examined: CTL +AB? If so, please include. Figure 6c – siRNA for HIF1a only seemed to reduce mRNA levels to that of control – why is this? What happens with PFKFB3 if you inhibit HIF1a?

[17] The representative image for figure 7 do not reflect changes in bar charts. Please address.

[18] The authors state that amylin could increase the calcium transient but ruled out changes in SERCA, PLB, and NCX as contributing factors as they would be slow to act. Only expression was examined which does not correlate to activity. What was the rationale for performing cardiomyocyte experiments at room temperature?

[19] Some of the data presented only in the rebuttal letter would benefit from being addressed in the discussion i.e PKC as potential mechanism.

[20] It may be worth to not include the amylin staining measured by vlna green as the staining is weak and looks like background fluorescence and Dab staining has been used. If Vlna green is included in order to appreciate any staining in HF samples, control samples are needed.

Reviewer #2 (Remarks to the Author):

The report by Liu and colleagues investigated a mechanism by which amylin aggregates in cardiomyocytes (CMs) could induce heart failure (HF). For their studies, the authors analyzed tissue from non-human primates with or without HF, and used two in vitro methods to determine if amylin treatment upregulates HIF1a pathways in CMs. In general the manuscript was well-written and easy to follow. The connection among HF, amylin aggregates and HIF1a is a novel finding and the authors have made a thorough effort to improve their manuscript.

I have two minor points that still require some clarification.

1) I would still like more information on the "high fat diet." What was the macronutrient content of the diet (% kcal from fat)? Can you please provide the vendor?

2) Regarding my suggestion to label the scatter plots in Figure 3, is it possible to label the data points with color scheme used in the other figures? Using black to represent CTRL and orange to represent HF?

Reviewers' comments:

Reviewer #1 (Remarks to the Author):

The authors goal was to address the question what is drives the accumulation of Amylin during heart failure in non-human primates and how this amylin may regulate cardiac metabolism and function. The authors showed that non-human primates (NHP) with heart failure (HF) had increase cardiac and circulating Amylin levels in the setting of changes consistent with systolic HF which contributed to increases in the expression of the metabolic proteins, HIF1 α and PFKFB3 and which may contribute to changes in Ca²⁺ levels and parameters of mitochondrial function. The authors have been responsive to the previous review; however, a few concerns exist, which would need to be addressed.

Major comments

[1] Although additional information was included in supplementary table 1 regarding the NHP used in this study, the differences in age between control and HF should be addressed in the discussion as the NHP with HF are significantly older than the control NHPs. Normal diet and high fat diet compositions should be included (i.e. fat, carbohydrate, protein contents, etc.). Were all of the HF NHP in both the histological studies and echocardiography studies given high fat diet for 2 years? Did the HFD NHP have any additional systemic pathology? In addition, the newly included NHP demographic data suggest that differences may exist regarding weight between sexes (although the sample size is small). Therefore, sex differences cannot be ruled out with regard to amylin accumulation and its metabolic effects during HF and needs to be discussed as there are documented differences in HF with respect to sex and it has been shown by Liu et al that cardiac Amylin accumulation is greater in males.

[3] Since HF and changes in metabolism will impact the activity/ expression of other PFK isoenzymes, these potential changes should be discussed. Given the changes in lactate in response to amylin treatment, are there changes in Ldh activity/ expression and potential changes in PKM2 activity? Does amylin accumulation in nucleus impact any additional nuclear driven processes. These should also be discussed. An additional important discussion topic would be examination of cardiac amylin levels and HIF1/PFKFB3 with regard to the hypoxia and HF induced by myocardial infarction

[4] Do you have data examining diastolic function in the NHP with HF? Since increased ST2 levels are observed in NHP with HF and increased ST2 levels have been associated with diastolic dysfunction. This should be discussed. In addition, in lieu of actual heart weights, do you have LV corrected mass from the echocardiographic studies?

Minor comments

- [1] Line 69 – briefly elaborate on the inherent differences in amylin actions between species. In addition, need to discuss the potential differences between human, NHP, and rat experiments since for example PFKFB3 staining in IPSC cells shows a nuclear position which is not the case in rat cardiomyocytes.
- [2] Supplementary Fig 1 – include CTL images.
- [3] With regard to mRNA expression data, please reference the housekeeping gene used for normalization in the y-axis.
- [4] Figure 2a and Supplemental figure 2a are the same image with different magnifications but scale bar is the same. This should be indicated, or alternative example images used. In addition, supp fig 2 would benefit from a higher degree of magnification to appreciate the changes. Further, what is being quantified in Figure 2 panels J and K. Please clarify.
- [5] Line 185- should read “protein levels (Fig 4g and h)”. Please adjust.
- [6] Figures 4f and supplemental figure 7d are included but are not discussed in the manuscript text.
- [7] For phosphorylation of PFKFB3, it would be beneficial to present phospho-to-total ratio.
- [8] Line 206- should this be PFKFB3 rather than PFKFB
- [9] Provide additional information on the Amy-KO mice. Is this a cardiomyocyte KO or global KO? Global KO could have compensatory or additional non-cardiac effects.
- [10] Figure 8a – does not depict Ca²⁺ flux changes rather steady state concentrations, please relabel.
- [11] Line 435- All the staining figures suggest DAPI was used as the nuclear stain but methods state that Hoechst 33342 was used. Please correct.
- [12] Line 427 – please include the dilution of phospho-PFKFB3 used
- [13] Lines 513-515- details of the non-parametric correlation analysis performed in figure 3 should be added to statistics section.
- [14] Figure 3B- Panel for cTnI does not include data points on the bar chart for the CTL. Please address.
- [15] Figure 4- show image quantification and either provide higher magnification or highlight specific mentioned changes in the image.
- [16] Figure 6- was the following condition examined: CTL +AB? If so, please include. Figure 6c – siRNA for HIF1a only seemed to reduce mRNA levels to that of control – why is this? What happens with PFKFB3 if you inhibit HIF1a?
- [17] The representative image for figure 7 do not reflect changes in bar charts. Please address.
- [18] The authors state that amylin could increase the calcium transient but ruled out changes in SERCA, PLB, and NCX as contributing factors as they would be slow to act. Only expression was examined which does not correlate to activity. What was the rationale for performing cardiomyocyte experiments at room temperature?
- [19] Some of the data presented only in the rebuttal letter would benefit from being addressed in the discussion i.e PKC as potential mechanism.
- [20] It may be worth to not include the amylin staining measured by vlna green as the staining is weak and looks like background fluorescence and Dab staining has been used. If Vlna green is included in order to appreciate any staining in HF samples, control samples are needed.

Reviewer #2 (Remarks to the Author):

The report by Liu and colleagues investigated a mechanism by which amylin aggregates in cardiomyocytes (CMs) could induce heart failure (HF). For their studies, the authors analyzed tissue from non-human primates with or without HF, and used two in vitro methods to determine if amylin treatment upregulates HIF1a pathways in CMs. In general the manuscript was well-written and easy to follow. The connection among HF, amylin aggregates and HIF1a is a novel finding and the authors have made a thorough effort to improve their manuscript.

I have two minor points that still require some clarification.

1) I would still like more information on the "high fat diet." What was the macronutrient content of the diet (% kcal from fat)? Can you please provide the vendor?

2) Regarding my suggestion to label the scatter plots in Figure 3, is it possible to label the data points with color scheme used in the other figures? Using black to represent CTRL and orange to represent HF?

Response to Reviewer #1

We appreciate these highly informative and constructive comments. We have revised and improved our manuscript accordingly. Our responses to the specific comments are as follows.

Reviewers' comments:

Reviewer #1 (Remarks to the Author):

The authors goal was to address the question what is drives the accumulation of Amylin during heart failure in non-human primates and how this amylin may regulate cardiac metabolism and function. The authors showed that non-human primates (NHP) with heart failure (HF) had increase cardiac and circulating Amylin levels in the setting of changes consistent with systolic HF which contributed to increases in the expression of the metabolic proteins, HIF1 α and PFKFB3 and which may contribute to changes in Ca²⁺ levels and parameters of mitochondrial function. The authors have been responsive to the previous review; however, a few concerns exist, which would need to be addressed.

Major comments

[1] Although additional information was included in supplementary table 1 regarding the NHP used in this study, the differences in age between control and HF should be addressed in the discussion as the NHP with HF are significantly older than the control NHPs. Normal diet and high fat diet compositions should be included (i.e. fat, carbohydrate, protein contents, etc.). Were all of the HF NHP in both the histological studies and echocardiography studies given high fat diet for 2 years? Did the HFD NHP have any additional systemic pathology?

Response:

We appreciate these comments and certainly agree that age is a major determinant of the risk for HF and overall cardiovascular disease. We have added “We also observed that NHPs with HF were significantly older than the control NHPs (Table 2, Supplementary Table 1), a finding consistent with what has been reported that age is a major determinant of the risk for HF and overall cardiovascular disease in humans” in the discussion section on page 11, line 281-283. We thank the reviewer for pointing this out and the opportunity to correct the miscalculated statistics for age in Table 2.

We have included the diet composition used in this study in the method section (page 17, line 440-443).

Modified sentences (page 17, line 440-443):

“normal calorie diet (3.81 kcal/g of total energy, 33% of calories from protein, 14% of calories from fat, 53% of calories from carbohydrate). Meanwhile, NHPs with HF were fed the vendor’s proprietary high fat diet (4.15 kcal/g of total energy, 12% of calories from protein, 32% of calories from fat, 56% of calories from carbohydrate, with 1.20 mg/kcal cholesterol) for 2 years before developing HF.”

All the NHPs with HF in both the histological studies and echocardiography studies were given high fat diet for 2 years before developing to HF.

As the reviewer expected, additional systemic pathologies were also observed in some of these HF NHPs treated with high fat diet such as lipid accumulation in the aorta, arteriopathy in the aorta, brain, lung as well as other major organs, vacuolation in the liver, hyperplasia in the pancreatic islets, mononuclear cells infiltration and glomerulopathy in the kidney. We have added related sentences for the histopathological findings in the discussion section (page 11, line 275-279).

Inserted sentences (page 11, line 275-279):

“It is known that high fat diet negatively affects multiple organs and tissues such as the aorta, heart, lungs, livers, kidneys and pancreas. Consistent with the previous findings, we observed lipid accumulation in the aorta, arteriopathy in the aorta, brain, lung as well as other major organs, vacuolation in the liver, hyperplasia in the pancreatic islets, mononuclear cells infiltration, and glomerulopathy in the kidneys in some of the HF NHPs treated with high fat diet.”

In addition, the newly included NHP demographic data suggest that differences may exist regarding weight between sexes (although the sample size is small). Therefore, sex differences cannot be ruled out with regard to amylin accumulation and its metabolic effects during HF and needs to be discussed as there are documented differences in HF with respect to sex and it has been shown by Liu et al that cardiac Amylin accumulation is greater in males.

Response:

We thank the reviewer for highlighting this point. As pointed out by the reviewer, sex differences exist in HF and it impacts almost every aspect of HF from epidemiology and risk factors to pathophysiology (Lam CSP. et al., *Eur Heart J*, 2019,40(47):3859-3868c). Men are predisposed to HF with reduced EF, whereas more women have HF with preserved EF (Savarese G and D'Amario D, *Adv Exp Med Biol*, 2018,1065:529-544). Our results (Supplementary Table 1) also indicated that differences may exist regarding weight between sexes (although the sample size is small) in NHPs. Indeed, it has been reported that cardiac amylin level was higher in males Amy-KO mice infused with human amylin, compared to females (Liu M. et al., *Biochim Biophys Acta Mol Basis Dis*, 2018,1864:1923-1930), and female HIP rats developed diabetes later in life compared to males (Ly H. et al., *Ann Neurol*, 2017,82(2):208-222), indicating a sex-dependent effect in amylin-induced pathology. The previous data (Zhao H. et al., *Pancreas*, 2008,37(3):e68-e73) showed greater pancreatic amylin deposition in men compared to women, most likely due to increased insulin resistance in men (Kahn SE. et al., *Diabetes*, 1993,42(11):1663-1672; Geer EB and Shen W, *Gend Med*, 2009,6 Suppl 1(Suppl 1):60-75). What's more, sex differences were demonstrated in cardiomyocyte ion channels (Zhu Y. et al., *Pflugers Arch*, 2013,465(6):805-818; Yan S. et al., *PLoS One*, 2011,6:e25455; Baraj as-Martinez H. et al., *Cardiovasc Res*, 2009,81:82-89; Sims C., et al., *Circ Res*, 2008,102:e86-e100), intracellular Ca²⁺ handling (Wasserstrom JA. et al., *Am J Physiol Heart Circ Physiol*, 2008,295:H1895-H1904; Farrell SR., et al., *Am J Physiol Heart Circ*

Physiol, 2010,299:H36-H45; Liu M. et al., *Biochim Biophys Acta Mol Basis Dis*, 2018,1864:1923-1930), contractile functions (Farrell SR., et al., *Am J Physiol Heart Circ Physiol*, 2010,299:H36-H45; Parks RJ and Howlett SE, *Pflugers Arch*, 2013, 465(5):747-763.) and cardiac metabolism (Lagranha CJ. et al. *Circ Res*, 2010,106:1681-1691; Wang F. et al., *Hypertension*, 2010, 55:1172-1178; John C. et al., *Front Endocrinol (Lausanne)*, 2018;9:732), which were typically linked to human amylin induced effects. Following the reviewer's suggestion, we have added related sentences in the discussion section (page 11-12, line 283-293).

Inserted sentences (page 11-12, line 283-293):

“Sex differences exist in HF and it impacts almost every aspect of HF from epidemiology and risk factors to pathophysiology. Men are predisposed to HF with reduced EF, whereas more women have HF with preserved EF. Our results (Supplementary Table 1) indicated that differences may exist on weight between sexes (although the sample size is small) in NHPs. Indeed, it has been reported that cardiac amylin level was higher in males Amy-KO mice infused with human amylin, compared to females, and female HIP rats developed diabetes later in life compared to males, indicating a sex-dependent effect in amylin-induced pathology. The previous data showed greater pancreatic amylin deposition in men compared to women, most likely due to increased insulin resistance in men. What's more, sex differences have been reported existing in cardiomyocyte ion channels, intracellular Ca²⁺ handling, contractile functions, and cardiac metabolism, which were typically linked to human amylin induced effects.”

[3] Since HF and changes in metabolism will impact the activity/ expression of other PFK isoenzymes, these potential changes should be discussed.

Response:

We appreciate these comments and agree with the reviewer that further discussion would help us to get a better understanding of this study. PFKFB3 allosterically activates its downstream isoenzyme phosphofructokinase 1 (PFK1), the rate-limiting enzyme of glycolysis (Depre C. et al., *Circulation*, 1999, 99(4):578-588; Gibb AA. et al., *Biochem J*, 2017; 474 (16): 2785-2801), through its product fructose 2,6 biphosphate (F2,6BP), a potent activator of PFK1. PFK1 functions as a gatekeeper to glycolysis and its activity is tightly controlled by AMP, ADP, ATP, and citrate (Mor I, Cheung EC, Vousden KH. *Cold Spring Harb Symp Quant Biol*, 2011,76:211-216). In the failing heart, increased intracellular free AMP and ADP in the cardiomyocytes consequently transduce signaling through AMPK (Ingwall JS. *Cardiovasc Res*, 2009, 81(3):412-419), leading to the enhanced synthesis of F2,6BP, an adaptive response to cardiac pressure overload whereas the production of ATP is impaired. Therefore, the acceleration of glycolytic flux is attributed to an activation of F2,6BP and PFK-1 by both an increase of AMP, an activator of PFK-1, and a decrease of ATP, an inhibitor of the enzyme. Per the reviewer's suggestion, we have added related sentences in the discussion section (page 14, line 350-358).

Inserted sentences (page 14, line 350-358):

“PFKFB3 allosterically activates its downstream isoenzyme phosphofructokinase 1 (PFK1), the rate-limiting enzyme of glycolysis, through its product fructose 2,6 biphosphate (F2,6BP),

a potent activator of PFK1. PFK1 functions as a gatekeeper to glycolysis and its activity is tightly controlled by adenosine monophosphate (AMP), adenosine diphosphate (ADP), ATP and citrate. In the failing heart, increased intracellular free AMP and ADP in the cardiomyocytes consequently transduce signaling through AMPK, leading to the enhanced synthesis of F2,6BP, an adaptive response to cardiac pressure overload whereas the ATP production is impaired. Thus, the acceleration of glycolytic flux in HF could be partly attributed to an activation of F2,6BP and PFK-1 by both an increase of AMP, an activator of PFK-1, and a decrease of ATP, an inhibitor of the enzyme.”

Given the changes in lactate in response to amylin treatment, are there changes in Ldh activity/ expression and potential changes in PKM2 activity?

Response:

We thank the reviewer for this valuable suggestion. We found the elevated level of glycolytic end product lactate in amylin stressed hiPSC-CMs, consistent with the upregulated lactate dehydrogenase (LDH) gene expression observed in HIP rat pancreatic islets (Montemurro C. et al., *Nat Commun*, 2019,10(1):2679). It has also been reported that human amylin evoked LDH release and enhanced LDH activity in rat pancreatic insulinoma beta-cells, human islets cells, and human brain vascular pericytes (Tripathi S and Jeremic AM. *J Biol Chem*, 2011, 286(41):36086-36097; Tripathi S, Jeremic AM, *Plos One*, 2013,8(9): e73080). Thus, we expect a similar effect of human amylin on LDH in hiPSC-CMs and isolated RVCs. Pyruvate kinase M2 (PKM2), regulating the final rate-limiting step of glycolysis, is known to reduce pyruvate kinase activity and promote the glycolytic pathway in the failing heart and the activation of HIF1 α could lead to an induction of PKM2 (Rees ML. et al., *Biochem Biophys Res Commun*, 2015,459(3):430-436). The upregulated PKM2 in HF is most likely due to the lower enzymatic activity of PKM2 that disfavors oxidative phosphorylation, a maladaptation to hypoxia. PKM2 can then be modified by signaling proteins and posttranslational modifications to adjust its enzymatic activity to favor higher proliferation or energy production as needed (Williams AL, et al., *Physiol Genomics*, 2018,50(7):479-494). Based on these findings, we postulate that there might be a dysregulation of PKM2 expression and activity in cardiomyocytes treated with amylin as increased PKM2 gene expression was also observed in pancreatic islets from HIP rats (Montemurro C. et al., *Nat Commun*, 2019,10(1):2679). Following the reviewer’s suggestions, we have added related sentences in the discussion section (page 16, line 414-426).

Inserted sentences (page 16, line 414-426):

“We found the elevated level of glycolytic end product lactate in amylin stressed hiPSC-CMs, consistent with the upregulated lactate dehydrogenase (LDH) gene expression observed in HIP rat pancreatic islets. It has been reported that human amylin evoked LDH release and enhanced LDH activity in rat pancreatic insulinoma beta-cells, human islets cells and human brain vascular pericytes. Therefore, we expect a similar effect of human amylin on LDH in hiPSC-CMs and isolated RVCs. Pyruvate kinase M2 (PKM2), regulating the final rate-limiting step of glycolysis, is known to reduce pyruvate kinase activity and promote the glycolytic pathway in the failing heart and the activation of HIF1 α could lead to an induction of PKM2. The upregulated PKM2 in HF is most likely due to the lower enzymatic activity of PKM2 that disfavors oxidative phosphorylation, a maladaptation to hypoxia. PKM2 can then be modified

by signaling proteins and posttranslational modifications to adjust its enzymatic activity to favor higher proliferation or energy production as needed. Based on these findings, we postulate that there might be a dysregulation of PKM2 expression and activity in amylin stressed cardiomyocytes as increased PKM2 gene expression was also observed in pancreatic islets from HIP rats”

Does amylin accumulation in nucleus impact any additional nuclear driven processes. These should also be discussed.

Response:

We thank the reviewer for this insightful question. Amylin deposits are often seen at sites with myocyte multinucleation and variation in nuclear size. Amylin deposit in the nucleus is capable of inducing hypertrophic transcriptional effects, such as activation of Ca²⁺/calmodulin-dependent protein kinase II (CaMKII)-histone deacetylase (HDAC) and calcineurin-nuclear factor of activated T cells (NFAT) hypertrophic pathways. The presence of nuclear amylin might be a further driver for HIF1 α /PFKFB3 activation in the nuclear because a previous finding reported that nuclear HIF1 α and PFKFB3 levels were both increased in islets from T2D patients and HIP rats (Montemurro C. et al., *Nat Commun*, 2019,10(1):2679), which could be attributed partly to the nuclear amylin deposition. We have added related sentences in the discussion section (page 12, line 317-320; page 14, line 368-369).

Inserted sentences (page 12, line 317-320):

“where amylin deposit in nucleus is capable of inducing hypertrophic transcriptional effects, such as activation of Ca²⁺/calmodulin-dependent protein kinase II (CaMKII)-histone deacetylase (HDAC) and calcineurin-nuclear factor of activated T cells (NFAT) hypertrophic pathways”

Inserted sentences (page 14, line 368-369):

“This was supported by the finding that nuclear HIF1 α and PFKFB3 levels were both increased in islets from T2D patients and HIP rats, which could be attributed partly to the nuclear amylin deposition.”

An additional important discussion topic would be examination of cardiac amylin levels and HIF1/PFKFB3 with regard to the hypoxia and HF induced by myocardial infarction.

Response:

We appreciate your authoritative advice. Myocardial infarction (MI) is one the most common cause of HF, defined as heart muscle necrosis secondary to prolonged lack of oxygen and nutrient supply (ischemia) (Sandoval Y and Thygesen K, *Clin Chem*, 2017,63(1):101-107). HIF1 α was reported to increase in ischemic heart tissues (Lee SH. et al., *N Engl J Med*, 2000, 342,626-633) and cardiac hypertrophy (Krishnan J. et al., *Cell Metab*, 2009,9:512-524,) in humans. Meanwhile, PFKFB3, a target gene of HIF1 α , has low basal expression levels but is strongly induced by hypoxia upon myocardial ischemia (Minchenko O. et al., *FEBS lett*, 2003,554:264-270). Interestingly, significant amylin accumulation was observed in human ischemic failing heart (Despa S. et al., *Circ Res*, 2012,110(4):598-608). In our study, we found elevated levels of serum cTnI (Fig. 3), the preferred biomarker for myocardial injury and MI

(Sandoval Y and Thygesen K, *Clin Chem*, 2017,63(1):101-107), circulating amylin (Fig. 3), and cardiac HIF1 α /PFKFB3 (Fig. 4) in NHPs with HF. Further study is required to establish a causal link between cardiac amylin and HIF1 α /PFKFB3 regarding the hypoxia and HF induced by MI. We have added related discussion in the discussion section (page 13, line 340-341; page 13, line 343-347).

Inserted sentences (page 13, line 340-341):

“Myocardial infarction (MI) is one the most common cause of HF, defined as heart muscle necrosis secondary to prolonged lack of oxygen and nutrient supply (ischemia).”

Inserted sentences (page 13, line 343-347):

“Interestingly, significant amylin accumulation was observed in the human ischemic failing heart. In our study, we found elevated levels of serum cTnI (Fig. 3), the preferred biomarker for myocardial injury and MI, circulating amylin (Fig. 3), and cardiac HIF1 α /PFKFB3 (Fig. 4) in NHPs with HF. Further study is required to establish a causal link between cardiac amylin and HIF1 α /PFKFB3 regarding the hypoxia and HF induced by MI.”

[4] Do you have data examining diastolic function in the NHP with HF? Since increased ST2 levels are observed in NHP with HF and increased ST2 levels have been associated with diastolic dysfunction. This should be discussed. In addition, in lieu of actual heart weights, do you have LV corrected mass from the echocardiographic studies?

Response:

We appreciate the reviewer for these intriguing comments. We added the data for the MV E/A ratio (the ratio of mitral peak E-wave velocity and mitral peak A-wave velocity) and LV mass in Table 1. We observed an obvious decrease in the MV E/A ratio, an important parameter for diastolic function, in NHPs with HF. However, this change does not appear particularly relevant to this study, because they remain within the normal reference range for humans and other animals (Nakayama S. et al., *Exp Anim*, 2020,69(3):336-344). Interestingly, as the reviewer pointed out, increased ST2 levels are observed in NHP with HF (Fig. 3b), and increased ST2 levels have been associated with diastolic dysfunction (Ojji, D. et al. *J Hum Hypertens*,2014,28,432-437). This concept is supported by our finding of significantly increased LVDP and EDV (Table 1), characteristics of diastolic function, in NHPs with HF. Besides, elevated LV mass (Table 1), an indicator of hypertrophy, is also evidenced to be a strong predictor for diastolic dysfunction (Kimura H. et al., *Nephron Clin Pract*, 2011,117,c67-c73). We have made related changes in the discussion section (page 12, line 308-line 315; line 317).

Inserted sentences (page 12, line 308-line 315):

“It is reported that increased ST2 level was associated with diastolic dysfunction. Hence, our present data suggest that amylin oligomer accumulation may accelerate the onset of diastolic dysfunction. This concept is supported by our finding of significantly increased LVDP and EDV (Table 1), characteristics of diastolic function, in NHPs with HF. Although we observed an obvious decrease in the MV E/A ratio (the ratio of mitral peak E-wave velocity and mitral peak A-wave velocity) in NHPs with HF (Table 1), an important parameter for diastolic function,

this change does not appear particularly relevant to this study, because they remain within the normal reference range for humans and other animals. Besides, elevated LV mass (Table 1), an indicator of hypertrophy, is also evidenced to be a strong predictor for diastolic dysfunction.”

Modified sentences (page 12, line 317):

“and diastolic dysfunction where amylin deposits in nucleus is”

Minor comments

[1] Line 69 – briefly elaborate on the inherent differences in amylin actions between species. In addition, need to discuss the potential differences between human, NHP, and rat experiments since for example PFKFB3 staining in iPSC cells shows a nuclear position which is not the case in rat cardiomyocytes.

Response:

Thanks for your suggestion. We have made related changes in the Introduction section (page 4, line 64) and in the discussion section (page 13, line 331-333).

Modified sentences (page 4, line 64):

“to inherent differences such as pharmacology and receptor splice variants that exist between the species.”

Inserted sentences (page 13, line 331-333):

“Interspecies differences in physiology and genetics would account for some of the differences in results such as PFKFB3 staining in NHPs (Fig. 4f) and hiPSC-CMs (Fig. 5h) shows a nuclear position which is not the case in RVCMs (Fig. 7e).”

[2] Supplementary Fig 1 – include CTL images.

Response:

We appreciate your advice. We have included CTL images in Supplementary Fig. 1.

[3] With regard to mRNA expression data, please reference the housekeeping gene used for normalization in the y-axis.

Response:

We thank the reviewer for pointing this out and have made changes in related figures (Fig. 4-7)

[4] Figure 2a and Supplemental figure 2a are the same image with different magnifications but scale bar is the same. This should be indicated, or alternative example images used. In addition, supp fig 2 would benefit from a higher degree of magnification to appreciate the changes. Further, what is being quantified in Figure 2 panels J and K. Please clarify.

Response:

Thanks for your suggestion. We have included new images with higher degree of magnification in Supplementary Fig. 2 and made changes in the results section (page 6, line 121-122) and in the figure legends of Fig. 2g.

Inserted sentences (page 6, line 121-122):

“Fig. 2g presented the relative positive signal intensity for amylin in heart samples from NHPs

with HF and CTL groups.”

[5] Line 185- should read “protein levels (Fig 4g and h)”. Please adjust.

Response:

Thank you for pointing this out. We have corrected this error.

[6] Figures 4f and supplemental figure 7d are included but are not discussed in the manuscript text.

Response:

We appreciate your questions, and have added related sentences in the results section (page 8, line 174-175, page 10, line 245-248).

Modified sentences (page 8, line 174-175):

“we performed immunohistochemical staining for PFKFBs in heart sections from NHPs (Fig. 4f and g)

Inserted sentences (page 10, line 245-248):

“Also, illustrative isochronal maps (Supplementary Fig. 7d) showed shortened calcium activation time and CaD90 (the duration of calcium transient at 90% decay) and augmented calcium amplitude in hiPSC-CMs exposed with amylin, indicating Ca²⁺ mishandling in these cells.”

[7] For phosphorylation of PFKFB3, it would be beneficial to present phospho-to-total ratio.

Response:

Thanks for your insightful suggestion. We have made the change in the new Fig. 4l.

[8] Line 206- should this be PFKFB3 rather than PFKFB

Response:

Thank you for pointing this out. We have corrected this error.

[9] Provide additional information on the Amy-KO mice. Is this a cardiomyocyte KO or global KO? Global KO could have compensatory or additional non-cardiac effects.

Response:

We thank the reviewer for this valuable question. The Amy-KO mouse was generated via the CRISPR/Cas9 gene editing technique targeting mouse IAPP exon 3 in the C57BL/6J strain, resulting in a deletion of IAPP exon 3 coding sequence (gRNA1: 5'-CAGTGTACATAGTCAATGAC-3'; gRNA2: 5'-ATTGTGCATTCTCACTGAGG-3'). Amylin (IAPP) is expressed exclusively in pancreas, therefore, we believe the differences between knock-out and WT are attributed to the loss of amylin in pancreas and this Amy-KO knockout model should not have compensatory effects. We have modified related sentences in the method section (page 17, line 450-452).

Inserted sentences (page 17, line 450-452):

“resulting in a deletion of IAPP exon 3 coding sequence (gRNA1: 5'-CAGTGTACATAGTCAATGAC-3'; gRNA2: 5'-ATTGTGCATTCTCACTGAGG-3').”

[10] Figure 8a – does not depict Ca²⁺ flux changes rather steady state concentrations, please relabel.

Response:

We appreciate this comment and have relabeled the Fig. 8a as the reviewer suggested.

[11] Line 435- All the staining figures suggest DAPI was used as the nuclear stain but methods state that Hoechst 33342 was used. Please correct.

Response:

Thanks for your advice and we have corrected those errors in all the related figures.

[12] Line 427 – please include the dilution of phospho-PFKFB3 used

Response:

We appreciate this suggestion and have included the dilution of anti-phospho-PFKFB3 antibody and moved the use of anti-phospho-PFKFB3 antibody into the “Immunoblotting part of method section” (page 18, line 491-492) since this antibody was only used for western blotting.

[13] Lines 513-515- details of the non-parametric correlation analysis performed in figure 3 should be added to statistics section.

Response:

We thank the reviewer for pointing this out and have added the non-parametric correlation analysis performed in Fig. 3 in the statistics section (page 21, line 562-564).

Inserted sentences (page 21, line 562-564):

“The Spearman nonparametric correlation analysis for Fig. 3c-e were performed in GraphPad (GraphPad Inc., San Diego, CA), and the values for the Spearman r are indicated on the plots”.

[14] Figure 3B- Panel for cTnl does not include data points on the bar chart for the CTL. Please address.

Response:

Thanks for your advice and we have included data points of CTL in panel for cTNI (Fig. 3b).

[15] Figure 4- show image quantification and either provide higher magnification or highlight specific mentioned changes in the image.

Response:

We appreciate your advice. We have provided higher magnification for IHC images in new Fig. 4b and 4f and also included the IHC image quantification for the positive signal of HIF1 α and PFKFB3 in new Fig. 4c and 4g.

[16] Figure 6- was the following condition examined: CTL +AB? If so, please include. Figure 6c – siRNA for HIF1a only seemed to reduce mRNA levels to that of control – why is this? What happens with PFKFB3 if you inhibit HIF1a?

Response:

We appreciate your comments and thanks for the opportunity to clarify our findings. As the reviewer suggested, we have included the results of CTL+AB in new Fig. 6a and 6b. The results of CTL+AB showed that AB alone treatment did not affect the mRNA level of HIF1 α and PFKFB3. To answer the reviewer’s question about Fig. 6c, we have added additional data for

the mRNA level of HIF1 α and PFKFB3 in hiPSC-CMs treated with siCTL, siHIF1 α or siPFKFB3 alone, in new Fig. 6c and d. For the siRNA treatment experiments, hiPSC-CMs were pre-treated with different siRNAs for 72 hours, and then were incubated with or without human amylin for another 2 hours. In new Fig. 6c, we observed that significantly decreased HIF1 α mRNA level when hiPSC-CMs were treated with siHIF1 α alone (magenta color) compared to hiPSC-CMs treated with siCTL only (grey color) or hiPSC-CMs pre-treated with siHIF1 α then treated with H-amy (purple color). In new Fig. 6d, we observed that the PFKFB3 mRNA level was decreased in hiPSC-CMs treated with siHIF1 α alone (magenta color) compared to hiPSC-CMs treated with siCTL only (grey color). We found that in our experiments, siPFKFB3 can totally block the PFKFB3 mRNA transcription when the cells were pre-treated siPFKFB3 and then treated with or without H-amy (Fig. 6d, siPFKFB3+H-amy group, light blue color; siPFKFB3, dark blue color). We observed that siHIF1 α significantly reduced the HIF1 α mRNA level induced by amylin (siCTL+H-amy group, green color compared to the siHIF1 α +H-amy group, purple color). However, the efficiency for siHIF1 α is relatively low compared to siPFKFB3 as we still observed elevated HIF1 α mRNA in the cells pre-treated siHIF1 α and then treated with H-amy compared to the cells were treated with siHIF1 α only (Fig. 6c, siHIF1 α +H-amy group, purple color; siHIF1 α , magenta color). We believe this is mainly because different siRNAs targets had different transfect efficiency. We observed similar results from 3 different siHIF1 α s (siRNA ID: s6539, Cat# 4390824; siRNA ID: s131712, Cat# 4390771; siRNA ID: s131712, Cat# 4390771, Thermofisher). Another most likely reason is that hiPSC-CMs are known to be difficult to transfect.

[17] The representative image for figure 7 do not reflect changes in bar charts. Please address.

Response:

We thank the reviewer for pointing this out and have replaced the representative image in Fig. 7a.

[18] The authors state that amylin could increase the calcium transient but ruled out changes in SERCA, PLB, and NCX as contributing factors as they would be slow to act. Only expression was examined which does not correlate to activity. What was the rationale for performing cardiomyocyte experiments at room temperature?

Response:

We thank the reviewer's comment. We have modified the related sentences (page 15, line 379-383).

Modified sentences (page 15, line 379-383):

“although these Ca²⁺ handling proteins are known as contributing factors in response to amylin stress. We postulate that amylin oligomers could acutely elevate Ca²⁺ transients, while altered expressions of SERCA, PLB, and NCX may require longer-term effects, as NCX expression was significantly decreased in NHPs with HF (Supplementary Fig. 11). Further experiments are needed to study the direct connection between the activities of Ca²⁺ handling proteins and amylin.”

The experiments for hiPSC-CMs are performed at 37 °C. For the isolated RVCs experiments, we followed the detailed protocol provided by those publications (Despa S. et al., Circ Res,

2012,110(4):598-608, supplemental material; Despa, S, et al., J Am Heart Assoc.2014;3:e001015) where the human amylin treatment in isolated RVCMs are performed at room temperature. Their data (Fig. A, below) showed the presence of human amylin monomers, dimers and trimers in the lysates of RVCMs treated with 50 μ M human amylin for 2 h at room temperature. Confocal fluorescence images (Fig. B, below) also showed human amylin binds to the myocyte membrane after 2h of amylin treatment at room temperature.

[19] Some of the data presented only in the rebuttal letter would benefit from being addressed in the discussion i.e PKC as potential mechanism.

Response:

We appreciate the reviewer’s suggestions and have added the data of PKC as a potential mechanism in Supplementary Fig. 10. We have inserted related sentences in the discussion section (page 13, line 334-339) and in the method section (page18-19, line 492-495).

Inserted sentences (page 13, line 334-339):

“Besides, we observed significantly increased levels of protein kinase C (PKC) in HF NHPs. Meanwhile, no protein expression alterations were found for adenosine monophosphate (AMP)-activated protein kinase (AMPK) and protein kinase A (PKA) (Supplementary Fig. 10). Therefore, increased expression of PKC may contribute to the phosphorylation of PFKFB3-Ser461 in HF NHPs, consistent with a previous finding that PKC regulates the PFKFB3 isoenzyme by covalent modification of its C-terminal domain”

Inserted sentences (page18-19, line 492-495):

“rabbit anti-AMPK antibody (1:1000, 2532s, Cell Signaling Technology, Danvers, MA), rabbit anti-PKC antibody (1:1000, 2056S, Cell Signaling Technology, Danvers, MA), rabbit anti-PKA antibody (1:1000, 4782S, Cell Signaling Technology, Danvers, MA),”

[20] It may be worth to not include the amylin staining measured by vina green as the staining is weak and looks like background fluorescence and Dab staining has been used. If Vina green is included in order to appreciate any staining in HF samples, control samples are needed.

Response:

We thank the reviewer’s comments and agreed with the reviewer that the via green staining is weak and looks like background fluorescence. We have removed the vina green staining images as suggested.

Response to Reviewer #2

We thank the reviewer for the very positive and comprehensive appraisal of our work. We have revised and improved our manuscript accordingly. Our responses to the specific comments are as follows.

Reviewer #2 (Remarks to the Author):

The report by Liu and colleagues investigated a mechanism by which amylin aggregates in cardiomyocytes (CMs) could induce heart failure (HF). For their studies, the authors analyzed tissue from non-human primates with or without HF, and used two in vitro methods to determine if amylin treatment upregulates HIF1a pathways in CMs. In general the manuscript was well-written and easy to follow. The connection among HF, amylin aggregates and HIF1a is a novel finding and the authors have made a thorough effort to improve their manuscript.

I have two minor points that still require some clarification.

- 1) I would still like more information on the "high fat diet." What was the macronutrient content of the diet (% kcal from fat)? Can you please provide the vendor?

Response:

We appreciate the reviewer's comments and have included the diet composition used in this study in method section (page 17, line 440-443). The vendors are Wuxi AppTec and Kunming Biomedical International which are stated in the method section (page 16, line 436).

Modified sentences (page 17, line 440-443):

"normal calorie diet (3.81 kcal/g of total energy, 33% of calories from protein, 14% of calories from fat, 53% of calories from carbohydrate). Meanwhile, NHPs with HF were fed the vendor's proprietary high fat diet (4.15 kcal/g of total energy, 12% of calories from protein, 32% of calories from fat, 56% of calories from carbohydrate, with 1.20 mg/kcal cholesterol) for 2 years before developing HF."

- 2) Regarding my suggestion to label the scatter plots in Figure 3, is it possible to label the data points with color scheme used in the other figures? Using black to represent CTRL and orange to represent HF?

Response:

We thank the reviewer for pointing this out. We have corrected the scatter plots with color scheme in Fig. 3 as the reviewer suggested.

REVIEWERS' COMMENTS:

Reviewer #1 (Remarks to the Author):

The authors goal was to address the question what is drives the accumulation of Amylin during heart failure in non-human primates and how this amylin may regulate cardiac metabolism and function. The authors showed that non-human primates (NHP) with heart failure (HF) had increase cardiac and circulating Amylin levels in the setting of changes consistent with systolic HF which contributed to increases in the expression of the metabolic proteins, HIF1 α and PFKFB3 and which may contribute to changes in Ca²⁺ levels and parameters of mitochondrial function. The authors have been responsive to the previous review(s) through the inclusion of relevant data and revisions to the discussion which have greatly improved the manuscript.

Minor comments

- [1] Line 46- define abbreviation HIP at first use
- [2] Lines 205-207- mention that Hif1 α and PFKFB3 were examined in Amy-KO hearts
- [3] Lines 439-443- Provide company details and product numbers for the high fat diets used
- [4] Line 468, 472- There appears to an issue with the following symbol "oC"
- [5] Figure 2- panels C and F look like they are the same magnification, but the scale bars are the same. If they are the same magnification, please revise.
- [6] Figure 7- the y axis for panels D and H need to mention Gadph i.e. Hif1 α /Gapdh or PFKFB3/Gadph
- [7] List specific PKC isoform targeted i.e. PKCa

REVIEWERS' COMMENTS:

Reviewer #1 (Remarks to the Author):

The authors goal was to address the question what is drives the accumulation of Amylin during heart failure in non-human primates and how this amylin may regulate cardiac metabolism and function. The authors showed that non-human primates (NHP) with heart failure (HF) had increase cardiac and circulating Amylin levels in the setting of changes consistent with systolic HF which contributed to increases in the expression of the metabolic proteins, HIF1 α and PFKFB3 and which may contribute to changes in Ca²⁺ levels and parameters of mitochondrial function. The authors have been responsive to the previous review(s) through the inclusion of relevant data and revisions to the discussion which have greatly improved the manuscript.

Minor comments

[1] Line 46- define abbreviation HIP at first use

[2] Lines 205-207- mention that Hif1a and PFKFB3 were examined in Amy-KO hearts

[3] Lines 439-443- Provide company details and product numbers for the high fat diets used

[4] Line 468, 472- There appears to an issue with the following symbol "oC"

[5] Figure 2- panels C and F look like they are the same magnification, but the scale bars are the same. If they are the same magnification, please revise.

[6] Figure 7- the y axis for panels D and H need to mention Gadph i.e. Hif1a/Gadph or PFKFB3/Gadph

[7] List specific PKC isoform targeted i.e. PKCa

Response to Reviewer #1

We thank the reviewer for the very positive and comprehensive appraisal of our work. We have revised and improved our manuscript accordingly. Our responses to the specific comments are as follows.

Reviewers' comments:

Reviewer #1 (Remarks to the Author):

The authors goal was to address the question what is drives the accumulation of Amylin during heart failure in non-human primates and how this amylin may regulate cardiac metabolism and function. The authors showed that non-human primates (NHP) with heart failure (HF) had increase cardiac and circulating Amylin levels in the setting of changes consistent with systolic HF which contributed to increases in the expression of the metabolic proteins, HIF1 α and PFKFB3 and which may contribute to changes in Ca²⁺ levels and parameters of mitochondrial function. The authors have been responsive to the previous review(s) through the inclusion of relevant data and revisions to the discussion which have greatly improved the manuscript.

Minor comments

[1] Line 46- define abbreviation HIP at first use

Response:

Thanks for your suggestion. In our study, HIP stands for human amylin in the pancreas. The abbreviation of HIP has been defined and clarified in line 43 (page 3, line 43).

[2] Lines 205-207- mention that Hif1a and PFKFB3 were examined in Amy-KO hearts

Response:

We appreciate your advice. We have made related changes in line 204 (page 9, line 204).

[3] Lines 439-443- Provide company details and product numbers for the high fat diets Used

Response:

We thank the reviewer for pointing this out. We have included the company details in line 439-440 (page 17, line 439-440). The high fat diet is the company's proprietary and they don't have a product number for it.

[4] Line 468, 472- There appears to an issue with the following symbol "oC"

Response:

Thank you for pointing this out. We have corrected all degree symbol errors in line 464, line 468, line 472, line 534, line 538, line 541, line 551, and line 558.

[5] Figure 2- panels C and F look like they are the same magnification, but the scale bars are the same. If they are the same magnification, please revise.

Response:

We appreciate your question. In Figure 2, image for panel c was taken at 40x magnification and image for panel f was taken at 100x magnification (oil immersion). They were not taken at the

same magnification.

[6] Figure 7- the y axis for panels D and H need to mention Gadph i.e. Hif1a/Gapdh or PFKFB3/Gadph

Response:

Thanks for your insightful suggestion. We have made the changes in the Fig. 7d and 7h. We also made related changes in Fig. 4e, 4j, 5f, and 5k.

[7] List specific PKC isoform targeted i.e. PKCa

Response:

We appreciate this comment and have listed the specific PKC isoform targeted (PKC α) in line 492 (page 18) and in Supplementary Fig. 10 as the reviewer suggested. We also listed the specific AMPK isoform (AMPK α) and PKA isoform (PKA C- α) in line 491, line 493 (Page 18) and in Supplementary Fig. 10.